



# Benchmarking Data-Driven Rainfall-Runoff Models in Great Britain: A comparison of LSTM-based models with four lumped conceptual models

Thomas Lees[1], Marcus Buechel[1], Bailey Anderson[1], Louise Slater[1], Steven Reece[2], Gemma Coxon[3], and Simon J. Dadson[1,4]

[1]School of Geography and the Environment, University of Oxford, South Parks Road, Oxford, United Kingdom, OX1 3QY
[2]Department of Engineering, University of Oxford, Oxford, United Kingdom
[3]Geographical Sciences, University of Bristol, Bristol, United Kingdom
[4]UK Centre for Ecology and Hydrology, Maclean Building, Crowmarsh Gifford, Wallingford, United Kingdom, OX10 8BB

**Correspondence:** Thomas Lees (thomas.lees@chch.ox.ac.uk)

**Abstract.** Long short-term memory models (LSTMs) are recurrent neural networks from the emerging field of Deep Learning (DL), which have shown recent promise when predicting time-series especially when data are abundant. Rainfall-runoff modelling presents a challenge, yet accurate hydrological models are vital for flood forecasting, hazard impact assessment, and to assess the potential effects of climate change on floods and water resources. In this study, we compare the performance of two

DL-based models, a LSTM and an Entity Aware LSTM (EA LSTM). The DL models were trained using a newly published data set, CAMELS-GB, for a sample of 518 catchments across Great Britain. To identify spatial and seasonal patterns in model performance, we compare the DL models against benchmark outputs from four lumped conceptual models recently configured for rainfall-runoff modelling in Great Britain. Our findings show that the LSTM models simulate discharge with consistently high model performance scores, including in catchments typically considered difficult to model. The LSTM achieves a mean

catchment NSE of 0.88 (0.86 for the EALSTM), which represents a performance improvement of 10% – 16% compared with the benchmark conceptual models. Seasonal and spatial patterns indicate that the largest performance improvement relative to the benchmark is in the drier summer months and in drier catchments in the South East of England. By comparing LSTMs with conceptual models, we diagnose possible reasons for their different performance. We suggest that LSTMs offer useful predictive capability for rainfall-runoff modelling in Great Britain and elsewhere and note their value to support process

understanding in locations where processes are less well understood.

## 1 Introduction

Rainfall-runoff models have evolved over many decades, reflecting a diversity of applications and purposes. These models range from physically based, spatially explicit models such as SHE (Abbott et al., 1986) and CLASSIC (Crooks et al., 2014),



to lumped conceptual models, such as TOPMODEL (Beven and Kirkby, 1979) and VIC (Liang, 1994), and data-driven models (Reichstein et al., 2019; Elshorbagy et al., 2010; Wilby et al., 2003; Nourani et al., 2014; Le et al., 2019; Gauch et al., 2021). Physical and conceptual models can struggle with certain catchments and hydrological conditions which do not conform to the assumptions of the analyst's underlying perceptual model. For example, modelling difficulties may arise in catchments where the effects of groundwater or non-diffuse macropore flow dominate (Wheater et al., 2007), where observations poorly

constrain the water balance (Beven, 2020), or in river basins where the topographic water catchment is not the appropriate surface across which water is conserved and there are inter-catchment transfers of water through groundwater processes (Liu et al., 2020). Finding generally-applicable model structures has long posed a challenge for hydrological sciences (Linsley, 1982), and models require considerable effort to build, calibrate and maintain. It has been suggested that techniques from machine learning might offer promising predictive capability (Reichstein et al., 2019), particularly in situations where rainfall

and river flow data are plentiful, yet where the appropriate perceptual models of surface and subsurface hydrological processes are poorly understood.

     Artificial Neural Networks (ANN) have shown skill when modelling complex and highly nonlinear systems. Due to the stacking of multiple connected layers, these models are often referred to as "Deep Learning" (DL) models. DL methods have been used in hydrology and meteorology since the early 1990s (Daniell, 1991; Halff et al., 1993; Dawson and Wilby, 1998;

Wilby et al., 2003; Peel and McMahon, 2020). For many environmental applications, including rainfall-runoff modelling, temporal structure in the data is important. Yet simple, feed-forward neural networks cannot capture information about the sequential nature of time series. By contrast, Recurrent Neural Networks (RNNs) aim to account for temporal dependence using a series of recurrent layers which incorporate new information at each time-step, and pass processed information as input to the next layer in the model. In this way, information is retained in the model over time, a feature which is important

when simulating time-series with persistence (e.g., in meteorology and hydrology) (Hochreiter et al., 2001). Hochreiter (1991) overcame problems with traditional RNNs by proposing a novel architecture to account for long-term dependencies. Long-Short Term Memory Networks (LSTMs) have an explicit memory state which is updated through a series of gates to model these long-term dependencies. LSTMs have been used successfully for speech recognition (Graves et al., 2013) and natural language processing (Wang and Jiang, 2015). More recently, they have been applied in hydrology. What follows is a partial

review of recent studies using DL in hydrology. For a more complete picture on the uses of DL techniques in hydrology, an interested reader is referred to Shen (2018); Beven (2020); Nearing et al. (2020).

     Kratzert et al. (2018) showed that an LSTM trained on 241 catchments in the US achieved similar performance metrics to the Sacramento Soil Moisture Accounting model coupled with a snow model across the USA (Burnash et al., 1973). When the LSTM was calibrated on all basins, it outperformed various benchmark models (Kratzert et al., 2019). The authors set out

to address how to make predictions in ungauged basins, transferring knowledge from one basin to another. In doing so they introduced a new LSTM architecture, the Entity Aware LSTM (EA LSTM). The "entities" refer to the spatial units (catchments) defined and measured by catchment attributes that are time invariant in the input data, such as topography, mean climate conditions and land-cover characteristics. The EA LSTM learned a high-dimensional vector that represents how the catchment attributes condition the relationship between the dynamic forcings (rainfall, temperature, potential evapotranspiration etc.)





and the outputs (specific discharge). Therefore, the EA LSTM explicitly conditions the discharge response to meteorological forcing on time-invariant properties of river catchments, such as soil and topographic attributes. While the EA LSTM offered new potential for interpreting what the model had learned, model performance suffered when compared with that of the LSTM (Kratzert et al., 2019).

The performance of the LSTM for reproducing US streamflow has been further demonstrated by (Duan et al., 2020; Feng
et al., 2020; Gauch et al., 2021). Other studies have considered the uses of the LSTM for producing forecasts of soil moisture, also focused on the US (Fang et al., 2018, 2020). More recent work has begun not only to explore the accuracy of forecasts, but also to use LSTMs to: (i) provide estimates of uncertainty (Klotz et al., 2020); (ii) explore the ability of the LSTM to integrate prior physical knowledge into DL model architectures (Hoedt et al., 2021; Jiang et al., 2020); and (iii) use LSTMs to produce predictions at multiple timescales from a single model (Gauch et al., 2020). Gauch et al. (2020) recently demonstrated that the
LSTM can accurately predict water discharge at multiple timescales, such as daily and hourly. In contrast with the US National Water Model (Viterbo et al., 2020), the Multi-TimeScale LSTM (MTS LSTM) showed a considerably smaller performance decrease when predicting hourly instead of daily streamflow. Their approach offers the potential to use LSTMs operationally, ingesting data at different temporal frequencies to produce predictions at a desired resolution.

Taken together, these studies demonstrate that LSTM models have credible, often substantially improved, simulation accu-
racy when compared with traditional conceptual hydrological models. However, they have been predominantly tested in the US hydrological context. In this study we explore the potential for LSTMs to simulate discharge in Great Britain (GB), a temperate climate, using a newly published data set (CAMELS-GB) to train the LSTM models and benchmark their performance against four commonly-used conceptual models. We assess the predictive ability of the LSTM-based models and use them to understand the relationship between model performances and catchment characteristics. We test the LSTM performance under
conditions where observations of precipitation and river flow do not close the water balance for the topographic catchment, noting that under these conditions traditional hydrological models often struggle.

Our study poses the following four research questions: (i) How well do LSTM-based models simulate discharge in a temperate climate like Great Britain? (ii) Can the LSTM overcome limitations of previously calibrated models, producing accurate simulations in regions where hydrological models typically struggle to reproduce hydrological outputs? (iii) Are there hydro-
logical processes that data-driven models simulate better than traditional hydrological models, which can be diagnosed by benchmarking model performance? In turn, these research questions structure the remaining sections of the paper. First we describe the LSTMs' training and evaluation methods in detail. Next we present results at the national scale and then grouped by spatial and seasonal patterns of performance. In Section 4 we consider inter-model performance diagnostics and return to discuss the specific hydrological conditions in which LSTMs outperform traditional models. Finally we conclude with an
indication of further promising research topics motivated by this study.



## 2 Methods

### 2.1 Study Region

We focus our benchmarking study on catchments in Great Britain (GB), a temperate and humid region on the eastern edge of the North Atlantic. An overview of the hydro-meteorological conditions can be found in Figure 1.

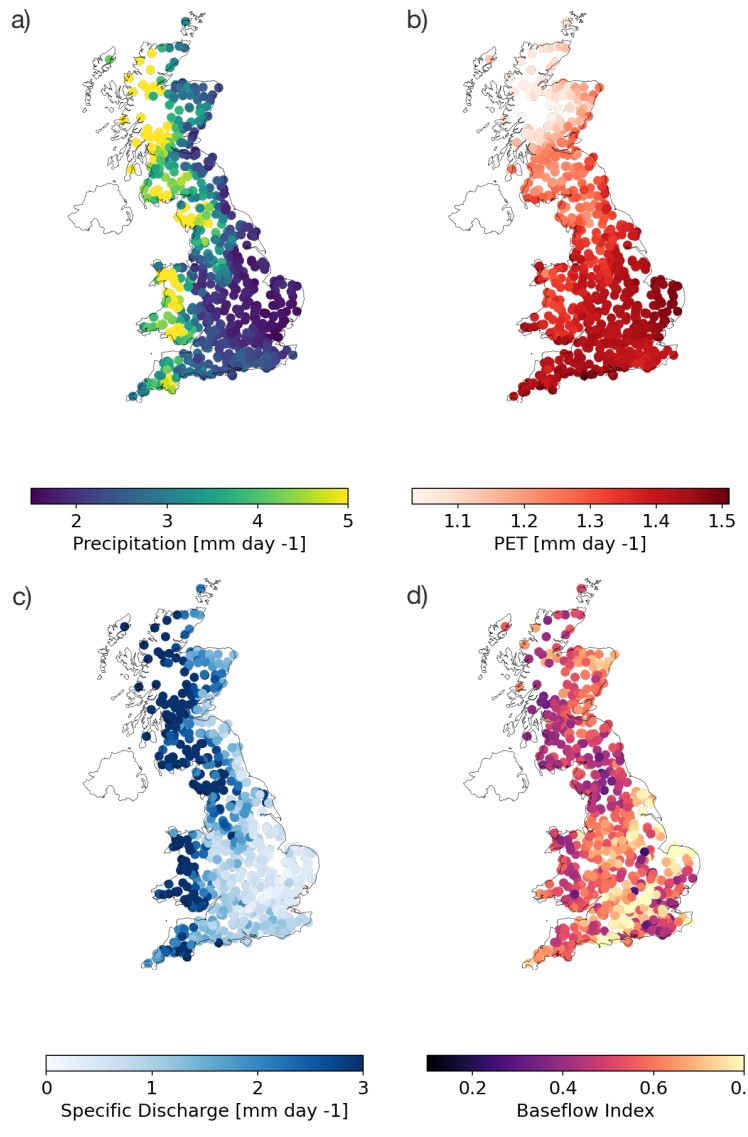

**Figure 1.** Key variables outlining the hydro-meteorological context for 671 stations in Great Britain (Coxon et al., 2020b). a) mean catchment precipitation (mm day$^{-1}$) b) shows the mean catchment potential evapotranspiration (mm day$^{-1}$) c) mean catchment discharge (mm day$^{-1}$) d) baseflow index, the ratio of the mean daily base flow to daily discharge.





Precipitation is higher in the west and north of GB and lowest in the east and south (Figure 1a), as a result of higher elevation and prevailing winds from the west that bring rainfall from the Atlantic. The wettest areas in the north-west of Scotland average 3500 mm yr$^{-1}$ of precipitation. Snow fractions are generally very low across GB, however there are a number of catchments in the Cairngorm mountains in north-east Scotland where the fraction of snow can reach 0.17. The driest areas are found in the South and East of GB, with a minimum of 500 mm yr$^{-1}$ in the East of England (Coxon et al., 2020b). Furthermore, there are a number of large chalk aquifers in the south east, which cause the hydrograph to respond more slowly to rainfall events (Lane et al., 2019). Seasonally, the highest monthly precipitation totals occur in winter months (DJF), and the least precipitation falls during the summer months (JJA). The temperature patterns enhance the availability of moisture. Evaporation losses are concentrated in the summer, from April to September (Lane et al., 2019). Anthropogenic and land-use changes significantly impact river flows (Prosdocimi et al., 2015; Vicente-Serrano et al., 2019). River discharge is most heavily modified in the south east and midland regions of England, in part due to high population density and a long history of human modifications to the environment.

## 2.2   Data - CAMELS GB

All data used in this analysis come from the CAMELS-GB data (Coxon et al., 2020a). CAMELS-GB is a recently-released, large-sample, long-term, daily data set that offers the potential for GB-wide modelling studies. CAMELS-GB collated hydro-logically relevant data for 671 GB catchments, between the years of 1970 and 2015. The data set includes daily time series for meteorology and discharge (dynamic data, $X_t, y_t$). Also included are catchment attributes (static data, $A$) such as topography, climate, hydrologic signatures, soil and land cover, hydrogeology, and human influence. These features are, in reality, not static over time. However, for the purposes of this study we treat these features as time-invariant. Further information on the variables we used as input to our model can be found in Table 2. The reader is directed to Coxon et al. (2020b) for details of the source of the data, how the data were processed and a discussion of data limitations.

The data set contains novel inputs compared with previous CAMELS (US, Chile, Brazil) data sets (Addor et al., 2017; Alvarez-Garreton et al., 2018; Chagas et al., 2020), such as human attributes, calculated potential evapotranspiration (pet) and uncertainty estimates. We do not use all of these features here. The static attributes we use to train the LSTM models are listed in Table 1. These static attributes were chosen to reproduce the experimental framework of Kratzert et al. (2019), however, the differences reflect the fact that the CAMELS-US and CAMELS-GB have slightly different attributes. These include both catchment properties and climate properties, describing the conditions relevant for rainfall-runoff modelling in different catchments.

## 2.3   An Overview of the LSTM and EALSTM

In this paper, we test two neural network architectures used in other hydrological studies (Shen et al., 2018; Kratzert et al., 2019). The first is the LSTM, which has been used in a variety of time-series modelling applications. The second model is the EA LSTM, which conditions the discharge response to meteorological forcings on time-invariant properties of river catchments, such as soil and topographic attributes, treating these time-invariant properties separately.





**Table 1.** Table describing the notation used throughout the paper.

| Symbol | Description | Notes |
|---|---|---|
| $y_t$ | Our target variable, specific discharge at time $t$ | $\mathrm{mm\,day}^{-1}$ |
| $n$ | Gauge ID | - |
| $\mathrm{p}_t$ | Precipitation inputs | $\mathrm{mm\,day}^{-1}$ |
| $\mathrm{peti}_t$ | Potential evapotranspiration and interception | $\mathrm{mm\,day}^{-1}$ |
| $\mathrm{t}_t$ | Temperature | $^\circ\mathrm{C}$ |
| $A$ | Catchment attributes (static data) | See Table 2 |
| $X$ | Hydro-meteorological data (dynamic data) | $[\mathrm{p}_t, \mathrm{peti}_t, \mathrm{t}_t]$ |
| hs | Hidden size | $\mathrm{hs} = 64$ |
| $\theta$ | Learned model parameters, representing all $W_{\mathrm{layer}}$ and $b_{\mathrm{layer}}$ | - |
| $\mathcal{M}_\theta$ | The model (LSTM or EA LSTM) with parameters $\theta$ | - |
| $[\boldsymbol{X}_t, A]$ | Concatenation of dynamic and static input data for a single catchment | - |
| $C_t$ | The cell state of the LSTM models. | $\mathbb{R}^{\mathrm{hs}}$ |
| $\tilde{C}_t$ | The candidate cell state values | $\{\tilde{C}_t \in \mathbb{R} \mid -1 < x < 1\}$ |
| $h_t$ | The hidden state of the LSTM models. | $\mathbb{R}^{\mathrm{hs}}$ |
| $W_{\mathrm{layer}}$ | The matrix of learnable weights | - |
| $b_{\mathrm{layer}}$ | The vector of learnable biases | - |
| $f_t$ | The forget gate of the LSTM models | $\{f_t \in \mathbb{R} \mid 0 < x < 1\}$ |
| $i_t, i$ | The input gate of the LSTM models | $\{i_t \in \mathbb{R} \mid 0 < x < 1\}$ |
| $o_t$ | The output gate of the LSTM models | $\{o_t \in \mathbb{R} \mid 0 < x < 1\}$ |

    The LSTM captures information that is important over both long and short term time horizons, overcoming a key difficulty with traditional RNNs, which are unable to retain information over longer sequences (Hochreiter, 1991; Bengio et al., 1994).

LSTMs do this by maintaining two state vectors, a cell memory vector that captures slowly evolving processes ($C_t$, Equation 5) and a more quickly evolving state vector, colloquially named the "hidden" vector ($h_t$, Equation 6). The cell memory vector $C_t$, accounts for longer-term dependencies, and a series of 'gates' control the information passing into and out of the memory vector. The hidden state vector ($h_t$) evolves more quickly depending on input information and the output of the memory vector (see Figure 2). The gates include: the forget gate ($f_t$), which controls the elements of the cell memory vector that are

forgotten (i.e. how long water persists in the system, Equation 1); the input gate ($i_t$), which controls what information from the new input data at that timestep will be incorporated into the cell memory vector (i.e. what information is stored for future timesteps, Equation 2); and finally the output gate ($o_t$), which determines what information from the cell memory will be used to update the hidden state (i.e. what information will impact discharge at the current timestep, Equation 3). These gates are neural network layers, made up of weights ($W_{\mathrm{layer}}$), biases ($b_{\mathrm{layer}}$) and activation functions. The activation functions allow





the LSTM to model nonlinear processes. During training, we seek the values for these weights and biases that best describe

observed discharge. The information that passes through the input gate to the cell state ($C_t$ - see Equation 5) is itself processed

through a neural network layer, producing a series of candidate values that may be used to update the cell state (Equation 4).

Finally, information from the cell state is passed through the output gate ($o_t$) to produce the hidden output ($h_t$) at that time-step

(Equation 6). Note that for the LSTM we have explicitly defined the inputs as the concatenation of the dynamic meteorological

data and the static catchment attributes, $[\boldsymbol{X}_t, A]$. That is, both LSTM models receive the same information. We refer the reader

to Kratzert et al. (2018) and Kratzert et al. (2019) for comprehensive descriptions of the LSTM and EA LSTM, and their

hydrological interpretation. .

$$\boldsymbol{f}_t = \sigma\left(\mathbf{W}_\mathrm{f}\left[[X_t, A], \boldsymbol{h}_{t-1}\right] + \boldsymbol{b}_\mathrm{f}\right) \tag{1}$$

$$\boldsymbol{i}_t = \sigma\left(\mathbf{W}_\mathrm{i}\left[[X_t, A], \boldsymbol{h}_{t-1}\right] + \boldsymbol{b}_\mathrm{i}\right) \tag{2}$$

$$\boldsymbol{o}_t = \sigma\left(\mathbf{W}_\mathrm{o}\left[[X_t, A], \boldsymbol{h}_{t-1}\right] + \boldsymbol{b}_\mathrm{o}\right) \tag{3}$$

$$\tilde{\boldsymbol{C}}_t = \tanh\left(\mathbf{W}_\mathrm{C}\left[[X_t, A], \boldsymbol{h}_{t-1}\right] + \boldsymbol{b}_\mathrm{C}\right) \tag{4}$$

$$\boldsymbol{C}_t = \boldsymbol{f}_t * \boldsymbol{C}_{t-1} + i * \tilde{C}_t \tag{5}$$

$$\boldsymbol{h}_t = \boldsymbol{o}_t * \tanh(\boldsymbol{C}_t) \tag{6}$$

The EA LSTM was developed specifically for rainfall-runoff modelling (Kratzert et al., 2019). The key difference between

the EA LSTM and the LSTM is that the input gate ($i$) is no longer conditional upon the dynamic (time-varying) data. Instead,

the static (time-invariant) catchment attributes ($A$) exclusively influence the input gate (Equation 2 is replaced with Equation

8), and all other gates are solely influenced by the dynamic input data (Equation 7, 9, 10).

$$\boldsymbol{f}_t = \sigma\left(\mathbf{W}_\mathrm{f}\left[X_t, \boldsymbol{h}_{t-1}\right] + \boldsymbol{b}_\mathrm{f}\right) \tag{7}$$

$$\boldsymbol{i} = \sigma\left(\mathbf{W}_\mathrm{i}A + \boldsymbol{b}_\mathrm{i}\right) \tag{8}$$

$$\boldsymbol{o}_t = \sigma\left(\mathbf{W}_\mathrm{o}\left[X_t, \boldsymbol{h}_{t-1}\right] + \boldsymbol{b}_\mathrm{o}\right) \tag{9}$$

$$\tilde{\boldsymbol{C}}_t = \tanh\left(\mathbf{W}_\mathrm{C}\left[X_t, \boldsymbol{h}_{t-1}\right] + \boldsymbol{b}_\mathrm{C}\right) \tag{10}$$

The EA LSTM is described as "entity-aware" because it explicitly learns how to use catchment attributes ($A$) to distin-

guish between similar dynamic inputs ($X_t$) for different catchments ("entities"). For the EA LSTM, $i$ is determined solely

by the catchment attributes (Equation 8). Therefore, each catchment has one unique $hs$ dimensional vector which controls

what information should persist in future timesteps. In contrast, the LSTM learns to modify the input gate $i_t$ based upon the

meteorological forcing data ($X_t$) *and* the catchment attributes ($A$). The output of the input gate ($i_t$ or $i$) is a vector of values

between 0 and 1, which is learned from data. This vector, also known as an "embedding", translates our catchment attributes

into a high-dimensional space that represents catchments in a manner optimised to differentiate between catchment rainfall-

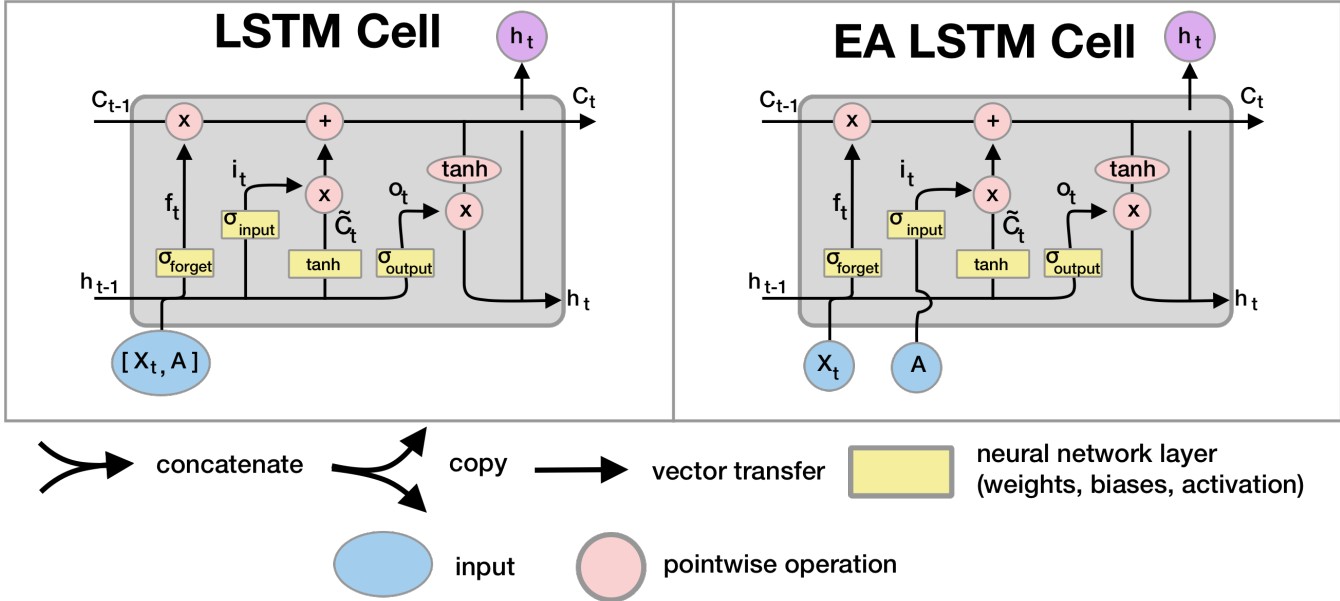

**Figure 2.** Wiring diagram for the LSTM and EA LSTM recurrent cells, adapted from Olah (2016). These cells are repeated for each input timestep in our sequence length, and so for 365 input timesteps, we have 365 cells. The key difference between the EA LSTM and the LSTM is the separation of the static data, $A$, from the dynamic data $X_t$. In the EA LSTM, the static data is the sole input to the input gate, producing an embedding, $i_t$. In both LSTM models there is a cell state $C_t$, that passes from cell to cell, capable of modelling longer-term dependencies. Note that the neural network layers correspond with the weights ($W$), biases ($b$) and activation functions ($\sigma$, $\tanh$). These operations correspond to the yellow layers in the diagram.

runoff behaviours. Kratzert et al. (2019) demonstrated how this embedding represents what the model has learned about our
catchments.

For the sake of clarity, it is important to note that both models receive the same information. The LSTM still receives the static catchment attributes. However, rather than affecting only the input gate, the static data can influence all gates, since they are appended to a vector of dynamic inputs ($[\boldsymbol{X}_t, A]$) and so the same information is given to the LSTM at each timestep. The static attributes are used by the LSTM in the same way as the dynamic data. This offers extra flexibility for the LSTM
compared with the EA LSTM, since the LSTM is able to modify the input gate based on information from time-varying data, whereas the EA LSTM is not. We are using the static nature of the data as a constraint on the EA LSTM to reflect the nature of the input data (separated into static and dynamic inputs - see Figure 2).

### 2.4  Model Training

We used the "neuralhydrology" codebase, written in Python 3.6 (Van Rossum et al., 2007), to train and evaluate the models,
found here: github.com/neuralhydrology/neuralhydrology/. The configuration files used to run the models can be found using





the links at the end of this article. The predictions and error metrics for the fitted models can be found online at Zenodo, zenodo.org/record/4555820.

The goal of rainfall-runoff modelling is to predict time-varying specific discharge, $y_t = (y_1, ..., y_T) \in \mathbb{R}^T$, measured in mm day$^{-1}$, for time $t = \{1, ..., T\} \in \mathbb{N}$ at measuring gauge $n$ of $N$, given hydro-meteorological forcing data, $X = (X_1, ..., X_T)$, and catchment attributes ($A$ - Table 2) within the catchment area upstream of the gauge. In the present case for GB, $N = 669$. The underlying CAMELS-GB data has 671 station gauges. We train on data from only 669 stations because two basins have missing data in the static attributes; stations 18011 and 26006 have missing mean elevation (elev_mean) and mean drainage path slope (dpsbar). The data set, $\mathcal{D}$, therefore consists of:

$$\mathcal{D} = \{([X_{t,n}, A_n], y_{t,n})\}_{t,n=1}^{T,N}. \tag{11}$$

Our task is to learn a single set of parameters, $\theta$, of a model, $\mathcal{M}_\theta$, that minimizes the loss function, $\ell(\hat{y}, y)$, globally, and thus accurately simulates discharge for all of the basins across GB:

$$\hat{y}_t = \mathcal{M}_\theta(A, X_{t-k+1}, \ldots, X_t; \theta). \tag{12}$$

We train our model using the modified Nash Sutcliffe Efficiency (NSE) loss as our objective function ($\ell$), described in Kratzert et al. (2019). Other objective functions could be used, however, we sought to use the same objective function as the conceptual models we compare against, in order to control the possible sources of performance differences. The NSE describes the squared error loss normalized by the total variance of the observations. In order to account for the fact that some basins will have lower variance than others, we follow Kratzert et al. (2019) to normalize by basin-specific variance. This prevents the loss from being overly weighted towards high-variance catchments.

For this study we trained the models on the days from January 1$^{st}$ 1988 December to 31$^{st}$ 1997 and tested on the days from January 1$^{st}$ 1998 to December 31$^{st}$ 2008. We withheld the years 1975 to 1980 from the training process to check the performance of the model during training (our validation set). This means that we have separate time periods for calibration (1988–1997), and test our performance on the hold-out time periods (1998–2008). These train and test periods were chosen in order to compare with the model benchmarking study whose published results for four lumped hydrological models we use as a benchmark (Lane et al., 2019). For further analysis of the train and test periods please see Appendix A.

Our input data were taken from CAMELS-GB, described above (Coxon et al., 2020b). We used precipitation, potential evapotranspiration and temperature as dynamic inputs ($X_t = [\mathrm{p}_t, \mathrm{pet}_t, \mathrm{t}_t]$). We used 21 static inputs ($A$). Each catchment was characterised using 21 individual features describing the topographic, soil, land-cover, and climatic properties. These catchment attributes are described in Table 2. For both LSTM models we pass the final hidden output through a fully connected (linear) layer. This final layer maps our hidden state vector ($\mathbb{R}^{hs}$) to a scalar prediction for the discharge of that gauge on that day





($\hat{y}_t \in \mathbb{R}$). We give the models one year of daily dynamic data (365 input timesteps, $X = [X_{t-365}, \ldots, X_t]$) to predict the final timestep of specific discharge ($\hat{y}_t$).

**Table 2.** Catchment attributes from the CAMELS-GB data set (Coxon et al., 2020b) used to train the LSTM based models, the static features included in $A$.

| Static Variables | Static Variable Description | Median | Range |
|---|---|---|---|
| area | catchment area (m a.s.l) | 152 | [2, 9931] |
| elev_mean | mean elevation (m a.s.l) | 163 | [25, 682] |
| dpsbar | slope of the catchment mean drainage path ($\mathrm{mkm}^{-1}$) | 79 | [12, 488] |
| sand_perc | percent sand (%) | 43 | [19, 82 ] |
| silt_perc | percent silt (%) | 30 | [9, 43] |
| clay_perc | percent clay (%) | 24 | [7, 51] |
| porosity_hypres | soil porosity calculated using the hypres pedotransfer function (-) | 47 | [34, 81] |
| conductivity_hypres | hydraulic conductivity calculated using the hypres pedotransfer function ($\mathrm{cmh}^{-1}$) | 1 | [0.5, 3] |
| soil_depth_pelletier | depth to bedrock (m) | 1 | [0.5, 42] |
| frac_snow | fraction of precipitation falling as snow (for days colder than 0°C) | 0.02 | [0.00, 0.17] |
| dwood_perc | percent of catchment that is deciduous woodland (%) | 6 | [0, 37] |
| ewood_perc | percent of catchment that is evergreen woodland (%) | 2 | [0, 93] |
| crop_perc | percent of catchment that is cropland (%) | 13 | [0.00, 91] |
| urban_perc | percent of catchment that is urban area (%) | 3 | [0.00, 81] |
| reservoir_cap | catchment reservoir capacity (ML) | 0 | [0, 8 x $10^7$] |
| p_mean | mean daily precipitation ($\mathrm{mm\,day}^{-1}$) | 2.57 | [1.54, 9.61] |
| pet_mean | mean daily PET ($\mathrm{mm\,day}^{-1}$) | 1.38 | [1.03, 1.51] |
| p_seasonality | seasonality and timing of precipitation (estimated using sine curves) | -0.14 | [-0.42, 0.14] |
| high_prec_freq | frequency of high-precipitation days ($\geq$ 5x mean daily precipitation) | 15.69 | [7.58, 20.73] |
| low_prec_freq | frequency of dry days (< $1\mathrm{mm\,day}^{-1}$) | | [1.63, 259.23] |
| high_prec_dur | average duration of high-precipitation events ($\geq$ 5x mean daily precipitation) | 1.14 | [1.05, 1.25] |
| low_prec_dur | average duration of dry periods (number of consecutive days < 1 $\mathrm{mm\,day}^{-1}$) | 3.70 | [2.64, 4.67 ] |

We train the LSTM models on 669 gauges with training data from our training period (1988–1997), which was chosen to match with the conceptual model experiments we compare against (Lane et al., 2019). Furthermore, the training data for both the LSTM based models and the conceptual models come from the same underlying sources. In order to make a fair

comparison all national results shown below are calculated for the 518 gauges that are found in both the CAMELS GB data and the benchmark data. We then evaluate model performance on all of these basins for our test (evaluation) period (1998–2008). For each model (LSTM, EA LSTM) we take the average of an ensemble of eight individually-trained models with different





random seeds. This strategy accounts for the random initialisation of the network and the stochastic nature of the optimisation algorithm. We used a hidden size ($hs$) of 64 and a final fully connected layer with a dropout rate of 0.4, which aims to avoid

overfitting. Dropout works by randomly forcing certain weights in the network to zero ("dropping them out"), forcing the remaining weights to model the discharge without that extra information. This has been found to prevent weights 'fixing' the erroneous outputs of other weights, preventing this co-adaptation of weights and, ultimately, encouraging the model to use a simpler and more robust representation of rainfall-runoff processes (Srivastava et al., 2014). We chose the hyper-parameters (dropout rate, hidden size - $hs$) based on the choices in previous studies (Kratzert et al., 2019). We used the Adam optimisation

algorithm (Kingma and Ba, 2014) and stopped training after 30 epochs. The LSTM ensemble took 10 hours to train. The EA LSTM ensemble took 96 hours to train. All models were trained on a machine with 188GB of RAM and a single NVIDIA V100 GPU.

### 2.5 Model Performance Comparisons

The LSTMs learn to represent hydrological processes directly from data. When the LSTMs perform well a necessary corollary

is that the data contains useful information about the hydrological processes. The differences in model performance between the LSTMs and the benchmark hydrological models can be used to determine hydrological processes that are described by the input data, but uncaptured or under-represented by the benchmark hydrological models.

### 2.5.1 Benchmark Models

We compare the performance of the LSTM based models against a range of lumped, conceptual models. We used predicted

discharge time series from Lane et al. (2019) who utilised the FUSE framework to train and evaluate four lumped conceptual models across Great Britain (Clark et al., 2008). The four conceptual models used are: TOPMODEL (Beven and Kirkby, 1979), Variable Infiltration Capacity (VIC) (Liang, 1994), Precipitation-Runoff Modelling System (PRMS) (Leavesley et al., 1983) and SACRAMENTO (Burnash et al., 1973). These conceptual models are often used in operational settings, due to the relative ease of use and lower data requirements when compared with physically-based models (Lane et al., 2019). These conceptual

models all explicitly maintain mass balance, and so assume no losses or gains of water other than flow from the catchment outlet or evaporation.

The calibration and evaluation of these models was performed using the same underlying data as in CAMELS-GB, i.e. the National River Flow Archive data (Centre for Ecology and Hydrology, 2016) for the specific discharge ($y_t$), the Centre for Ecology and Hydrology Gridded Estimates of Areal Rainfall, CEH-GEAR, for precipitation (Tanguy, 2014) and the Climate

Hydrology and Ecology research Support System Potential Evapotranspiration (CHESS-PE) data set for PET (Robinson et al., 2017).

518 basins in the published FUSE simulations overlap with the basins in the CAMELS-GB data used to train the LSTM models. Therefore, results from the LSTM based models and the conceptual models calibrated by Lane et al. (2019) were compared on these 518 basins.





A key difference in the calibration of these models is our separation of training (calibration) and testing (evaluation) periods, since with LSTMs it is trivial to reproduce test results if they have been used for training. Lane et al. (2019) chose parameters for the 4 lumped models from a grid of 10,000 parameters within certain specified ranges. The parameters for the conceptual models were chosen uniquely for each catchment, by comparing performance on the calibration period from 1988–2008. Each parameter set was evaluated using the Nash Sutcliffe Efficiency (NSE). The best parameter set for that catchment-model was

the parameter set with the highest NSE score (Lane et al., 2019). Therefore, their conceptual models were calibrated on the period from 1988–2008, and in that study, they were evaluated on the period 1993–2008. We use the published timeseries of model simulations for each catchment to compare results for the period 1998–2008, which overlaps with the evaluation period used in Lane et al. (2019). We are comparing results against the NSE scores calculated from the published simulated discharge time series, representing the simulations from the models with the best performing parameters (Lane et al., 2019). An important

difference between the LSTMs and the traditional hydrological models, is that traditional models perform best when calibrated for individual basins. The parameters that they use to produce simulations are unique to each basin. This often represents the state-of-the-art for traditional hydrological models. In contrast, the trained LSTM models learn one parameter set for all basins, using all basins to train the models. This represents a difference between testing model performance on hold-out data (as we do for the LSTM models), in contrast with the published results from (Lane et al., 2019), which test model performance on

in-sample data that was also used to select the optimum parameters.

     The conceptual models were calibrated and evaluated to produce simulated streamflows by Lane et al. (2019). We did not run these benchmarks ourselves. This is important because we have not biased the calibration of these models to favour the deep learning models. We have used the published time-series of model outputs to calculate performance scores for the conceptual models. This allows us to better understand the seasonal and geographical patterns in model performance.

Comparing the LSTM based models against these conceptual models allows us to determine the spatial and temporal patterns in performance, helping to identify flow-regimes where the LSTM models add significant value when simulating GB discharge.

### 2.5.2    Evaluation Protocol

Each model produces a daily simulated discharge value at each station. Three example hydrographs are shown in Appendix B. The evaluation protocol described below evaluates the overall performance of each model to reproduce the observed hydro-

graph.

     Since no single evaluation metric can fully capture the performance of streamflow simulations across all flow-regimes (Gupta et al., 1998), we use a number of metrics to address the performance of models across the flow regime, outlined below.

     We evaluate the goodness-of-fit of the LSTM based models and the conceptual models using six evaluation metrics. The Nash-Sutcliffe Efficiency (NSE) (Nash and Sutcliffe, 1970), Equation 13 is perhaps the most widely used performance measure

in hydrology (Ewen, 2011). It has been used for many years and there is extensive literature discussing its strengths and





weaknesses (Gupta et al., 2009). Owing to the squared term in the definition of NSE, it is more heavily influenced by high flows.

$$\text{NSE} = 1 - \frac{\sum_{i=1}^{N} \left(Q_{\text{o,i}} - Q_{\text{s,i}}\right)^2}{\sum_{i=1}^{N} \left(Q_{\text{o,i}} - \bar{Q}_{\text{o}}\right)^2} \tag{13}$$

The NSE can be decomposed into three components, a correlation term (Equation 14), a bias term (BiasError, Equation 15) and a variability (SDError Equation 16) term (Gupta et al., 2009). The bias term measures the error in predicting the mean flow. The variability term measures the error in predicting the standard deviation of discharge. We follow Lane et al. (2019) in using these decomposed aspects of NSE to diagnose the underlying causes of accurate or inaccurate simulations in a given catchment.

$$\text{Correlation} = \frac{\sum_{i}^{N} \left(Q_{s,i} - \bar{Q}_s\right)\left(Q_{o,i} - \bar{Q}_o\right)}{\sigma_{Q_s} \sigma_{Q_o}} \tag{14}$$

$$\text{BiasError} = \frac{\bar{Q}_s - \bar{Q}_o}{\bar{Q}_o} \tag{15}$$

$$\text{StdError} = \frac{\sigma_{Q_s} - \sigma_{Q_o}}{\sigma_{Q_o}} \tag{16}$$

To understand how well the LSTMs represent the different flow exceedences, we also consider the biases for different components of the flow duration curve: low flows, flows in the middle of the flow duration curve and high flows. The low flow bias (%BiasFLV, Equation 17) is the diagnostic signature measure for long term base flow (Yilmaz et al., 2008), and low flows are defined as those which are exceeded 70% of the time. For the middle of the flow duration curve we use the bias of the mid section of the flow duration curve, between the 20th and 70th percentiles (%BiasFMS, Equation 18). Finally, we also look at the bias of the high flows, considering the top 2% of flows (%BiasFHV, Equation 19).

$$\text{BiasFLV} = -1 \times \frac{\sum_{l=1}^{L} \left[\log\left(Q_{s,l}\right) - \log\left(Q_{s,L}\right)\right] - \sum_{l=1}^{L} \left[\log\left(Q_{o,l}\right) - \log\left(Q_{o,L}\right)\right]}{\sum_{l=1}^{L} \left[\log\left(Q_{o,l}\right) - \log\left(Q_{o,l}\right)\right]} \times 100 \tag{17}$$

Where $l = 1, 2, ..., L$ is the index of the flow value within the low-flow segment, defined as 0.7–1.0 flow exceedence probabilities, following (Yilmaz et al., 2008).

$$\text{BiasFMS} = \frac{\left[\log\left(Q_{s_{m1}}\right) - \log\left(Q_{s_{m2}}\right)\right] - \left[\log\left(Q_{o_{m1}}\right) - \log\left(Q_{o_{m2}}\right)\right]}{\left[\log\left(Q_{o_{m1}}\right) - \log\left(Q_{o_{m2}}\right)\right]} \times 100 \tag{18}$$





Where $m1$ corresponds to the lower bound of the middle section ($m1 = 0.2$) and $m2$ corresponds to the upper bound of the
middle section ($m2 = 0.7$), following (Yilmaz et al., 2008)

$$\text{BiasFHV} = \frac{\sum_{h=1}^{H}(Q_{s_h} - Q_{o_h})}{\sum_{h=1}^{H}Q_{o_h}} \times 100 \tag{19}$$

## 3   Results

### 3.1   National Scale Model Performance

The LSTM and EA LSTM models systematically outperform the conceptual lumped models across Great Britain when evalu-
ated using a variety of metrics, with differing levels of performance improvement (See Table 3).

**Table 3.** Summary of all goodness-of-fit metrics used to benchmark performance against the conceptual models for the validation period
1998–2008 on the 518 stations found in both CAMELS-GB data (Coxon et al., 2020a) and the FUSE conceptual models (Lane et al., 2019).
We have shown the median score. Values that are not significantly different from the best model are highlighted in bold ($\alpha = 0.001$).

|  | NSE | BiasError | SDError | Correlation | %BiasFMS | %BiasFLV | %BiasFHV |
|---|---|---|---|---|---|---|---|
| TOPMODEL | 0.76 | -0.04 | -0.10 | 0.88 | **5.70** | 42.22 | -13.04 |
| ARNOVIC | 0.78 | 0.06 | -0.10 | 0.90 | **2.25** | -60.34 | -14.66 |
| PRMS | 0.77 | 0.03 | **-0.03** | 0.89 | 35.24 | -315.25 | -15.11 |
| SACRAMENTO | 0.80 | **-0.01** | -0.07 | 0.90 | 27.91 | -195.92 | -16.19 |
| EALSTM | 0.86 | **-0.02** | -0.10 | 0.94 | -6.29 | **23.61** | -10.81 |
| LSTM | **0.88** | **-0.02** | -0.09 | **0.94** | -3.67 | **26.34** | **-9.09** |

Comparing the median NSE for all catchments, the LSTM (0.88) outperforms all other models, including the EA LSTM
(0.86). The slightly lower median NSE for the EA LSTM models is consistent with results from previous studies (Kratzert
et al., 2019). The difference between the LSTM based models is small relative to the difference between the LSTM based
models and the conceptual models. Of the conceptual models, SACRAMENTO performs best (0.80), followed by ARNOVIC
(0.78), PRMS (0.77) and TOPMODEL (0.76).

The CDFs (cumulative distribution functions) of the NSE (Figure 3a) show the entire distribution of LSTM scores is shifted
towards better performances. The LSTM NSE scores are significantly different from all comparison models at $\alpha = 0.001$
(Wilcoxon signed-rank-test). We see the same pattern for the EA LSTM models, where the distribution of NSE scores is also
different from all other models at the $\alpha = 0.001$ level. The performance improvement at the tails is particularly pronounced.
Neither the LSTM nor the EA LSTM model have any station gauges with an NSE of less than zero. This is in contrast to the
conceptual models where a number of gauges have a NSE of less than zero. Furthermore, at the 5th percentile, the LSTM has
an NSE score of 0.71, the EALSTM 0.68, TOPMODEL 0.37, PRMS -0.45, VIC 0.45 and Sacramento 0.20.





As discussed in the methods, we can decompose the NSE into three components, bias (BiasError), correlation and error in predicting the variability of flows (SDError). The pattern of correlation scores closely follows the pattern of NSE, with the entire distribution of catchment correlation scores shifted towards improved performance. The CDFs in Figure 3c show that the distribution of LSTM catchment bias scores are closer to zero. PRMS and ARNOVIC in particular have a number of stations with larger positive biases. The LSTM distribution of performance scores is significantly different from all models except SACRAMENTO (p=0.3) and EA LSTM (p=0.4). As with all conceptual models, the median variability error is negative (Figure 3d), showing that the LSTM also underpredicts the variability of flows. While PRMS has the smallest median variability error 3, the distribution of LSTM variability error scores show the LSTM has fewer overpredictions of variability, since a much smaller proportion of stations have a SDError greater than 0.2, although the EA LSTM has a number of large underpredictions of variability.

Looking at different segments of the flow duration curve, the LSTM models outperform the conceptual models at low flows (Table 3). The LSTM shows a much greater performance improvement for low-flow bias score (%BiasFLV). This can be seen in the empirical CDFs (Figure 3e), where the conceptual models have a large proportion of stations with biases more extreme than -100%. By contrast, the LSTM based models have very few (<5%). This finding is interesting because the performance of the LSTM at low-flows was previously identified as an area for further research and future improvement (Kratzert et al., 2019; Gauch et al., 2020) and indeed, we find that the LSTM based models outperform conceptual models on this low flow metric.

Comparing median scores, the LSTM has lower median bias in the slope of the midsection of the flow duration curve (%BiasFMS) than all models except ARNOVIC. When we consider the CDFs, both LSTMs have much shorter tails than the conceptual models, showing that a greater proportion of catchments have biases closer to zero. The high-flow biases (%BiasFHV) are relatively similar for all models, as shown by Figure 3g), although the median scores show that there is a small performance improvement.

Overall, the biases at different flow exceedances suggest that the conceptual models produce adequate simulations for the high flows, but are less able to simulate low flows. The LSTM shows a much smaller performance decline at the low flows and a competitive performance at high flows, suggesting that the LSTMs are more robust to extreme conditions. We also note that the negative bias, for the midsection and the upper-section of the flow duration curve, demonstrate that the LSTM model is conservative in its flow predictions.



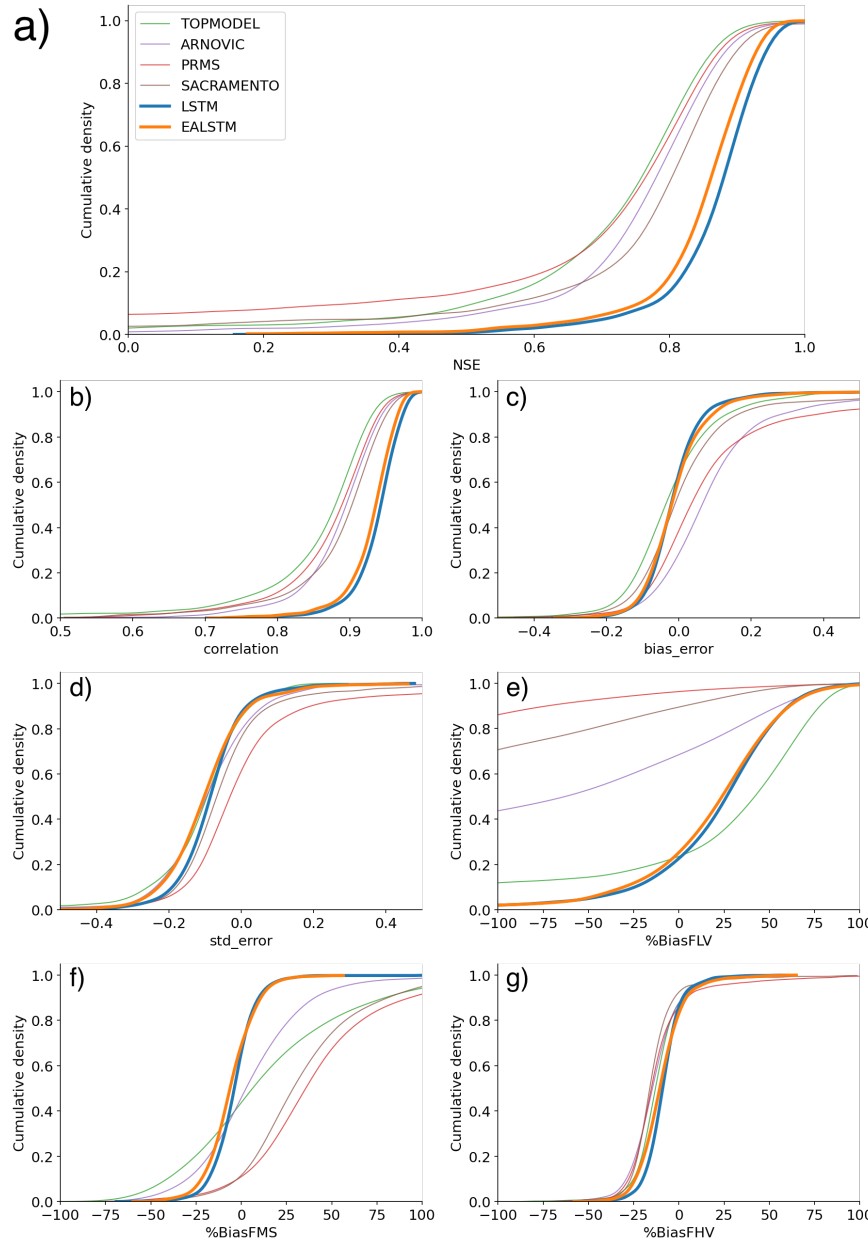

**Figure 3.** Cumulative Distribution Functions (CDFs) of station goodness-of-fit metrics scores for each model. EALSTM (orange) and the LSTM (blue), and the conceptual models: TOPMODEL (green), VIC (red), PRMS (purple), Sacramento (brown) (Lane et al., 2019). Panels indicate distribution of station: a) NSE scores b) correlation scores c) bias error scores d) variability error scores e) low-flow bias scores f) mid-range of flow bias scores g) high-flow bias scores





## 3.2 Spatial Patterns of Performance

The spatial patterns of model performance show that the LSTM improves simulation of discharge across Great Britain see Figure 4. The EA LSTM has very similar spatial patterns to the LSTM, and shows a consistently worse performance than the LSTM across GB.





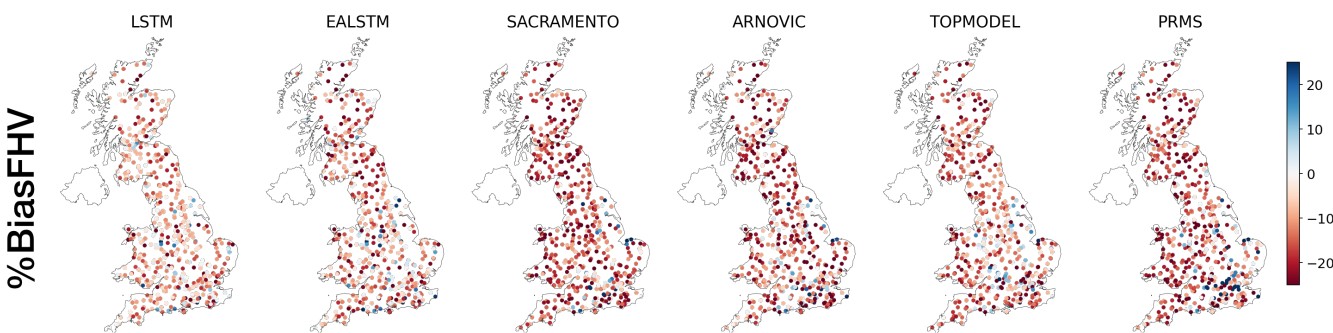

**Figure 4.** Spatial Patterns of different performance metrics. Each point is a single station-gauge, and the point is coloured according to the performance metric. For performance metrics with a diverging score (above and below an optimum, e.g. Bias Error) more intense colours represent worse performance. Red represents an under-prediction, blue an over-prediction. For scores which are increasing (e.g. NSE, Correlation), darker colours reflect improved performance.





Conceptual models struggled mostly when simulating discharge in catchments on the permeable bedrock in the South East of England and the mountainous catchments in the North East of Scotland. Performance metrics in the South East were lower

due to poor simulation of variance and correlation, and in North-Eastern Scotland due to poor simulation of the mean flow conditions (Lane et al., 2019; Clark et al., 2008). This performance deficit is due in part to the low rainfall and chalk aquifer in the South East of England, and to the lack of an additional snow module incorporated into the conceptual models for North East Scotland. Interestingly both LSTM models also simulate flows in the South East England less accurately than elsewhere in GB, although the model error in North East Scotland is largely mitigated.

Performance of the LSTM-based models suffer in a number of catchments in South East England relative to LSTM performance elsewhere in GB. In the South East the LSTM shows an underestimate of the variability and a cluster of high bias scores. The LSTM both overestimated and underestimated mean flows in catchments in the South East region, explaining the relative under-performance in the composite metric (NSE) for the LSTM relative to the rest of GB. Like all models except PRMS, the LSTM underestimates the variability of the flows. An initial hypothesis is that hydrological conditions in the

drier catchments with groundwater transfers remain difficult to model, requiring time-varying parameters and more detailed representation of hydrogeological properties. The LSTM partly addresses these challenges (demonstrated by the performance improvement over the conceptual models), but further research should address how the LSTM might be further improved in these low-flow regimes.

Unlike the conceptual models, the LSTM had no difficulty in reproducing flows in North-Eastern Scotland. This is likely

a result of the LSTM based models accurately simulating catchments in which snow processes are significant. The lack of inclusion of a snow module in the conceptual models used as a benchmark very likely explains at least part of this difference in performance. It is interesting that the LSTM also performs less well in the South East relative to the performance of the LSTM elsewhere in GB. An initial hypothesis is that hydrological conditions in the drier catchments with groundwater transfers remain difficult to model, requiring time-varying parameters and more detailed representation of hydrogeological properties.The

LSTM partly addresses these challenges (demonstrated by the performance improvement over the conceptual models), but further research should address how the LSTM might be further improved in these low-flow regimes.

Spatial patterns in the biases for different sections of the flow duration curve, Figure 4, also show improvement across GB for the LSTM based models compared with the conceptual models. The low flow biases (%BiasFLV) for the LSTM and EA LSTM are smaller than the conceptual models across GB, although the largest biases can be found in South East England and

South-Central Wales. The LSTM and EA LSTM tend to overpredict low flows, the same direction of biases as TOPMODEL. Whereas, SACRAMENTO, PRMS, and ARNOVIC have a negative bias. This means that the LSTM is overpredicting low flows, with a larger bias in the South East. Only the LSTMs show consistent underprediction of the midsection slope of the flow duration curve (%BiasFMS). The slope of the midsection of the flow duration curve reflects a watershed having a "flashy" response (Yilmaz et al., 2008), potentially due to small soil moisture capacity. Therefore, an underprediction of the

midsection reflects an underestimation of the "flashiness" of the catchment. The LSTM %BiasFMS is largest for the South East of England. All models show a consistent underprediction of the high flows (98% exceedance probability). While the





LSTM shows a smaller negative bias than the conceptual models, the spatial pattern is very similar. Overall, the LSTM shows considerable improvement across GB, including these under performing regions, the South East of England, and East Scotland.

The regional performance matrix (Figure 5) shows that while performance varies around GB for the conceptual models, the median regional performance of the LSTM is much more stable, ranging from an NSE of 0.85 (ANG - Anglia) to 0.91 (SWESW - SW England South Wales). The largest difference from GB average is 0.03 NSE for SWESW and ANG. In contrast the conceptual models have much more variable performances across the regions. PRMS for example ranges from 0.65 (ANG) to 0.84 (SWESW), a range of 0.19 NSE. The LSTM is more robust to different regional hydrological patterns, showing smaller variability in performance scores. In contrast, the conceptual models show are clearly more capable in certain hydrological regimes than others.

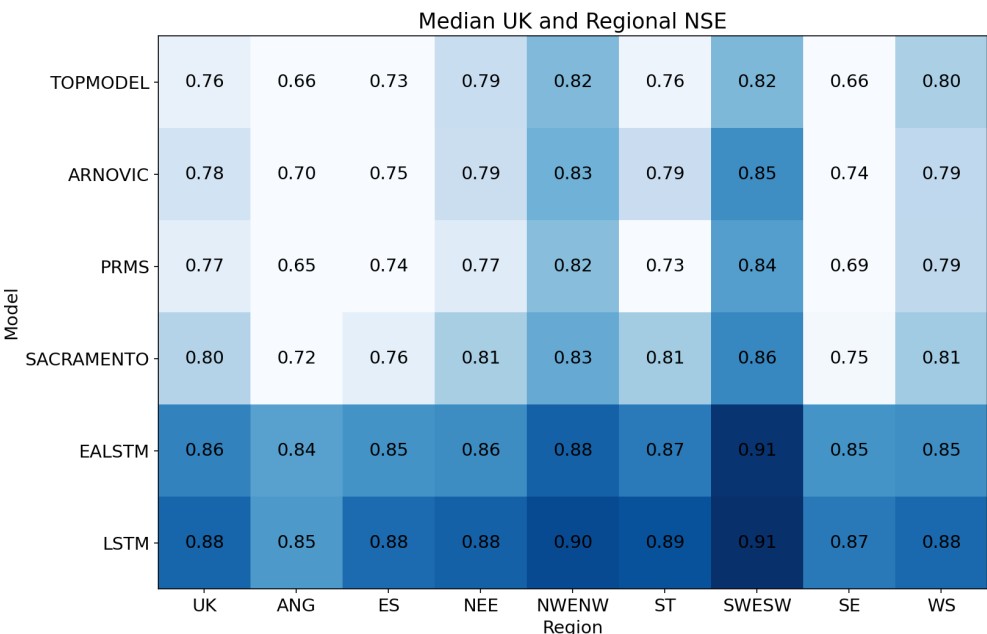

**Figure 5.** Median NSE scores for eight Great Britain river basin regions. The regions are based on the UKCP09 river basins (mur) aggregated from 21 river basin districts to eight regions. The leftmost column is the median score for all GB catchments, which is the same as in Table 3. It is included here for reference.

## 3.3 Seasonal Performance

LSTM based models reproduce similar seasonal patterns to the conceptual models, although the decline in LSTM summer NSE scores are more muted than the decline in conceptual model NSE scores (see Figure 6). Performances for all seasons are worse in the South East of England. This pattern is exacerbated in the summer months (JJA). The East-West gradient in model performances can be seen clearly in the seasonal errors of the conceptual models, whereas this East-West gradient is less pronounced for the LSTM based models, although summer performances are lower in summer (JJA).





**Figure 6.** Seasonal NSE patterns for the two LSTM based models (above) and the conceptual models (below). Each station in the evaluation data is shown as a point. The colour of the point reflects the NSE score. Brighter colors reflect lower NSE values, currently capped at a minimum of 0.7 NSE.





In order to visualise the performance improvement of the LSTM compared with the conceptual models, we calculated the difference in NSE to explore seasonal patterns (ΔNSE). For a catchment where the LSTM NSE is larger than the conceptual model NSE, the value will be positive. A more positive value reflects a larger performance improvement. We compare the four
conceptual models to the LSTM.

The seasonal pattern of ΔNSEs shows that the LSTM-based models most improve simulations of discharge in the summer seasons ("JJA" - the green line in Figure 7). The seasonal pattern is consistent for all of the conceptual models that we benchmark against, although the seasonal difference in ΔNSE is largest for TOPMODEL and ARNOVIC.

## 4  Discussion

This study benchmarks the performance of the LSTM compared to four commonly used conceptual models and two physically-based models. The comparison with the conceptual models was calculated over a large number of catchments, making the study representative of performance in different regions of Great Britain. The performance of the LSTM demonstrates that there is adequate information in the observational data to accurately simulate discharge behaviours in the various hydrological conditions found in Great Britain. The LSTM performance is likely a conservative estimate owing to the limited training period
and the lack of advanced hyperparameter tuning. The simulated time series can be found at: zenodo.org/record/4555820.

In the discussion that follows we will return to our three research questions: (i) How well do LSTM-based models simulate discharge in Great Britain? (ii) Can the LSTM overcome limitations of previously calibrated models, producing accurate simulations in regions where hydrological models typically struggle to reproduce hydrological outputs? (iii) Are there hydrological processes that data-driven models simulate better than traditional hydrological models, which can be diagnosed by
benchmarking model performance?

### 4.1  Inter-Model Performances

The LSTM based models produce accurate simulations of discharge across GB, a temperate region. Two findings from this research confirm and extend the conclusions of previous work. First, the LSTM consistently outperforms the EA LSTM, although the differences in performance are small compared with the difference in performance between the LSTM-based models and the
conceptual models. Secondly, LSTM-based models demonstrate state-of-the-art prediction accuracy for discharge modelling (Kratzert et al., 2019; Nearing et al., 2020).

The EA LSTM is constrained to treat information that does not vary over time (catchment attributes) separately from information that varies over time (hydro-meteorological forcings). However, the constraint clearly penalizes performance, which was also found by Kratzert et al. (2019). The underperformance of the EA LSTM relative to the LSTM suggests that the
value of the input gate ($i$) should be combined with time-varying information ($X_t$) to update the cell memory (Equation 8 compared with Equation 2). The catchment attributes alone are not sufficient to determine what information needs to be passed into the cell memory (Equation 8 compared with Equation 2). In other words, the LSTM learns more about the catchments' hydrological response to rainfall from the hydrographs themselves than from the static catchment attributes. This finding suggesting that the catchment properties,

**Figure 7.** Seasonal ΔNSE patterns for the conceptual models. A positive ΔNSE corresponds to the LSTM making a better prediction. The ΔNSE scores were calculated separately for each station and each season, and the overall distribution of these scores is shown in: blue (winter - DJF), orange (spring - MAM), green (summer - JJA), and red (autumn - SON).





whilst useful, do not fully capture all salient perceptual features of the range of hydrological systems present in GB. Rather,
the hydro-meteorological information is required to accurately parameterise the input gate, deciding which information should
pass to the "memory" of the network (Equations 2, 8). For example, we can imagine a snowy catchment where we also need
the temperature information to decide whether to store snow water in the network memory. The LSTM has this temperature
information fed through the input gate (Equation 2), whereas the EA LSTM does not (Equation 8). Moreover, the LSTM trains
much faster than the EA LSTM. The LSTM will train 30 epochs in 1 hour, compared with 30 epochs in 10 hours for the EA
LSTM. This is due to the CUDA optimised code (for running the models on a GPU) developed by Pytorch (v.1.7.1) for training
the LSTM.

Secondly, we have demonstrated that LSTM is a remarkably accurate model for simulating GB hydrology. The mean differences between the LSTM station NSE and the other models is smallest for the EALSTM ($\Delta$NSE = 0.02). This is unsurprising given the very similar architectures of the two models. The differences are larger for the conceptual models, and range from
TOPMODEL ($\Delta$NSE = 0.15); ARNOVIC ($\Delta$NSE = 0.17); SACRAMENTO ($\Delta$NSE = 0.20), and PRMS ($\Delta$NSE = 0.43).
While the mean performances show large differences, due to the presence of poorly performing stations, the median differences are smaller SACRAMENTO ($\Delta$NSE = 0.07); ARNOVIC ($\Delta$NSE = 0.09); PRMS ($\Delta$NSE = 0.10) and TOPMODEL
($\Delta$NSE = 0.10). Both summaries (median, mean) demonstrate that the LSTM offers a single model architecture that is flexible
enough to perform well in a variety of hydrological conditions found in GB.

For all the analyses in this paper, the LSTM-based models are trained on all basins, with a single set of weights for the
whole of GB, and tested (evaluated) on out-of-sample time periods. Therefore, these LSTM models are regional models that
are able to reproduce behaviours across Great Britain. In contrast, most hydrological models perform best when calibrated on
individual basins. This distinction is important because it reflects the situations in which different models will perform best.
The LSTM-based models are most accurate when trained with as much data from as many catchments as possible (Gauch et al.,
2021). In contrast, traditional hydrological models, including the lumped conceptual models we use as a benchmark, produce
their best simulations when trained on an individual catchment.

In order to further verify that the LSTM based models are producing comparatively accurate simulations, we took published
NSE scores for 13 test basins in the UK for two process-based models (Fatichi et al., 2016), JULES (Best et al., 2011; Clark
et al., 2011) and CLASSIC (Crooks et al., 2014). Both models use the same ancillary data such as discretised soil maps, land
cover and flow direction grids, JULES is driven by atmospheric information and computes land surface fluxes directly, conserving mass, energy and momentum. In contrast, CLASSIC-GB runs using precipitation and pre-calculated surface evaporation.
More detail can be found in the papers of Best et al. (2011) and Crooks et al. (2014) for the hydrological components of these
two models. Martínez-de la Torre et al. (2019) simulated catchments using the JULES land-surface model. Input data came
from 1 km gridded data for the 13 test catchments. The meteorological data comes from CHESS-met (radiation, tempera-
ture, humidity, wind-speed and pressure) and CEH-GEAR (precipitation), the same data sets used in CAMELS-GB. The input
data was 1 km gridded data rather than catchment averaged data. JULES was calibrated and tested on the period 1991-2000.
Crooks et al. (2014) ran CLASSIC-GB for 41 stations, including the 13 test stations used by Martínez-de la Torre et al. (2019).
CLASSIC-GB was run at four spatial resolutions ($100 km^2$, $25 km^2$, $6.25 km^2$ and $1 km^2$). We use the results from their exper-





iment at 1 km. The driving data used comes from the Met Office Rainfall and Evaporation Calculation System (MORECS). The model simulation was produced from 1980-1983. We are aware that these time periods do not match with our test periods, and therefore the following comparison should serve as a preliminary experiment to ascertain whether the LSTM is producing state-of-the-art simulation accuracy in GB. Further research should consider a more complete intercomparison of these models, and consider the robustness of model estimates to uncertain future conditions, as has been explored by Sungmin et al. (2020). Table 4 outlines the results for the 13 test basins for which results are published.

**Table 4.** NSE Scores comparing the FUSE models against the LSTM / EALSTM. Best score (highest) highlighted in bold.

| Station ID | Name | LSTM | EALSTM | CLASSIC | JULES | TOPMODEL | ARNOVIC | PRMS | SACRAMENTO |
|---|---|---|---|---|---|---|---|---|---|
| 12002 | Dee at Park | **0.90** | 0.87 | 0.55 | 0.51 | 0.65 | 0.71 | 0.68 | 0.71 |
| 15006 | Tay at Ballathie | **0.95** | 0.94 | 0.46 | 0.64 | 0.79 | 0.88 | 0.86 | 0.88 |
| 27009 | Ouse at Skelton | **0.94** | 0.91 | 0.80 | 0.69 | 0.86 | 0.89 | 0.88 | 0.91 |
| 27034 | Ure at Kilgram Bridge | **0.88** | 0.87 | 0.78 | 0.75 | 0.84 | 0.84 | 0.85 | 0.85 |
| 27041 | Derwent at Buttercrambe | **0.92** | 0.79 | 0.65 | 0.49 | 0.77 | 0.78 | 0.82 | 0.87 |
| 39001 | Thames at Kingston | **0.95** | 0.94 | 0.81 | 0.82 | 0.75 | 0.78 | 0.69 | 0.88 |
| 39081 | Ock at Abingdon | **0.91** | 0.86 | 0.80 | -0.21 | 0.73 | 0.81 | 0.76 | 0.82 |
| 43021 | Avon at Knapp Mill | **0.91** | **0.91** | 0.60 | -0.07 | 0.54 | **0.91** | 0.84 | 0.89 |
| 47001 | Tamar at Gunnislake | **0.94** | 0.93 | 0.82 | 0.63 | 0.86 | 0.89 | 0.88 | 0.89 |
| 54001 | Severn at Bewdley | **0.95** | 0.91 | 0.66 | 0.61 | 0.88 | 0.88 | 0.87 | 0.92 |
| 54057 | Severn at Haw Bridge | **0.93** | **0.93** | 0.78 | 0.72 | 0.88 | 0.88 | 0.88 | 0.92 |
| 71001 | Ribble at Samlesbury | **0.90** | 0.88 | 0.73 | 0.74 | 0.83 | 0.83 | 0.83 | 0.84 |
| 84013 | Clyde at Daldowie | 0.93 | **0.94** | 0.80 | 0.82 | 0.87 | 0.88 | 0.85 | 0.88 |

In summary, we have demonstrated that the LSTM models perform well relative to the conceptual models, demonstrating consistent improvements across GB. Furthermore, a preliminary comparison with the physical model further adds weight to our suggestion that the LSTM produces state-of-the-art simulation accuracy.

## 4.2 In what hydrological conditions does the LSTM outperform benchmark models?

At the outset we hypothesised that LSTM performance improvement would be largest under conditions that conceptual models most often underperform, thus overcoming the limitations of previously calibrated models. The conceptual models struggled to produce good simulations in two geographical regions. These were in the South East of England and North East of Scotland. The performance improvement ($\Delta$NSE) is indeed largest in the South East of England and North East Scotland (see Figure 8).

North East Scotland is one of the most mountainous regions of GB. The Cairngorm National Park is the only area of GB where snow processes are consistently important, owing to catchments having a higher elevation. The results in Figure 8 show that the LSTM largely overcomes the difficulties in modelling these catchments, since $\Delta$NSE is high. This is most likely due to



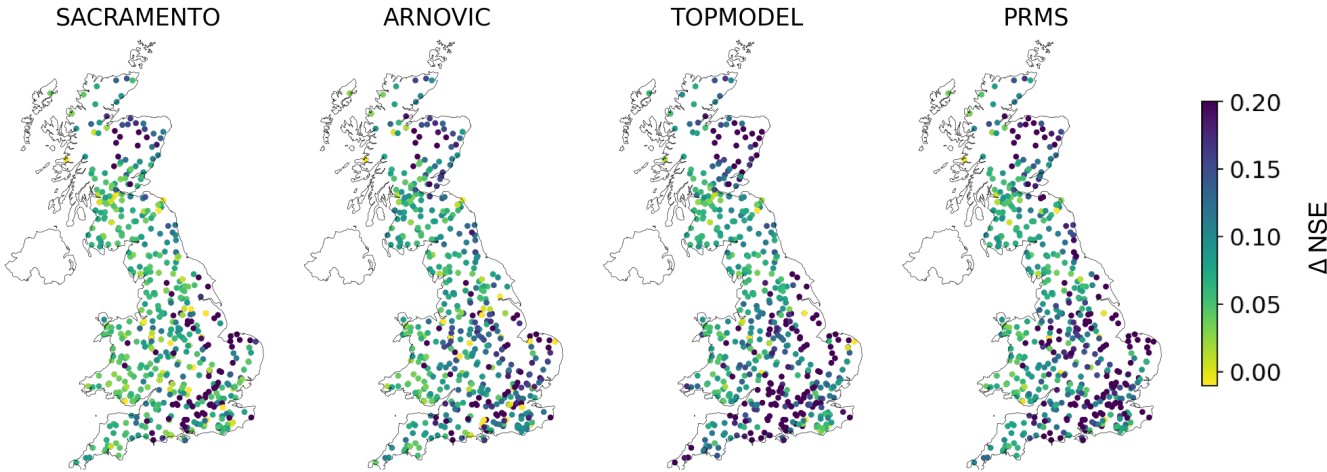

**Figure 8.** The performance improvement of the LSTM relative to the four conceptual models, SACRAMENTO, ARNOVIC, TOPMODEL and PRMS. The difference in NSE is calculated by subtracting the conceptual model NSE from the LSTM NSE ($\Delta NSE = NSE_{LSTM} - NSE_{conceptual}$). Each point represents a station and the colour reflects the performance improvement (measured by NSE) of the LSTM compared with the conceptual models. Positive values reflect stations where the LSTM outperforms the conceptual models.

the cell state (Equation 5) being able to represent longer-term stores and fluxes of water, therefore capturing the melting snow processes.

The South East is a relatively dry area (see Figure 1a), with large chalk aquifers contributing to a high baseflow index (see Figure 1d) and large urban and agricultural areas, contributing to a large anthropogenic signal in the hydrographs. Although the improvement in simulation accuracy compared to the conceptual models is large in the South East, the pattern of raw LSTM NSE shows that the LSTM still underperforms in the South East relative to elsewhere in GB. The seasonal patterns showed that the LSTMs performed worse in summer months, which is the drier period of the year. Consistent with this spatial pattern, aridity is negatively correlated with model performance for all models (Figure 9), although the magnitude of this association is smaller for the LSTM based models than the conceptual models.

We observe consistently poorer performance across all models, conceptual and LSTM, in drier hydrological conditions. We can think of two possible explanations. Either the use of NSE as an objective function fails to adequately weight performance in these low flow regimes (the NSE was the objective function across both the conceptual models and the DL models). An alternative explanation is that hydrological processes are significantly more complex in these drier regimes. For the former, there are other catchment attributes that point to improved modelling of high flows relative to low flows, since all model NSE scores show positive correlations with increased discharge (at mean flow, Q5 and Q95), as well as increased NSE as rainfall increases ($p\_mean$). A future study will consider the impact of different objective functions. For increased complexity of arid conditions (and therefore increased difficulty to model and lower performance scores) the lower catchment "connectivity" in these arid conditions could provide an explanation (Bracken and Croke, 2007). In winter, when soils are saturated, there are a



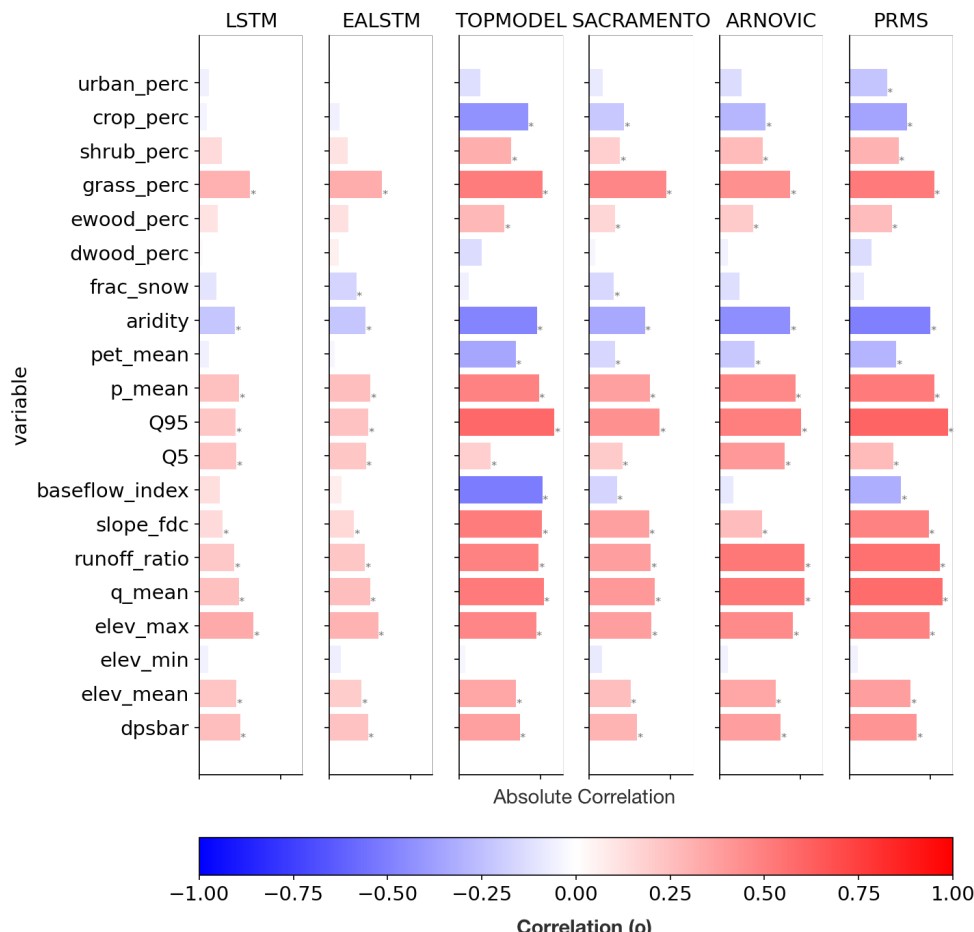

**Figure 9.** Static features (rows) and their Spearman's Rank Correlation Coefficient with model (columns) NSE scores. The positive correlations are in blue, the negative correlations are in red. Pale bars show very low correlations. (*) indicates that the correlation is significant at the $\alpha$=0.001 level. The first 6 features can be classified as landcover features. The next 4 features are climatic indices. The next 6 features are hydrologic attributes and the final 4 are topographic features. DPSBar refers to the mean drainage path slope, and reflects the average steepness of a catchment.

greater number of pathways for water to enter river channels. Therefore, we say that connectivity is high in winter. Whereas,
in summer there is greater resistance to water flow, since water can be absorbed and stored in drier soils, as found in Swiss catchments by van Meerveld et al. (2019). Therefore, in summer connectivity is lower. The proposed impact of catchment connectivity on the performance improvement of the LSTM based models is ultimately speculative, and future work will explore whether the LSTM has learned to represent the concept of connectivity.

Like aridity, the relative cover of cropland (crop_perc) shows a strong negative correlation with conceptual model performance. In contrast, there is little or no correlation for the LSTM-based models. This suggests that the LSTM has used the





information from the cropland cover variable in order to improve the representation of hydrology in those catchments with a strong agricultural signal. Further research is planned to attribute what the internal states of the LSTM suggest about the differences between agricultural catchments and catchments with a much smaller proportion of cropland cover.

Overall, we have found that the LSTM performance improvement is largest in the South East of England and North East of Scotland, where the conceptual models struggle. This is partly a result of the LSTM improving performance across GB, showing greater robustness to a variety of catchment conditions. However, modelling the dry South East remains difficult even for the LSTM, and aridity remains a catchment characteristic that is associated with poorer model performance. Our findings show that whilst the LSTM offers improved performance in conditions where lumped conceptual models have previously struggled, there remains potential for further improvement in these drier settings, such as in the South East of England.

**4.3    The impact of water balance closure on simulation accuracy**

One of the key conditions that conceptual models struggle with is when the catchment water balance does not close. The conceptual models we test here explicitly maintain mass balance. They define the topographic surface water catchment to be the surface over which water is conserved, i.e. the surface water catchment does not leak, nor that any water enters other than through measured precipitation, for example through undercatch, drifting snow, or advection of fog, groundwater, or
anthropogenic transfers into or out of the topographic catchment. Therefore, the conceptual models struggle to produce accurate simulations in catchments where the water balance (defined in the data) does not close. The LSTM, in contrast, is free to diagnose inter-catchment transfers (either through anthropogenic or groundwater processes). This was the rationale behind our final research question that benchmarking the LSTM can help researchers diagnose processes that traditional models could simulate data better, given that information exists in the data to describe them.

We plot catchments on two dimensions (Figure 10), their wetness index (P/PE) and the runoff coefficient (Q/P), to identify catchments where water transfers outside of the topographic surface water catchment may be occurring. Points above the horizontal line reflect catchments where the observed discharge is greater than the precipitation input to the catchment. This area of the graph represents catchments where the data has too little water to generate the observed runoff. Points below the curved line are where runoff deficits exceed total PET in a catchment. This area of the graph represents catchments where PET
is not large enough to describe the water remaining after runoff is accounted for, i.e. the data has "excess" water (Figure 10).

     Interestingly, both the LSTMs and the conceptual models produce a performance decline in catchments with an imbalanced water balance. This suggests either that the LSTM models still struggle with water-limited and energy limited (low runoff coefficient and low wetness index) catchments. Alternatively, the fact that both LSTMs and conceptual models struggle in catchments where data does not meet the water balance constraints might suggest that human impacts on the hydrograph are
ultimately unpredictable, such as abstraction and effluent returns. However, the performance decline is much less pronounced than the conceptual models and the LSTM continues to produce simulations with NSE scores greater than 0.6. This suggests there remains more information in the data that the conceptual models are currently unable to utilise.



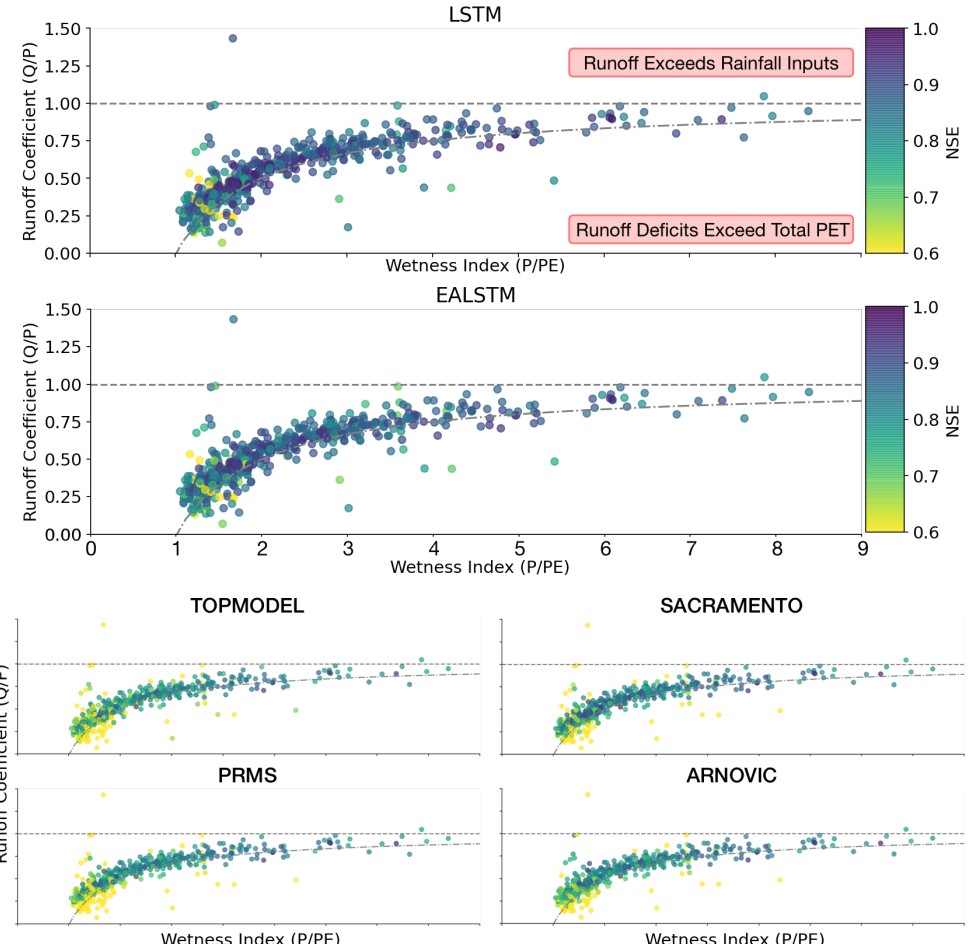

**Figure 10.** Scatter plot for the relationship between the wetness index, runoff coefficient and the model NSE score. Each point is a catchment, coloured by the NSE score ranging from 0.8 (lighter) to 1.0 (darker). Points above the horizontal line reflect catchments where the observed discharge is greater than the precipitation input to the catchment. Points below the curved line are where runoff deficits exceed total PET in a catchment, therefore, there is "excess water" in the data, since PET cannot explain the leftover water after accounting for runoff.

We tested whether the LSTM was better able to simulate discharge in catchments with "excess" water (i.e. the points below the curved line on the plots in Figure 11). The LSTM is much more robust to these conditions and produces NSE scores that
are comparable to the stations where the conceptual models perform best.

Overall, the results demonstrate that the LSTMs are better able to model conditions where the topographic surface water catchment has excess water. The performances are most improved for the conditions that are furthest from the water balance constraint being met. There are two key findings here. First, catchment transfers may be detectable from the data alone, assuming that the processes are at least in part due to real signals rather than data errors. Second, like conceptual models,
LSTM performance declines in these catchments where runoff deficits exceed total PET. This could either be because the NSE



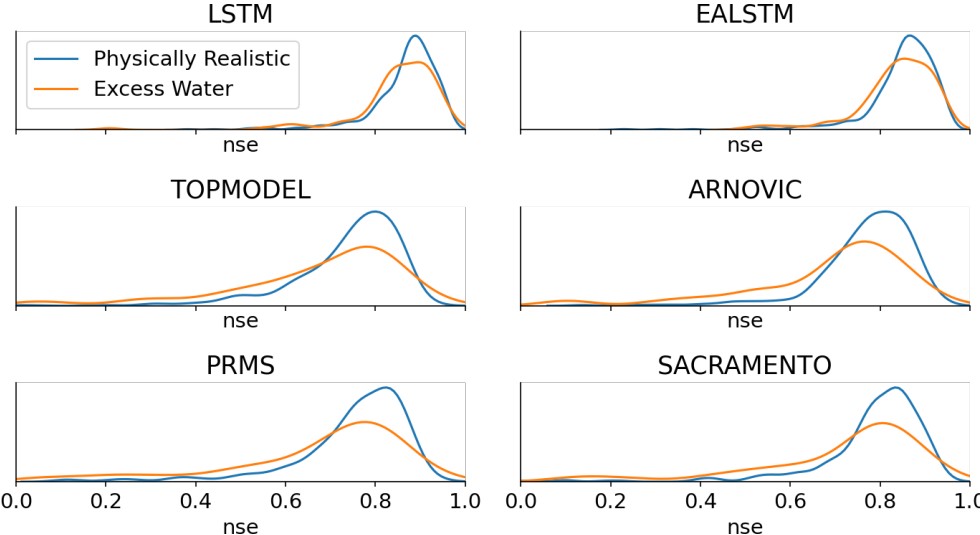

**Figure 11.** Comparing the NSE results for catchments that have excess water, where runoff deficits exceed total PET (orange) to those catchments that have physically realistic conditions (blue). The orange line shows the histograms for stations that fall below the curved line in the Budyko analysis above (the runoff deficit exceeds total PET, therefore there is excess water in the model). The blue line shows the histograms for those stations between the two dashed lines.

is an inappropriate objective function for these catchments, or else that hydrological processes in these catchments are not able to be modelled with the available data.

## 5   Conclusions

In this study we have compared two LSTM based models to four conceptual models over 518 catchments across Great Britain.
We have demonstrated that LSTM-based models trained on a large sample of catchment-averaged hydro-meteorological time-series consistently outperform a suite of conceptual models calibrated on individual basins (Lane et al., 2019). The LSTM models are capable of producing simulations with median NSE scores of 0.88 (LSTM) and 0.86 (EA LSTM). The results of the experiments described above show that LSTM-based models are capable of producing accurate simulations across a variety of hydrological conditions in Great Britain, consistent with the findings from Kratzert et al. (2019) in a different geographical
context. Based on our results the LSTMs do produce more accurate simulations across GB. We have shown that the EA LSTM performance suffers compared to the LSTM. The separate treatment of variables that are treated as static over time (catchment characteristics) restricts the model, by assuming the constant importance of those variables over time. However, the performance of both models is similar.

The spatial patterns of performance demonstrate that while the LSTMs improve simulations most in South East England
and North East Scotland, they continue to underperform in South East England relative to elsewhere in GB. When we consider the catchment conditions that are associated with this pattern it is clear that all models struggle with drier conditions and





catchments where the water balance does not close. This demonstrates the importance of accounting for water balance losses and gains in hydrological modelling across GB and more research should be directed at quantifying water transfers through groundwater and human management.

We identified a number of hydrological characteristics that correlate with model performance. These correlations are similar in direction for the conceptual models and the LSTMs. However, they differ in magnitude, and rank correlation scores are lower for all of the variables we tested. Overall, we find that the LSTMs are more robust to the diversity of observed hydrological conditions found across GB.

    Future research will consider the internal states of the LSTM to identify how the LSTM learns to reproduce hydrological

signatures. The LSTM is structured such that its internal states reflect subsurface stores and fluxes of water, such as soil moisture.We expect that the internal states will allow us to explore how the LSTM has learned to diagnose between catchments with different degrees of external influence on the hydrograph. Finally, we will consider the impact of different objective functions on improving simulations of different parts of the flow duration curve.

    This work demonstrates the ability of LSTM-based models to accurately simulate the hydrological response to rainfall in

Great Britain. We have used a new data set, CAMELS-GB, and demonstrated that there is sufficient information in this data to create state-of-the-art data-driven models based on an LSTM. The results provide a credible baseline for comparison of future models, and we make model predictions from trained LSTMs and error metrics for each catchment available here: zenodo.org/record/4555820.

*Code and data availability.* CAMELS-GB data is available at: https://catalogue.ceh.ac.uk/documents/8344e4f3-d2ea-44f5-8afa-86d2987543a9.

The FUSE benchmark model simulations are available at: https://data.bris.ac.uk/data/dataset/3ma509dlakcf720aw8x82aq4tm. The neuralhydrology package is available on github here: https://github.com/neuralhydrology/neuralhydrology

## Appendix A: Comparison of the Train and Test Periods

The calibration (train) period and the evaluation (test) period are similar in terms of their predictability, although the evaluation period was slightly less predictable, seen by the distributions of two baseline models being shifted towards lower NSE values

(see Figure A1). We used two baseline models to test how "predictable" the catchment hydrographs are in these two time periods. Climatology makes a prediction based on the mean discharge for that day of the year. Persistence is equivalent to predicting yesterday's value today, predicting the future will be the same as the past. Figure A1 shows that the processes are largely stationary, and the period we use for calibration is similar to the period we use for evaluation. Indeed, the period we use for calibration is slightly easier to predict than the test period, since the benchmark models perform better, i.e. the

distribution of catchment NSE scores is shifted towards higher NSE scores during the train period. Furthermore, the conditions for precipitation, PET, temperature and specific discharge are very similar between the train and test period. The temperatures have warmed slightly and there are slightly more days with zero precipitation, however, it is unlikely that such small changes





have impacted the ability of the DL model to generalize. Discharge has risen slightly in the period of interest, across Great Britain.



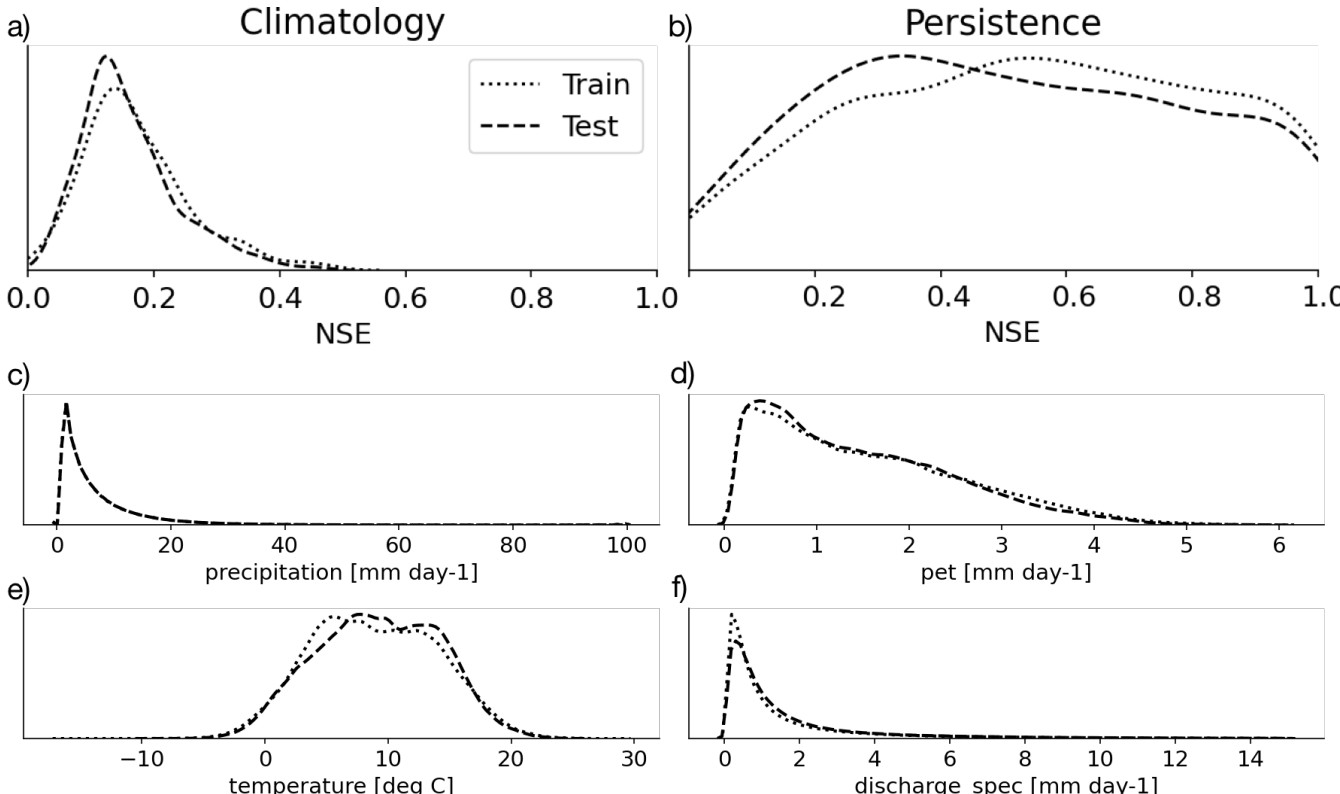

**Figure A1.** Kernel Density Estimates (KDE) of NSE scores for two baseline models (above), Climatology (a) and Persistence (b). Below, there are Kernel Density Estimates for hydro-meteorological variables, precipitation (c), potential evaporation (pet) (d), temperature (e) and specific discharge (f) in the training period (1980–1997, dotted line) and the test period (1998–2008, dashed line). Climatology represents the mean conditions for that day of the year. Persistence reflects predicting yesterday's values today, i.e. predicting no change from yesterday. These give an overview of how "predictable" a time period is, since if these baseline models perform well, it will be easier to score at least as well as the baseline.

**Appendix B: Model Hydrographs**

We illustrate the model predictions by showing the hydrographs for three stations from the Thames, the Severn and the Tay, as the biggest rivers having at least part of their catchment in England, Wales and Scotland respectively.

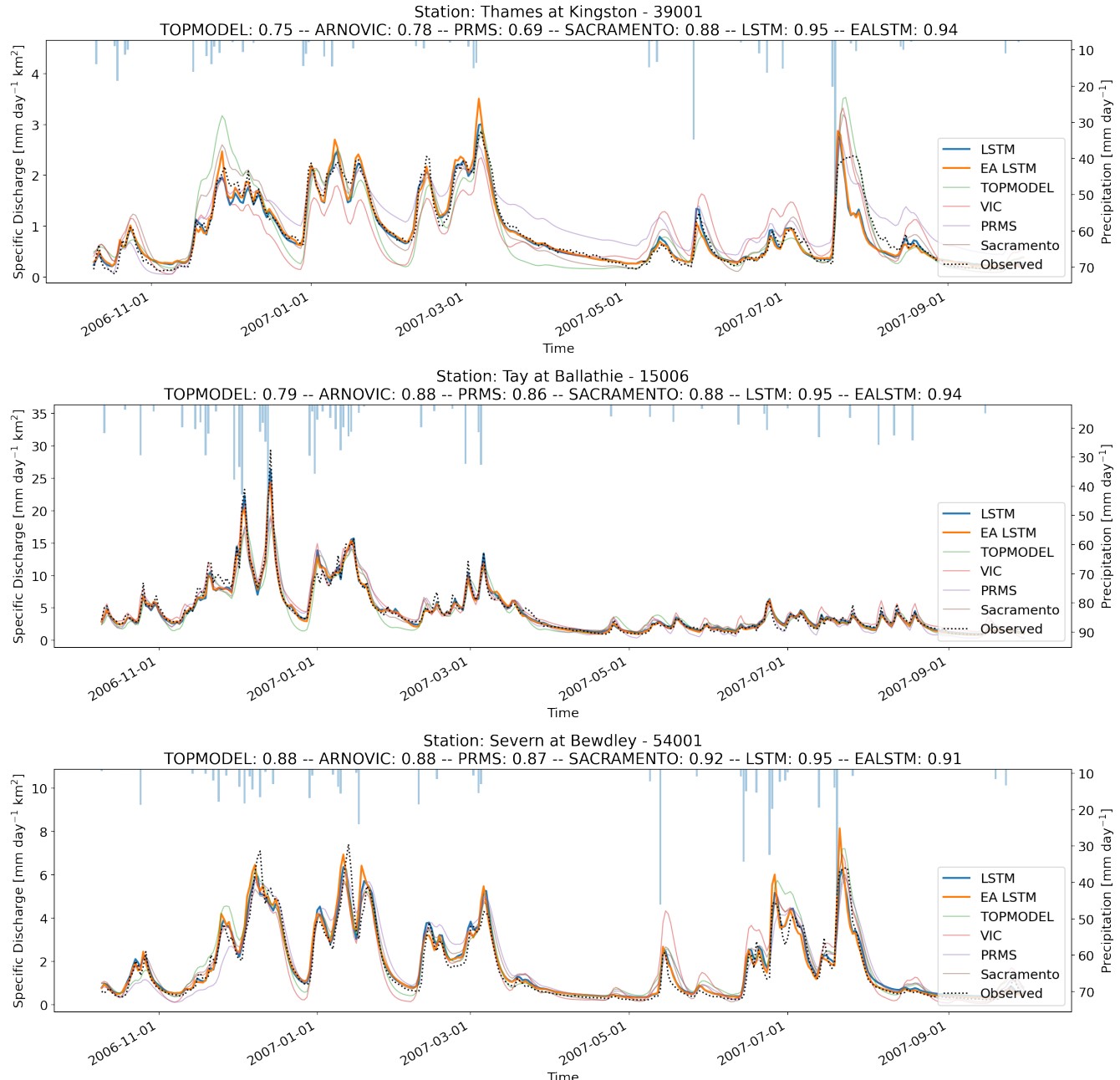

**Figure B1.** Hydrographs for the Thames at Kingston (Station 39001), the Tay at Ballathie (Station 15006) and the Severn at Bewdley (Station 54001), for the hydrological year from October 2006 – September 2007. The model performances displayed in the header reflect the performance of each model on the entire test period (1998–2008), not just the displayed period. The observed discharge, from (Coxon et al., 2020a), is shown as a dotted black line. The bars reflect catchment averaged precipitation with the axis shown on the right side. The LSTM and EA LSTM simulations are shown in blue and orange respectively. Conceptual model simulations for Sacramento (brown), VIC (red), PRMS (purple) and TOPMODEL (green) are taken from published timeseries from (Lane et al., 2019).



*Author contributions.* TL designed and conducted all experiments and analysed results with advice from SD, LS, and SR. BA compiled the comparison results from JULES and CLASSIC-GB; MB performed preliminary geospatial analysis; GC guided the water balance analysis.
All authors discussed and assisted with interpretation of the results and contributed to the manuscript.

*Competing interests.* The authors declare that they have no conflict of interest.

*Acknowledgements.* The authors would like to thank the teams responsible for releasing CAMELS GB (Coxon et al., 2020b), the FUSE benchmarking study (Lane et al., 2019) and the authors and maintainers of the neuralhydrology codebase for training machine learning models for rainfall-runoff modelling. TL is supported by the NPIF award NE/L002612/1; MB is supported by NERC DTP studentship
NE/L002612/1, and BA is supported by the Clarendon Scholarship. SD is supported by NERC grant NE/S017380/1.





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
