# Peer review of "Benchmarking Data-Driven Rainfall-Runoff Models in Great Britain: A comparison of LSTM-based models with four lumped conceptual models"

_Hydrology and Earth System Sciences, 2021_

## Referee Comment (RC2)

**Review of "Benchmarking Data-Driven Rainfall-Runoff Models in Great Britain: A comparison of LSTM-based models with four lumped conceptual models" by Lees et al.**

This manuscript investigates the following research questions:
- Are (regional) LSTM-based models able to simulate the rainfall-runoff process in GB and how do they compare against different, well established hydrological models?
- Can we extract any insights from this comparison, e.g. are there certain types of catchments that are consistently modeled better by one class of models (here LSTMs), which may hint to a missing representation for a dominant hydrological process in the other model class (here the benchmark models)?

Overall, I think this is a very good manuscript that only needs minor modifications. No additional experiments are required. The list of my suggestions might seem long in the first place, however most things should be very easy to fix. I tried to list everything I found, because I hope that will make the manuscript better. However, I am happily open to discuss any of my points.

Before I start with my comments, I want to highlight the points of the manuscript that I liked:
- The study is conducted on a large-sample, public dataset that was never used before for this kind of studies.
- All code is published and it seems trivial to reproduce the results of this manuscript.
- The benchmarking includes model outputs from different research groups, making it less likely that the model comparison is biased.
- The evaluation is performed on multiple metrics that account for different parts of the hydrograph.
- The discussion and analysis of the results w.r.t. the hydrological context/region was insightful.

Next, a few general points:

**Format**
- The mathematical notation is inconsistent and not in line with the HESS guidelines. E.g.
  - Vectors (e.g. all gates, the cell and hidden state, and the inputs at a particular timestep) should be boldface italics lower case.
  - Matrices (e.g. weights) should be printed in upper boldface roman (upright) font.
- Abbreviations are no in line with the HESS guidelines. E.g. "Figure" and "Equation" (as the entire word) should only be used at the beginning of a sentence. Mid-sentence "Fig." and "Eq." should be used.
- Dates are not in line with the HESS guidelines. The format of the dates should be dd mm yyyy (e.g. 31 December 2008).

**Paper length**
- The manuscript is quite long but there is potential for shortening certain parts. I think a shorter, more concise paper will ultimately make this manuscript more read by people. I do have a few suggestions, where the manuscript could be shortened but feel free to ignore all of them if you would like these sections as is:
  - The model description of the LSTM and EA-LSTM is quite long and little information is added in comparison to the original manuscript that is cited. Personally, I don't think that the equations and the entire formal explanation is needed and in my opinion it could be removed. I think it would suffice to have a short, one paragraph explanation of the main difference between those two models (maybe with Fig. 2) and then link the reader to the original citation. We have seen quite a few LSTM publications recently, and similar to papers with traditional hydrological models, in my opinion it is not needed again and again to write down the model equations.
  - Very similar in my opinion is the section about the evaluation protocol. Personally, I don't think we need to see the equation of e.g. the Nash-Sutcliffe-Efficiency in every hydrology paper. Also the other equations could maybe be removed and you only keep a short explanation of the metrics with a reference to the original manuscripts. If you want to keep all equations, why not list them in e.g. a compact table like in Best et al. (2015).
  - The manuscript contains a lot of figures. Generally I like that. However, due to the length of the manuscript, there are some figures that I think could be shortened or removed (or moved to the supplement). E.g. Figure 4 spans 3 pages, however most of these maps show the same pattern.
  - Lastly, I was a bit confused about the section starting in L 452ff. At this point, we reached the discussion of the results. You evaluated and compared the LSTM-based models in hundreds of basins with established hydrological models. Why do you add another comparison in the discussion with two additional models in (only) 13 basins? I see no real motivation for this additional comparison and there are no new insights gained from this comparison. Personally, I think this section can be removed entirely, or I would like to see a better explanation why

this additional comparison is wanted/needed and what we get out of it that we did not know from the first, very large-sample, comparison.

**Minor line-by-line comments**:
- L 38: "*account for temporal dependence using a series of recurrent layers*": RNNs account for temporal dependencies by processing the input time series timestep by timestep. This can (as in your LSTM model) also be done by a single layer. Using a "series of recurrent layers" would mean to stack RNN layers, each with it's own set of weights.
- L 42: "*Long Short Term Memory*" -> "Long Short-Term Memory"
- L 50: The cited reference for the EA-LSTM did not investigate prediction in ungauged basins. It was done in a different publication, by the same authors though, in which they did not use the EA-LSTM however.
- L 77: "*Our study poses the following four research questions:*" Your enumeration only contains three research questions.
- L 109: Table 2 mentioned before Tab 1.
- L 114: "*The static attributes we use to train the LSTM models are listed in Table 1.*" Table 1 does not list the static attributes but the notation of the mathematical symbols.

(Next points are only important if you decide to keep Section 2.3)
- In Eq. 1-4: To simplify, you can write $[[X\_t, A], h\_t\text{-}1]$ as $[X\_t, A, h\_t\text{-}1]$, since all three vectors are stacked. Although note, as stated above, that it should be boldface italics lowercase for all three vectors
- L 159 "hs" is explained in L. 219 but not here. Maybe better to explain "hs" at the first occurrence and remove the explanation in L 219.
- Figure 2: caption "we have 365 cells" maybe confusing with cells also regularly used to describe the number of memory cells. Maybe simply remove the last part of the sentence.
- L 185 y_hat is not explained.
- L 185, Eq. 12 $M\_theta$ (which is a function) should be typeset in roman (upright) font, see HESS guidelines for mathematical symbols and functions.
- L 200 The following two sentences could be rephrased or maybe one could be removed: "*We used 21 static inputs (A). Each catchment was characterised using 21 individual features describing the topographic, soil, land-cover, and climatic properties.*". Maybe just write "*Each catchment was characterised using 21 individual features (A) describing the topographic, soil, land-cover, and climatic properties.*".
- Table 2, The median for low_prec_freq is missing.
- L 207-208 Slightly repetitive to the preceding paragraphs. Can maybe be deleted!?
- L 212 Just to be sure, are you using the average discharge of the ensemble or are you later reporting the average metric value, calculated as the mean/median over the ensemble members? I think it is the former, but it would maybe help to be more explicit here.

(End of Section 2.3 comments)

- L 247: "...*chose parameters for the 4 lumped models from a grid of 10,000 parameters...*"
  I think the formulation is slightly wrong. It is not a "grid of 10.000" parameters", but they
  sampel 10.000 parameter sets from a grid defined by the user defined parameter
  boundaries. Maybe "...chose parameters for the 4 lumped models by sampling 10,000
  random parameter sets from a grid with predefined parameter boundaries..."?
- L 245-254 What you describe here, especially the difference of how the model
  parameters are selected, is indeed a huge difference. The section however, is quite long
  and you could maybe try to rewrite this section in a more concise/structured way. If I
  understand you correctly, there are two main points you want to say:
    - Lane et al. "calibrate" their models by sampling 10.000 different parameter sets
      and evaluating the models on the entire data record, then picking the parameter
      set with the highest NSE. In contrast, you train your model on one data split, and
      you use a different data split (of unseen data) to evaluate the models and to
      calculate the metrics.
    - Lane et al. find individual parameter sets per basin. In contrast, you train one
      model with a single set of parameters for all basins at once.
  Maybe you find a way to shorten this section and boil it down to the main points.
- L 256f "*An important difference between the LSTMs and the traditional hydrological
  models, is that traditional models perform best when calibrated for individual basins. The
  parameters that they use to produce simulations are unique to each basin. This often
  represents the state-of-the-art for traditional hydrological models.*" I think the last
  sentence can be removed, as you already said that in the first sentence. Although I
  agree, it might be good to have a reference for such a statement.
- L 256 I feel like such a sentence needs either a reference or an experiment.
- L 261ff "*The conceptual models were calibrated and evaluated to produce simulated
  streamflows by Lane et al. (2019). We did not run these benchmarks ourselves. This is
  important because we have not biased the calibration of these models to favour the
  deep learning models. We have used the published time-series of model outputs to
  calculate performance scores for the conceptual Models.*" You can maybe remove this
  sentences, since you already stated the same at the beginning of Sect. 2.5.1. The only
  thing added here is the sentence of being unbiased. You could maybe add this to the
  first sentences of this paragraph as well. E.g. Maybe (L. 229) "We compare the
  performance of the LSTM based models against a range of lumped, conceptual models.
  To be unbiased on the model calibration, we used predicted discharge time series from
  Lane et al. (2019) who utilised the FUSE framework to train and evaluate four lumped
  conceptual models across Great Britain (Clark et al., 2008)."
- L 263 Why does using simulations from someone else help you to better understand the
  seasonal and geographical patterns?
- Table 3: Are the LSTM metrics the mean/median over the 10 repetition, or the metric
  value given the ensemble mean discharge? If the former, you could/should report the
  std/interquartile range as an error metric.
- Table 3 and L 313: Since all models model the same basins, you should use the "paired"
  Wilcoxon test. Furthermore, I am a bit surprised by the results of %BiasFMS: Did you
  test for significance using the absolute metric values of he signed metric values?

Because the LSTM is actually closer to zero as TOPMODEL, making it actually the better model. Since, in my opinion, neither over- nor underestimating is better, you should test for significance using the absolute metric values per basin for the metrics that go from -inf to +inf (with zero being the best).

- L 332 The difference in low flow metrics is indeed interesting. I am not too familiar with the CAMELS GB data but I could imagine that there is one of the reasons might also be the difference between the two datasets (CAMELS US and GB). In CAMELS US, there are a number of basins that fall completely dry during long periods of the year, which are generally hard to model. What is "dry" in CAMELS GB, might not be in the spectrum that was reported as difficult for CAMELS US. So maybe the LSTM is not good at zero flow predictions but good for "lower flows" (with water)?!
- Figure 4: As stated above, these are a lot of plots/pages. Maybe not all are necessary so the paper becomes shorter? E.g. for most metrics, the patterns are pretty much identical. There is only a visible difference in the pattern for %BiasFLV, where TOPMODEL is different to the other benchmark models. Maybe just include figures for one (or two?) metric(s) in the main paper and put all others into the supplementary?
- L 360f "*An initial hypothesis is that hydrological conditions in the drier catchments with groundwater transfers remain difficult to model, requiring time-varying parameters and more detailed representation of hydrogeological properties.*" Or maybe different/better inputs? Something like groundwater transfer might be hard to learn from the limited inputs that are used in this study.
- L 362 "*...but further research should address how the LSTM might be further improved in these low-flow regimes.*" Here, you are saying that LSTM performance suffers in low-flow regimes, which would be inline with the studies you referenced above (see L332f).
- L 376f "*This means that the LSTM is overpredicting low flows, with a larger bias in the South East.*" Slightly repetitive to L 375, consider rephrasing.
- L 386 "*The largest difference from GB average is 0.03 NSE*". Isn't the difference 0.05?
- L 389 "*...the conceptual models show are clearly more capable in...*" -> "the conceptual models are clearly more capable in..."
- L 395 Delete "clearly".
- L 394ff I'm not sure if I agree with your summary. For me, it is almost easier to see the East-West gradient in the LSTM/EA-LSTM figures, because they switch from darker colors (all but JJA) to lighter colors (JJA), whereas the others have East-West gradient almost always (all but JJA), where as in JJA almost the entire map has lighter colors.
- L 397ff. Figure 7 and Fig. 8 are not linked in the text and the order of the figures should probably be switched to account for the occurrence in the text (map before cdf). Also: I'm not sure if Fig. 7 and Fig. 8 are needed or if their results can be described with 2-3 sentences. Since the pattern and the results are basically always the same, i.e. LSTM is generally better everywhere and at any time. So this could be a good opportunity to shorten the paper.
- L 426f: "*The catchment attributes alone are not sufficient to determine what information needs to be passed into the cell memory (Equation 8 compared with Equation 2). In other words, the LSTM learns more about the catchments' hydrological response to rainfall from the hydrographs themselves than from the static catchment attributes.*" I

agree with the finding that EA-LSTM seems to be worse than LSTM, however I am not sure if I agree with the statement of these sentences. The EA-LSTM has the same discharge available to "learn from" and the LSTM has the same static attributes. You mention in other places that probably the main reason for the difference is that the EA-LSTM "freezes" one of the gates, making the EA-LSTM less flexible, which I think is the main reason for the difference.

- L 428 "*For example, we can imagine a snowy catchment where we also need the temperature information to decide whether to store snow water in the network memory. The LSTM has this temperature information fed through the input gate (Equation 2), whereas the EA LSTM does not (Equation 8).*" The EA-LSTM is still able to model snow, as you described in this example. The only difference is that for the EA-LSTM, the cell-update gate has to model the entire process (i.e. "that there is snow" and "how much" snow is added to the cell). In the standard LSTM, the two things can be modeled by two gates (input gate + cell update gate), making it more flexible to learn this process.

- L 436 I think you mean the correct thing but just to clarify. As far as I understand, both models run on GPUs, the difference is that the standard LSTM makes use of a CUDA optimized implementation in the background, while the EA-LSTM is custom code.

- L 438 At first, I was confused if the deltas are the differences of the means or medians, which is explained then in the next sentence. Maybe you could move this explanation to the beginning?

- L445ff Coming back to an earlier point of my review: I feel like it is worth repeating that you compare against the calibration period of the benchmark models. Most likely, a fair comparison, where you compare to out-of-sample periods of the benchmark models, would further increase the performance difference.

- L 478ff: I'm not familiar with all 4 conceptual models but if I'm not wrong, at least not all of them contain a snow-module in the setting used for this study. Maybe this does also explain some of the differences in North East Scotland?

- L 490ff: Isn't another possible option based on the way how those models are trained? Imagine a basin that has constant low flow (or zero flow) for an extended period each year. All those timesteps yield little information that can be used to update the weights, since for all different meteorological inputs, the output would always have to be the same. So the underlying physical processes can only be inferred from those timesteps with varying discharge.

- L516ff I am not an expert with these models, but how strict is mass conservation really? We can't see more water than what has fallen as precipitation (upper bound) but there is no lower bound, or? Since the models are not calibrated on evapotranspiration, it can vary this model output at will, to e.g. remove less water from the system than it would evaporate in reality. Additionally, some conceptual hydrology models (e.g. SACRAMENTO) have an additional option to remove water from the system that then does not reach the channel, which is the baseloss flow. This is another degree of freedom, which can be fitted at will, since the models are only calibrated on discharge. What I want to say is: Conceptual models can't "invent" water (e.g. by water transfer from a different catchment) but water can be removed at will. So personally, I don't think that a "*leaking catchment*" (L. 518) has to be a problem, or?

- L 533ff "*Alternatively, the fact that both LSTMs and conceptual models struggle in catchments where data does not meet the water balance constraints might suggest that human impacts on the hydrograph are ultimately unpredictable, such as abstraction and effluent returns.*" Or maybe just unpredictable from the given model inputs? Even if anthropogenic influences are included in the catchment attributes, water extraction is most likely a dynamic process and would require additional dynamic inputs. But conceptually, I don't see why a data-driven model should not be able to learn this process? The process is either driven by physical processes or by a human factor, which is most likely driven by a management plan. Both things could in theory be learned, given enough (informative) inputs.
- L 539 Do you mean to link Figure 10? The link to Figure 11 is not clear to me.
- Figure 11: x-axis label (NSE) should be in capital letters
- L 551 Again, since this is the conclusion, worth that your comparison is biased (towards the lumped hydrology models), since you compare your hold-out period to their calibration period.

References:

Best, M. J., Abramowitz, G., Johnson, H. R., Pitman, A. J., Balsamo, G., Boone, A., ... & Vuichard, N. (2015). The plumbing of land surface models: benchmarking model performance. *Journal of Hydrometeorology*, *16*(3), 1425-1442.

---

## Community Comment (CC2)

**Table 4.** NSE Scores comparing the FUSE models against the LSTM / EALSTM. Best score (highest) highlighted in bold.

| Station ID | Name | LSTM | EALSTM | CLASSIC | JULES | TOPMODEL | ARNOVIC | PRMS | SACRAMENTO |
|---|---|---|---|---|---|---|---|---|---|
| 12002 | Dee at Park | **0.90** | 0.87 | 0.64 | 0.51 | 0.65 | 0.71 | 0.68 | 0.71 |
| 15006 | Tay at Ballathie | **0.95** | 0.94 | 0.50 | 0.64 | 0.79 | 0.88 | 0.86 | 0.88 |
| 27009 | Ouse at Skelton | **0.94** | 0.91 | 0.78 | 0.69 | 0.86 | 0.89 | 0.88 | 0.91 |
| 27034 | Ure at Kilgram Bridge | **0.88** | 0.87 | 0.78 | 0.75 | 0.84 | 0.84 | 0.85 | 0.85 |
| 27041 | Derwent at Buttercrambe | **0.92** | 0.79 | 0.74 | 0.49 | 0.77 | 0.78 | 0.82 | 0.87 |
| 39001 | Thames at Kingston | **0.95** | 0.94 | 0.84 | 0.82 | 0.75 | 0.78 | 0.69 | 0.88 |
| 39081 | Ock at Abingdon | **0.91** | 0.86 | 0.80 | -0.21 | 0.73 | 0.81 | 0.76 | 0.82 |
| 43021 | Avon at Knapp Mill | **0.91** | **0.91** | 0.68 | -0.07 | 0.54 | **0.91** | 0.84 | 0.89 |
| 47001 | Tamar at Gunnislake | **0.94** | 0.93 | 0.83 | 0.63 | 0.86 | 0.89 | 0.88 | 0.89 |
| 54001 | Severn at Bewdley | **0.95** | 0.91 | 0.77 | 0.61 | 0.88 | 0.88 | 0.87 | 0.92 |
| 54057 | Severn at Haw Bridge | **0.93** | **0.93** | 0.81 | 0.72 | 0.88 | 0.88 | 0.88 | 0.92 |
| 71001 | Ribble at Samlesbury | **0.90** | 0.88 | 0.79 | 0.74 | 0.83 | 0.83 | 0.83 | 0.84 |
| 84013 | Clyde at Daldowie | 0.93 | **0.94** | 0.82 | 0.82 | 0.87 | 0.88 | 0.85 | 0.88 |

---

## Author Comment (AC1)

**Response to Reviewers for Manuscript: Benchmarking Data-Driven Rainfall-Runoff Models in Great Britain**

**Anonymous Reviewer #1:**

Comments/Text of Anonymous Referee posted in **black**, our text in **blue**.

The manuscript outlines an application of LSTM-based runoff models, which were introduced in previous studies (Kratzert et al, 2018; 2019). In the present contribution, the focus of analysis is catchments in Great Britain. Similar to previous studies, the objective is to demonstrate the competitive ability of LSTM in rainfall-runoff simulations over traditional process-based models. The authors made considerable efforts to set up experiments and perform relevant analyses. Results are compared with four lumped conceptual models and show that the LSTM models outperform the traditional models as well when applied in Great Britain.

The manuscript is generally well written and organized, figures and tables support the results.

My main concern is the degree of innovation and scientific significance of this work compared to already published works. This is a critical aspect of the manuscript that should be improved.

A large section of the manuscript is dedicated to a discussion of the advantages reported in the previously developed LSTM model. This discussion focuses on predictive ability, without much methodological improvement and innovations in ideas, that in turn may impair the scientific importance of the research.

In recent years, LSTM models have been broadly assessed. Most of these studies indicate the generally better performance of LSTM models over lumped models. The results reported in this manuscript seem to confirm the previously reported conclusions. By comparison, the analysis in Sections 4.2 and 4.3 is limited, whereas IMHO this is the most insightful section of the paper which deserves additional in-depth discussion. I think the authors should dedicate more space to discuss the implication of their findings.

Below are more detailed comments, questions, and suggestions that hopefully initiate a fruitful discussion and help improve the paper.

We thank the reviewer for their sincere comments and suggestions. We intend to make two major revisions to the paper based on these comments.

1) **Shorten certain sections**: Make the description of the LSTM/EA-LSTM experiment and the presentation of the predictive ability results more concise, moving some material and plots into supplementary information.
2) **More explicit description and discussion of novelty**: We propose to expand the sections that discuss in what conditions the LSTM produces different simulations to the lumped conceptual models (Sect 4.2) and the different performance in catchments where water is balanced vs. imbalanced (Sect 4.3).

We recognise the need to discuss the implications of these findings more fully. We propose three ways to do this:
1) More critically engaging with the experimental structure and outlining what is and is not possible to conclude with the given experiment.
2) Focus attention on the interpretation of the performance differences and away from the performance improvement of the LSTMs.
3) Expand our analysis of specific events and catchments in order to provide case studies of where there are large discrepancies in model performance.

**ABSTRACT:**

I would suggest mentioning the challenges in present LSTM applications for hydrological modeling and what is to be addressed, otherwise, it is difficult to tell the significance and necessity of the work.

Some major challenges of LSTM applications include:
1) How to incorporate uncertainty in inputs and outputs? See Klotz et al 2020.
2) How well do LSTM based models perform under changing climate conditions? See Sungmin, Dutra and Orth 2019.
3) How we learn from the performance improvement of LSTM based models, e.g. to diagnose missing process representations by comparison with the benchmark models.
4) How does the number of parameters in the LSTM model interact with issues of equifinality;

We are addressing the third challenge in this paper. One of the promises of deep learning (DL) is that DL models can help identify any limitations of hydrological data and process representations, conditional on there being a pattern to these limitations. The LSTM is a demonstrably effective architecture for modelling systems with short (long-term) and fast (short-term) signals, such as hydrological systems. The LSTM can efficiently extract information from large sample datasets to link meteorological inputs to discharge, simulating the catchment system. The next step is then learning how to extract this information from the LSTM. We have proposed and explored a number of ways of doing this, and outline one of our methods in this paper, intercomparing diagnostic measures (the different goodness-of-fit metrics for different parts of the hydrograph).

We will make these challenges and the key contribution and novelty of the study clearer in the abstract.

**INTRODUCTION:**

I do not think research gaps are well defined in the introduction. The research objectives should be motivated by the research gaps. The latter two of the three questions raised in the manuscript are related to overcoming limitations and model diagnosis, without indication of the explicit research gaps to be addressed. Are there some additional studies that investigate the correlation between LSTM model performance and catchment attributes?

The background should be more concise and emphasizes more about what is still to be investigated regarding the usage of LSTM models.

We agree with the need to make the introduction more concise and focused on the underlying goals. The introduction and objectives will be rewritten to more clearly reflect the research gaps remaining for LSTM based rainfall-runoff models.

The existing literature on LSTMs has focussed primarily on the skill of these models, as applied to CONUS, but has not:
   a) Performed a comparison with conceptual models in GB
   b) Used LSTM performance improvements to provide insights into the catchment characteristics and time-periods where LSTM models provide better simulations, and using this to diagnose the conditions in which the LSTM has a significant advantage.

To address these gaps the paper asks the following questions:
   1) How does the LSTM **perform in GB**, and how do the results compare against commonly used conceptual models? We use benchmark model performances to give context to the LSTM model performances (i.e. to demonstrate if the LSTM is competitive against other models). We use results from a previously published model benchmarking paper.
   2) How do we **extract information** from the spatial and temporal patterns in diagnostic measures? The aim is to relate these patterns back to the hydrological conditions in those catchments/time periods with the largest differences in model performance.

Furthermore, LSTM is but one of several machine learning frameworks used in rainfall-runoff modelling. Recent advances in evolutionary computation report theory guided and "hydrological informed" approaches that result in not only highly accurate but also readily interpretable models. See for example:

J Chadalawada, et al, 2020, Hydrologically Informed Machine Learning for Rainfall-Runoff Modeling: A Genetic Programming-Based Toolkit for Automatic Model Induction, Water Resources Research 56 (4), e2019WR026933

HMVV Herath, 2020, Hydrologically Informed Machine Learning for Rainfall-Runoff Modelling: Towards Distributed Modelling, Hydrology and Earth System Sciences Discussions, 1-42

I have read these papers and will include a more complete set of references with regards to machine learning based approaches to hydrological modelling. Although the major focus for our review here is on neural network / deep learning methods.

Line 77: It seems only THREE research questions are being proposed.

We will update this to reflect the three research questions outlined.

**METHODS:**

Section 2.3: It is more suitable to use the term "layer" (e.g., LSTM layer and EA LSTM layer) when describing the specific layer structure.

Yes that makes sense, and this allows us to incorporate the comment below about the "final layer", the fully connected layer used to map the hidden state vector to a single discharge prediction.

Line 158: Please keep consistent notation using curly quotes or straight quotes throughout the manuscript.

We will check and update notation for consistency.

Figure 2: In EA LSTM cell, is the input gate "i_t" or "i"? (see Equation 8)

This should read "**i**", since the input gate does not receive time varying inputs but only the catchment attributes (**A**). Hence the output of the input gate is a unique vector for each catchment (but *static over time*).

Lines 203-206: The fully connected layer should be a part of the model architecture. It seems strange to introduce them in this subsection (model training).

We agree, and will rewrite this.

Section 2.5.1: A brief description of the process-based models is required, especially what hydrological processes are included in the respective models because the discussion section involves the consideration of processes.

We completely agree that this is a necessary part of the methods section. We will include a more detailed description of the benchmark models, but aim not to repeat what has already been written by the original benchmarking paper (Lane et al 2019).

Furthermore, looking at all reviews, there is a consistent call for the paper to be made more concise. We will **rewrite the methods and experimental design** to be a shorter summary of the main components of LSTM based models that make them suitable for rainfall-runoff

modelling. We will aim to draw parallels with the traditional hydrological models, outlining the key similarities and differences between these models. We agree with all reviewers that there is no need to repeat the equations for the LSTM based models, in line with similar papers for other hydrological models.

**RESULTS AND DISCUSSION**

Table 3, Figure 3, Figure 4, Figure 6, and Figure 7: All the results seem to merely be used to show the outperformance of LSTM models than other models in various cases. I think this part should be more concise if the result is not out of expectations, and more other implications should be discussed from the results.

We agree and will move most of these plots into the supplementary information. We believe that keeping table 3 (the overall median goodness-of-fit metrics) Figure 3 (CDFs) and perhaps Figure 6 (seasonal NSE spatially). This should give an overview of the overall pattern, the spatial pattern and the seasonal pattern which form the three key goodness-of-fit intercomparisons.

Lines 496-503: The speculation of "connectivity" is interesting, while how the connectivity can be "learned" by LSTM models should be clarified, say whether the connectivity can be represented by hidden information within data or the model architecture (such as the memory of LSTM).

Great point. We believe that the information captured within the model architecture may be used to tell us something interesting about connectivity. The idea is that the vectors that represent the fast and short information processed by the LSTM (h_t and C_t) have learned something useful about summer (semi-arid) hydrology that we can extract and interpret.

Lines 505-507: A simple strategy to examine the speculation is to train an LSTM model with/without crop_perc included for checking its role in improving the representation of hydrology in those catchments with a strong agricultural signal.

This is a very useful suggestion. Ultimately, in order to properly account for the contribution of many different factors we would require a more comprehensive analysis, rather than performing a somewhat ad-hoc analysis on a single variable. All reviewers agree that we need to make the paper more concise, however, we intend on expanding the discussion of the hydrological conditions in which the LSTM outperforms the benchmarking models. Therefore, we intend to flag this topic as warranting further discussion in our upcoming paper on LSTM interpretability and propose to add a sentence to explain our intentions of pursuing this in future work.

---

## Author Comment (AC2)

**Response to Reviewers for Manuscript: Benchmarking Data-Driven Rainfall-Runoff Models in Great Britain**

**Anonymous Reviewer #2:**

Comments/Text of Anonymous Referee posted in **black**, our text in blue.

This manuscript investigates the following research questions:
- Are (regional) LSTM-based models able to simulate the rainfall-runoff process in GB and how do they compare against different, well established hydrological models?
- Can we extract any insights from this comparison, e.g. are there certain types of catchments that are consistently modeled better by one class of models (here LSTMs), which may hint to a missing representation for a dominant hydrological process in the other model class (here the benchmark models)?

Overall, I think this is a very good manuscript that only needs minor modifications. No additional experiments are required. The list of my suggestions might seem long in the first place, however most things should be very easy to fix. I tried to list everything I found, because I hope that will make the manuscript better. However, I am happily open to discuss any of my points.

Before I start with my comments, I want to highlight the points of the manuscript that I liked:
- The study is conducted on a large-sample, public dataset that was never used before for this kind of studies.
- All code is published and it seems trivial to reproduce the results of this manuscript.
- The benchmarking includes model outputs from different research groups, making it less likely that the model comparison is biased.
- The evaluation is performed on multiple metrics that account for different parts of the hydrograph.
- The discussion and analysis of the results w.r.t. the hydrological context/region was insightful.

We thank Reviewer #2 for their careful reading of the manuscript and thoughtful comments. They have identified the research aims of our paper and we are keen to incorporate their suggestions into our revision.

Next, a few general points:

**Format**

- The mathematical notation is inconsistent and not in line with the HESS guidelines. E.g.
- Vectors (e.g. all gates, the cell and hidden state, and the inputs at a particular

timestep) should be boldface italics lower case.
- Matrices (e.g. weights) should be printed in upper boldface roman (upright) font.
- Abbreviations are no in line with the HESS guidelines. E.g. "Figure" and "Equation" (as the entire word) should only be used at the beginning of a sentence. Mid-sentence "Fig." and "Eq." should be used.
- Dates are not in line with the HESS guidelines. The format of the dates should be dd mm yyyy (e.g. 31 December 2008).

Thank you. We will fix the notation in line with HESS guidelines. Based on the next comment, we will make the description of the models and experiment more concise. Therefore, we will either move this material into the supplementary information or remove it from the manuscript.

**Paper length**

- The manuscript is quite long but there is potential for shortening certain parts. I think a shorter, more concise paper will ultimately make this manuscript more read by people. I do have a few suggestions, where the manuscript could be shortened but feel free to ignore all of them if you would like these sections as is:

We agree with the need to shorten the paper for clarity and to avoid repetition. We believe that the steps outlined here are sensible and we propose to:
- Move repetitive figures to Appendices / Supplementary Information
- Make the description of the models and experimental setup more concise

- The model description of the LSTM and EA-LSTM is quite long and little information is added in comparison to the original manuscript that is cited. Personally, I don't think that the equations and the entire formal explanation is needed and in my opinion it could be removed. I think it would suffice to have a short, one paragraph explanation of the main difference between those two models (maybe with Fig. 2) and then link the reader to the original citation. We have seen quite a few LSTM publications recently, and similar to papers with traditional hydrological models, in my opinion it is not needed again and again to write down the model equations.

We agree that there is no need to rewrite the equations in every paper applying the LSTM technique. Therefore, we propose to update Section 2.3 to address two key points:
1) The LSTM, EA LSTM and their key differences
2) A hydrological justification for such an architecture (time series data; fast and slow signals)

- Very similar in my opinion is the section about the evaluation protocol. Personally, I don't think we need to see the equation of e.g. the Nash-Sutcliffe-Efficiency in every hydrology paper. Also the other equations could maybe be removed and you only keep a short explanation of the metrics with a reference to the original manuscripts. If you want to keep all equations, why not list them in e.g. a compact table like in Best et al. (2015).

We will update section 2.5.2 to more concisely outline the metrics we have chosen and link back to the original publications, rather than re-writing the equations for ourselves. As suggested, we propose to write a table for each of the metrics with columns describing:

1. what the purpose of such a metric is
2. a reference to the underlying paper and equations

- The manuscript contains a lot of figures. Generally I like that. However, due to the length of the manuscript, there are some figures that I think could be shortened or removed (or moved to the supplement). E.g. Figure 4 spans 3 pages, however most of these maps show the same pattern.

This was a very useful comment, and one that has been corroborated by all of the reviewers. We agree and will move most of these plots into the supplementary information. We propose to keep Table 3 (the overall median goodness-of-fit metrics) Figure3 (CDFs) and Figure 6 (seasonal NSE spatially). This should give an overview of the overall pattern, the spatial pattern and the seasonal pattern.

- Lastly, I was a bit confused about the section starting in L 452ff. At this point, we reached the discussion of the results. You evaluated and compared the LSTM-based models in hundreds of basins with established hydrological models. Why do you add another comparison in the discussion with two additional models in (only) 13 basins? I see no real motivation for this additional comparison and there are no new insights gained from this comparison. Personally, I think this section can be removed entirely, or I would like to see a better explanation why this additional comparison is wanted/needed and what we get out of it that we did not know from the first, very large-sample, comparison.

We agree and will remove the intercomparison with the process-based models. It was added in order to address potential concerns about ignoring a whole class of hydrological models. However, in hindsight, the comparison is somewhat ad-hoc. A more complete and rigorous experiment will be required to address such a comparison. However, in order to address the wider concerns that the paper should be more concise we propose to remove this comparison completely, in line with the reviewer's request (Table 4 and Lines 453-469).

**Minor line-by-line comments:**

- L 38: "account for temporal dependence using a series of recurrent layers": RNNs account for temporal dependencies by processing the input time series timestep by timestep. This can (as in your LSTM model) also be done by a single layer. Using a "series of recurrent layers" would mean to stack RNN layers, each with it's own set of weights.
Thank you for bringing that to our attention. That is indeed misleading and we will update this sentence accordingly.

- L 42: "Long Short Term Memory" -> "Long Short-Term Memory"
We will update this.

- L 50: The cited reference for the EA-LSTM did not investigate prediction in ungauged basins. It was done in a different publication, by the same authors though, in which they did not use the EA-LSTM however.
Apologies, we will fix the citation.

- L 77: "Our study poses the following four research questions:" Your enumeration only contains three research questions.
We will update this.

- L 109: Table 2 mentioned before Tab 1.
We will update this. We will reference the mathematical notation in Table 1 and do so above Table 2. Since we propose removing the equations for the LSTM/ EA LSTM the table of notation will also be shortened to reflect this change.

- L 114: "The static attributes we use to train the LSTM models are listed in Table 1." Table 1 does not list the static attributes but the notation of the mathematical symbols.
We propose to update this to read "... listed in Table 2"

(Next points are only important if you decide to keep Section 2.3)
Given Reviewer #2s comments, we propose to significantly change Section 2.3 in order to avoid repeating the equations from previous (other) LSTM papers. We will update the equations and move into supplementary information.

- In Eq. 1-4: To simplify, you can write $[[X\_t, A], h\_t-1]$ as $[X\_t, A, h\_t-1]$, since all three vectors are stacked. Although note, as stated above, that it should be boldface italics lowercase for all three vectors
We will update everything with HESS guidelines. Thank you very much for making that clear.

- L 159 "hs" is explained in L. 219 but not here. Maybe better to explain "hs" at the first occurrence and remove the explanation in L 219.
We will explain hs here and in the table since it is an important hyperparameter. We propose to explicitly outline the number of free parameters within the LSTM model, and will need to have defined hs in order to make this description precise.

- Figure 2: caption "we have 365 cells" maybe confusing with cells also regularly used to describe the number of memory cells. Maybe simply remove the last part of the Sentence.
We will update this as proposed.

- L 185 y_hat is not explained.
We will update the text and Table 1 with an explanation of y_hat. "$y\_hat$ represents the discharge values predicted by the model $M$".

- L 185, Eq. 12 M_theta (which is a function) should be typeset in roman (upright) font,

see HESS guidelines for mathematical symbols and functions.
We will update this as proposed.

- L 200 The following two sentences could be rephrased or maybe one could be removed:
"We used 21 static inputs (A). Each catchment was characterised using 21 individual features
describing the topographic, soil, land-cover, and climatic properties.". Maybe just write "Each
catchment was characterised using 21 individual features (A) describing the topographic, soil,
land-cover, and climatic properties.".
We will update this as proposed. (Thanks!)

- Table 2, The median for low_prec_freq is missing.
We will update the table with the Median( low_prec_freq ).

- L 207-208 Slightly repetitive to the preceding paragraphs. Can maybe be deleted!?
We will delete the sentence "We train the LSTM models on 669 gauges with training data from
our training period (1988–1997), which was chosen to match with the conceptual model
experiments we compare against (Lane et al., 2019). "

- L 212 Just to be sure, are you using the average discharge of the ensemble or are you later
reporting the average metric value, calculated as the mean/median over the ensemble
members? I think it is the former, but it would maybe help to be more explicit here.
You are correct in assuming the former. We will update this sentence to more clearly reflect the
ensemble averaging procedure. "We train 8 individual models with different random seeds. We
then present the average discharge of the ensemble and calculate the error metrics for this
ensemble-averaged discharge. This accounts for the random…."
(End of Section 2.3 comments)

 - L 247: "...chose parameters for the 4 lumped models from a grid of 10,000 parameters..." I
think the formulation is slightly wrong. It is not a "grid of 10.000" parameters", but they sampel
10.000 parameter sets from a grid defined by the user defined parameter boundaries. Maybe
"...chose parameters for the 4 lumped models by sampling 10,000 random parameter sets from
a grid with predefined parameter boundaries..."?

We propose to update the text with the following: "The results from the four benchmark models
we compare against from Lane et al (2019) are the best-estimate model outputs, obtained from
randomly sampling 10,000 parameter sets within a user-defined parameter space. The
best-estimate model was chosen using the NSE score". We also propose to update this
discussion to more critically engage with this comparison. The Lane et al paper was not only a
benchmarking study, but also focused on uncertainty quantification and that should be made
clearer to the reader. We propose to give a more critical reflection of the intercomparison
between the FUSE benchmark and the LSTM-based models trained here.

- L 245-254 What you describe here, especially the difference of how the model parameters are
selected, is indeed a huge difference. The section however, is quite long and you could maybe

try to rewrite this section in a more concise/structured way. If I understand you correctly, there are two main points you want to say:

- ● - Lane et al. "calibrate" their models by sampling 10.000 different parameter sets and evaluating the models on the entire data record, then picking the parameter set with the highest NSE. In contrast, you train your model on one data split, and you use a different data split (of unseen data) to evaluate the models and to calculate the metrics.
- ● - Lane et al. find individual parameter sets per basin. In contrast, you train one model with a single set of parameters for all basins at once.
- ● Maybe you find a way to shorten this section and boil it down to the main points.

We will update this section in line with comments from both Reviewer #1 and Reviewer #2. We propose to have a more in depth explanation of the benchmark models, outlining the structure of the four conceptual models that we compare against, and a more critical reflection of the intercomparison. The fact that the LSTM based models are compared on out-of-sample predictions is a key part of this critical discussion and will be included, albeit with a single sentence and in a more concise manner.

L 256f "An important difference between the LSTMs and the traditional hydrological models, is that traditional models perform best when calibrated for individual basins. The parameters that they use to produce simulations are unique to each basin. This often represents the state-of-the-art for traditional hydrological models." I think the last sentence can be removed, as you already said that in the first sentence. Although I agree, it might be good to have a reference for such a statement.
We will update this as proposed.

- L 256 I feel like such a sentence needs either a reference or an experiment.
We will update this as proposed.

- L 261ff "The conceptual models were calibrated and evaluated to produce simulated streamflows by Lane et al. (2019). We did not run these benchmarks ourselves. This is important because we have not biased the calibration of these models to favour the deep learning models. We have used the published time-series of model outputs to calculate performance scores for the conceptual Models." You can maybe remove this sentences, since you already stated the same at the beginning of Sect. 2.5.1. The only thing added here is the sentence of being unbiased. You could maybe add this to the first sentences of this paragraph as well. E.g. Maybe (L. 229) "We compare the performance of the LSTM based models against a range of lumped, conceptual models. To be unbiased on the model calibration, we used predicted discharge time series from Lane et al. (2019) who utilised the FUSE framework to train and evaluate four lumped conceptual models across Great Britain (Clark et al., 2008)."
We will update this as proposed, moving the claim that this prevents arguments of biasing the results to L229. Thank you.

- L 263 Why does using simulations from someone else help you to better understand the seasonal and geographical patterns?

This is a miscommunication on our part. We are arguing that the intercomparison of models enables you to compare the temporal and spatial patterns in model performance, helping to derive insights about which models do better where, and suggest hypotheses for why. We propose to remove this sentence because it simply outlines why an intercomparison of models is a good idea, which is somewhat self-explanatory.

- Table 3: Are the LSTM metrics the mean/median over the 10 repetition, or the metric value given the ensemble mean discharge? If the former, you could/should report the std/interquartile range as an error metric.
Yes we will include a measure of variability in order to reflect uncertainty. We will also update the hydrographs in the appendix with shaded areas to reflect the ensemble spread for each individual prediction.

- Table 3 and L 313: Since all models model the same basins, you should use the "paired" Wilcoxon test. Furthermore, I am a bit surprised by the results of %BiasFMS: Did you test for significance using the absolute metric values of the signed metric values? Because the LSTM is actually closer to zero as TOPMODEL, making it actually the better model. Since, in my opinion, neither over- nor unterestimating is better, you should test for significance using the absolute metric values per basin for the metrics that go from -inf to +inf (with zero being the best).
We used the "Paired Wilcoxon Test" (Scipy Function with the "alternative" parameter set to "two-sided"). We did not use the absolute values and will change that as proposed. What we did was:
   1) Calculate the paired wilcoxon test for each model intercomparison
         a) LSTM vs. TOPMODEL, SACRAMENTO, PRMS, VIC, EALSTM
         b) EALSTM vs. TOPMODEL, SACRAMENTO, PRMS, VIC
         c) TOPMODEL vs. , SACRAMENTO, PRMS, VIC
         d) SACRAMENTO vs. PRMS, VIC
         e) PRMS vs. VIC
   2) We presented only the results showing significant difference between the *best* model, (which was VIC for %BiasFMS). The difference was significant for the comparison with the LSTM but insignificant for TOPMODEL.
I think the confusion comes from the fact that the median score obscures the similarity in the distributions of catchment %BiasFMS scores (in this case between TOPMODEL, ARNOVIC and LSTM).

- L 332 The difference in low flow metrics is indeed interesting. I am not too familiar with the CAMELS GB data but I could imagine that one of the reasons might also be the difference between the two datasets (CAMELS US and GB). In CAMELS US, there are a number of basins that fall completely dry during long periods of the year, which are generally hard to model. What is "dry" in CAMELS GB, might not be in the spectrum that was reported as difficult for CAMELS US. So maybe the LSTM is not good at zero flow predictions but good for "lower flows" (with water)?!

The results do not rule out this hypothesis. We propose to include this idea, that US low-flows are more extreme than GB low-flows, in the description of these results. We propose to quantify the range that constitutes low flows / "arid" conditions in both datasets.

- Figure 4: As stated above, these are a lot of plots/pages. Maybe not all are necessary so the paper becomes shorter? E.g. for most metrics, the patterns are pretty much identical. There is only a visible difference in the pattern for %BiasFLV, where TOPMODEL is different to the other benchmark models. Maybe just include figures for one (or two?) metric(s) in the main paper and put all others into the supplementary?
We agree with this and propose to move all goodness-of-fit metrics except NSE into the supplementary material.

- L 360f "An initial hypothesis is that hydrological conditions in the drier catchments with groundwater transfers remain difficult to model, requiring time-varying parameters and more detailed representation of hydrogeological properties." Or maybe different/better inputs? Something like groundwater transfer might be hard to learn from the limited inputs that are used in this study.
We agree that groundwater transfers may be hard to learn from these inputs. Furthermore, since we do not explicitly connect the catchments in the LSTM-based models, i.e. no information flows from one catchment to another, there is no way that water transfers between catchments can be explicitly modelled. The best we can do is to examine the excess/deficit of water predicted for one gauge and try to match these with the excess/deficit of surrounding catchments. However, given the uncertainties in each of the components of the water balance (rainfall, evapotranspiration, discharge) this will still be highly uncertain. Ultimately, architectural innovations are beyond the scope of this study and remain a focus for future work.

- L 362 "...but further research should address how the LSTM might be further improved in these low-flow regimes." Here, you are saying that LSTM performance suffers in low-flow regimes, which would be inline with the studies you referenced above (see L332f).
We propose to make this clearer by including the hypothesis that the range of catchment aridity values is much smaller for GB compared with US studies, and therefore, the results do not necessarily disagree with the studies we reference.

- L 376f "This means that the LSTM is overpredicting low flows, with a larger bias in the South East." Slightly repetitive to L 375, consider rephrasing.
We propose to remove this sentence.

- L 386 "The largest difference from GB average is 0.03 NSE". Isn't the difference 0.05?
The difference was compared to the GB average (|0.88 - 0.91| for SWESW and |0.88 - 0.85| for ANG). We propose to change this comparison to be maximum difference between the regions, which would be 0.06 (ANG - SWESW, |0.85 - 0.91|).

- L 389 "...the conceptual models show are clearly more capable in..." -> "the conceptual models are clearly more capable in..."

We will update this as proposed.

- L 395 Delete "clearly".
We will update this as proposed.

- L 394ff I'm not sure if I agree with your summary. For me, it is almost easier to see the East-West gradient in the LSTM/EA-LSTM figures, because they switch from darker colors (all but JJA) to lighter colors (JJA), whereas the others have East-West gradient almost always (all but JJA), where as in JJA almost the entire map has lighter colors.
Good point. We propose to update this to read: "The East-West gradient in model performances can be seen for all models, particularly in JJA. However, the range of errors is smaller for the LSTM based models when compared with the conceptual models.". We propose to keep this figure since it efficiently shows both spatial and temporal patterns, as opposed to the majority of Figure 4 which shows the same pattern for different error metrics.

- L 397ff. Figure 7 and Fig. 8 are not linked in the text and the order of the figures should probably be switched to account for the occurrence in the text (map before cdf). Also: I'm not sure if Fig. 7 and Fig. 8 are needed or if their results can be described with 2-3 sentences. Since the pattern and the results are basically always the same, i.e. LSTM is generally better everywhere and at any time. So this could be a good opportunity to shorten the paper.
We agree. We propose to move Figure 7 and Figure 8 to Supplementary Information. The results from Figure 7 can be described using Figure 6 (that all models struggle most with summer flows - JJA) and Figure 8 results can be described from the map of NSE scores shown in Figure 4. We propose moving the other figures of Figure 4 to supplementary information.

- L 426f: "The catchment attributes alone are not sufficient to determine what information needs to be passed into the cell memory (Equation 8 compared with Equation 2). In other words, the LSTM learns more about the catchments' hydrological response to rainfall from the hydrographs themselves than from the static catchment attributes." I agree with the finding that EA-LSTM seems to be worse than LSTM, however I am not sure if I agree with the statement of these sentences. The EA-LSTM has the same discharge available to "learn from" and the LSTM has the same static attributes. You mention in other places that probably the main reason for the difference is that the EA-LSTM "freezes" one of the gates, making the EA-LSTM less flexible, which I think is the main reason for the difference.
- L 428 "For example, we can imagine a snowy catchment where we also need the temperature information to decide whether to store snow water in the network memory. The LSTM has this temperature information fed through the input gate (Equation 2), whereas the EA LSTM does not (Equation 8)." The EA-LSTM is still able to model snow, as you described in this example. The only difference is that for the EA-LSTM, the cell-update gate has to model the entire process (i.e. "that there is snow" and "how much" snow is added to the cell). In the standard LSTM, the two things can be modeled by two gates (input gate + cell update gate), making it more flexible to learn this process.
We agree and propose to shorten the discussion of EA LSTM performance.

"The EA LSTM is forced to keep the *input gate* static through time. The input gate receives only information about catchment attributes. This means that no time-varying information is passed through the EA LSTM input gate. In contrast, the LSTM gates receive information from both time-varying meteorological inputs and static catchment attributes."

- L 436 I think you mean the correct thing but just to clarify. As far as I understand, both models run on GPUs, the difference is that the standard LSTM makes use of a CUDA optimized implementation in the background, while the EA-LSTM is custom code.
We will update this to make the point clearer. You have interpreted this correctly.

- L 438 At first, I was confused if the deltas are the differences of the means or medians, which is explained then in the next sentence. Maybe you could move this explanation to the beginning?
We will update this as proposed. Thank you.

- L445ff Coming back to an earlier point of my review: I feel like it is worth repeating that you compare against the calibration period of the benchmark models. Most likely, a fair comparison, where you compare to out-of-sample periods of the benchmark models, would further increase the performance difference.
We propose a more critical reflection of the intercomparison between the FUSE benchmark and the LSTM-based models trained here. This will outline two things:
1) The FUSE benchmark models were concerned primarily with quantifying uncertainty as well as model performance
2) The FUSE model performance is compared on data used for calibration, in contrast the LSTM results are presented for out-of-sample time periods.

- L 478ff: I'm not familiar with all 4 conceptual models but if I'm not wrong, at least not all of them contain a snow-module in the setting used for this study. Maybe this does also explain some of the differences in North East Scotland?
You are correct this definitely explains some of the performance differential. Reviewer 3 has suggested removing these catchments from our intercomparison.

- L 490ff: Isn't another possible option based on the way how those models are trained? Imagine a basin that has constant low flow (or zero flow) for an extended period each year. All those timesteps yield little information that can be used to update the weights, since for all different meteorological inputs, the output would always have to be the same. So the underlying physical processes can only be inferred from those timesteps with varying discharge.
This is a very good point. We will outline this hypothesis as a competing explanation.

- L516ff I am not an expert with these models, but how strict is mass conservation really? We can't see more water than what has fallen as precipitation (upper bound) but there is no lower bound, or? Since the models are not calibrated on evapotranspiration, it can vary this model output at will, to e.g. remove less water from the system than it would evaporate in reality. Additionally, some conceptual hydrology models (e.g. SACRAMENTO) have an additional

option to remove water from the system that then does not reach the channel, which is the baseloss flow. This is another degree of freedom, which can be fitted at will, since the models are only calibrated on discharge. What I want to say is: Conceptual models can't "invent" water (e.g. by water transfer from a different catchment) but water can be removed at will. So personally, I don't think that a "leaking catchment" (L. 518) has to be a problem, or?

This is a very interesting point and something that we have discussed. The particular models that we benchmark against here were constrained to not remove any more water than the maximum defined by the input potential evapotranspiration. You are correct that conceptual models often have a baseloss flow, however, in the models used for comparison here, baseloss flow parameters were set to zero (i.e. excluded) and there is no baseloss flow. We propose to include a sentence to outline this in the revision.

- L 533ff "Alternatively, the fact that both LSTMs and conceptual models struggle in catchments where data does not meet the water balance constraints might suggest that human impacts on the hydrograph are ultimately unpredictable, such as abstraction and effluent returns." Or maybe just unpredictable from the given model inputs? Even if anthropogenic influences are included in the catchment attributes, water extraction is most likely a dynamic process and would require additional dynamic inputs. But conceptually, I don't see why a data-driven model should not be able to learn this process? The process is either driven by physical processes or by a human factor, which is most likely driven by a management plan. Both things could in theory be learned, given enough (informative) inputs.

We agree and propose to reformulate this argument. We are not trying to claim that these human abstractions are unpredictable, but that given the inputs that we have from CAMELS GB, it seems unlikely that the model is going to accurately reproduce human management decisions. Unless of course there is a pattern in these anomalies, i.e. after every large rainfall event the hydrograph is altered due to dams consistently allowing more water through. We propose to update this sentence to read: "Ultimately, neither LSTMs nor conceptual models produce accurate simulations in catchments where the underlying data does not meet water balance constraints. This could suggest that human management decisions are unpredictable from meteorological inputs without further dynamic inputs, such as timings of abstractions and effluent returns".

- L 539 Do you mean to link Figure 10? The link to Figure 11 is not clear to me.
Yes, Figure 10 shows the curved line which determines catchments that have "excess water". Figure 11 shows the distribution of NSE scores for these catchments compared with the catchments above this line.

- Figure 11: x-axis label (NSE) should be in capital letters
We will update this.

- L 551 Again, since this is the conclusion, worth that your comparison is biased (towards the lumped hydrology models), since you compare your hold-out period to their calibration period.

We will engage more critically with the intercomparison between the FUSE benchmark models and the LSTM. We agree that the hold-out test period is one component of this discussion.

References:

Best, M. J., Abramowitz, G., Johnson, H. R., Pitman, A. J., Balsamo, G., Boone, A., ... & Vuichard, N. (2015). The plumbing of land surface models: benchmarking model performance. Journal of Hydrometeorology, 16(3), 1425-1442.

---

## Author Comment (AC3)

**Response to Reviewers for Manuscript: Benchmarking Data-Driven Rainfall-Runoff Models in Great Britain**

**Anonymous Reviewer #3:**

Comments/Text of Anonymous Referee posted in **black**, our text in blue.

This paper describes two versions of a national scale deep learning hydrological model for GB and compares them to 4 conceptual hydrological models from the FUSE framework. The effectiveness of LSTM has been well established in previous studies, and so the novelty of this paper lies in its application to GB catchments. As the code, data and outputs are all freely available, I consider this to be a useful study to hydrologists concerned with modelling GB catchments. I wonder if given the limited scientific insights of this paper may be better placed in the Journal of Hydrology: Regional studies, or Environmental Modelling and Software rather than HESS.

I would like to commend the authors on a very clearly written paper- it was very easy to follow and understand.

We thank Reviewer #3 for their comments and effective summary of the paper. We take on board the claims about scientific novelty and propose to update the paper to reduce the emphasis on outlining the performance improvement of the LSTM compared with the conceptual models. We will do this in three ways:

1) Reduce the length of our (re)introduction of LSTM/EA LSTM methods, pointing readers towards the original papers that introduced these methods to hydrology.
2) Reduce the number of figures that demonstrate performance improvement of the LSTM in comparison with the conceptual models.
3) Increase the emphasis on exploring what we can learn from the differences in performances. We propose to rename and restructure section 4.2. to better reflect the focus on intercomparison of simulations in different catchment attribute conditions.

My major criticism of the paper is that the authors never demonstrate the **model's applicability to a changing climate**. Even if the application of LSTM (and all models that rely entirely on calibration) is only for near term flood forecasting, it is likely that we will be modelling events outside of the training data of the model with increasing frequency. I think that an alternative calibration/validation strategy should be examined where extreme events are left out of the calibration of the model, to provide some confidence in its ability to model beyond its training dataset.

We agree that understanding model performances on out-of-sample events is an exciting area of study. However, we believe the calls from all reviewers for a more concise paper mean that a complete exploration of this question is beyond the scope of this study.

My other major criticism is that the authors never **discuss the insights gained from the LSTM model**. There is no discussion of the sensitivity of the model to the different inputs and how the model ends up being structured. **They never provide any evidence to answer their third research question**. I think this would add a lot more value to the paper and make it worthy of publication in HESS. In the conclusion the authors state that this will come in a subsequent paper, but I think it would be more valuable here (and some of the detail of the calibration/validation could be moved to the supplementary information).

We thank Reviewer #3 for their identification of research question 3 as the most scientifically valuable contribution of the paper. Given that we are shortening Section 2.3, removing a number of plots and reducing the emphasis on the performance comparison, we propose to expand the discussion of how we can learn from the performance differences. We propose to expand the discussion and results to address the question: How do we extract information from the spatial and temporal patterns in diagnostic measures? The aim is to relate these patterns back to the hydrological conditions in those catchments/time periods with the largest differences in model performance.

**Some more specific comments follow:**

line 19: There are more modern PBSD models than SHE. Reference Parflow, SUMA, SHETRAN, Hydrogeosphere etc.
We will update the references to include links to more modern PBSD models. Thank you!

line 77: there are only 3 research questions
We will update this to read "three research questions".

Figure 1: You can format text in python to include superscripts "$mm\ day^{-1}$". Reduce point size- they are overlapping and obscuring each other.
We will update this: thank you!

Table 1: Nice! Very useful table. Temperature should be referred to with a capital T. Should Xt actually be Xn if it is representing the concatenation of dynamic and static input data for a single catchment?
We will update this as proposed. $X_t$ should probably be $X_{t,n}$ to reflect that it contains information for the target time period and the target catchment. Great spot!

line 176: Include the link to the prediction and error metrics at the end of the article too.
We will update this link to include the metrics. Thank you!

Table 2: Why these attributes? Was LSTM sensitive to all of these?

We incorporate these attributes since they cover three families of catchment characteristics that are important for hydrological modelling: soil structure, landcover types and climatic conditions. We used the same attributes as the previous LSTM/EA LSTM paper which was completed on data from CAMELS-US (Kratzert et al 2019).

line 220: What is an epoch? how does this relate to number of catchments/years of data?
An epoch is a single pass through all of the data. We will clarify this point in the text. So the LSTM is trained using an iterative gradient based optimisation method, stochastic gradient descent. An epoch means that every single sample (catchment-time) in the training period is used to update the weights. It does not affect the number of catchments or years of data, but it reflects the iterative nature of the training process, i.e. that at each iteration the model will see all of the data and make steps in multi-dimensional parameter space towards a more effective representation of the hydrological system (through better predictions, defined by our loss function, NSE).

Table 3: How is statistical significance calculated here? Double check that it is the appropriate method.
We used the "Paired Wilcoxon Test" (Scipy Function with the "alternative" parameter set to "two-sided"). We did not use the absolute values and will change that as proposed. We propose to clarify this in the text.

Figure 3: Nice figure
Thankyou! We also liked this composite figure.

line 366: I don't think that the **catchments with significant snowfall** should be included in the comparison if the snow modules of the conceptual models have not been turned on- this does not seem like a fair comparison. Recalculate the statistics leaving these catchments out.
This is a very interesting point and something that we propose to expand our discussion about. One of the key benefits of using the LSTMs, and data-driven approaches, is that we do not need to pre-specify the modules/structures that need to be included. Instead we can learn this from the data. We propose to recalculate the statistics for the intercomparison as suggested and to include this information in the supplementary information, however, we believe that by providing a GB-wide benchmark it is important to show the performance across all of the catchments that have been modelled, especially since these results are being published as a comparison for future work. We propose to expand our discussion and critical evaluation of the experimental setup.

line 367-371: this is a repetition of the previous paragraph.
Thank you for drawing our attention to this. We will remove the repeated sentence.

Figure 5: cut. This is a long paper with a lot of figures. I don't think this figure adds much to the maps.
We agree and propose to keep table 3 (the overall median goodness-of-fit metrics) Figure 3 (CDFs) and perhaps Figure 6 (seasonal NSE spatially) and remove the other figures or move

into supplementary information. This should give an overview of the overall pattern, the spatial pattern and the seasonal pattern which form the three key goodness-of-fit intercomparisons.

Figure 6. Label missing on the colorbar
We will update the colorbar as proposed.

Discussion: Cut all references to the physically based models. The comparisons are not rigorous and so should not be presented.
We agree and will remove this section from the manuscript.

figure 9: significant correlations are not clear. consider showing this in an alternative way.
We propose using larger font for the "*" signifying significant correlations. We agree they are currently too small to be useful.

line 537: I think this is the most interesting point in the whole paper- I would love to read a lot more about this in the discussion.
We agree that this warrants further discussion and propose to expand our discussion of this point, drawing on references from outside hydrology, and particularly from atmospheric science.

**Uncertainty**: I would like to see some discussion of training models to uncertain flows and uncertain inputs.

We agree that addressing uncertainty in inputs and outputs is of vital importance for hydrological modelling. While we feel that full treatment of uncertainty in inputs and outputs is beyond the scope of this manuscript, we want to propose two methods for addressing this important point.
1) Expand discussion of recent advances in LSTM based modelling with uncertainty inputs. In particular, see Kratzert et al (2021) on the performance boost of using multiple rainfall datasets, highlighting that the LSTM can flexibly incorporate new information from highly co-linear input datasets. Secondly, work by Klotz et al (2021) demonstrating three different methods for uncertainty quantification.
2) We have trained an ensemble of 8 models. We propose to make better use of this ensemble of models to represent uncertainty. Firstly, we propose to demonstrate the hydrological conditions in which discharge estimates are most uncertain. We propose to include a variance-based metric to reflect the uncertainty of the underlying ensemble, and to present the hydrographs in Appendix 2 with uncertainty bands reflecting the interquartile range of predictions.

References:
Klotz, Daniel, et al. "Uncertainty Estimation with Deep Learning for Rainfall–Runoff Modelling." Hydrology and Earth System Sciences Discussions (2021): 1-32.

Kratzert, Frederik, et al. "A note on leveraging synergy in multiple meteorological data sets with deep learning for rainfall–runoff modeling." Hydrology and Earth System Sciences 25.5 (2021): 2685-2703.

---

## Author Comment (AC5)

**Response to Reviewers for Manuscript: Benchmarking Data-Driven Rainfall-Runoff Models in Great Britain**

Comments/Text of Anonymous Referee posted in **black**, our text in **blue**.

We thank all of the anonymous reviewers for their positive and constructive comments on the manuscript. There are more complete responses to every reviewer comment below. In this paragraph we wanted to summarise the major changes we propose to make to the manuscript in light of these reviews.

- We propose to make the manuscript more concise by moving repetitive figures to Supplementary Information and by making the description of the models and experimental setup more precise.
- We propose to focus more attention on interpreting the performance differences and expanding our discussion and interpretation of the spatial and temporal patterns in performance differences. We aim to reduce the space used by plots that demonstrate performance improvement of the LSTM models.
- We will critically address the role of uncertainty in hydrological predictions and use our ensemble of 8 LSTMs to provide an estimate of predictive uncertainty, updating the plots and performance metrics (where appropriate) to reflect this ensemble spread.

Our responses were also informed by insightful comments about the paper directly from members of the hydrological community. We gratefully received these comments, and used them to inform our responses to the three solicited reviews.

---

## Author Response (AR1)

**Response to Reviewers for Manuscript: Benchmarking Data-Driven Rainfall-Runoff Models in Great Britain**

**Anonymous Reviewer #1:**

Comments/Text of Anonymous Referee posted in **black**, our text in **blue**.

The manuscript outlines an application of LSTM-based runoff models, which were introduced in previous studies (Kratzert et al, 2018; 2019). In the present contribution, the focus of analysis is catchments in Great Britain. Similar to previous studies, the objective is to demonstrate the competitive ability of LSTM in rainfall-runoff simulations over traditional process-based models. The authors made considerable efforts to set up experiments and perform relevant analyses. Results are compared with four lumped conceptual models and show that the LSTM models outperform the traditional models as well when applied in Great Britain.

The manuscript is generally well written and organized, figures and tables support the results.

My main concern is the degree of innovation and scientific significance of this work compared to already published works. This is a critical aspect of the manuscript that should be improved.

A large section of the manuscript is dedicated to a discussion of the advantages reported in the previously developed LSTM model. This discussion focuses on predictive ability, without much methodological improvement and innovations in ideas, that in turn may impair the scientific importance of the research.

In recent years, LSTM models have been broadly assessed. Most of these studies indicate the generally better performance of LSTM models over lumped models. The results reported in this manuscript seem to confirm the previously reported conclusions. By comparison, the analysis in Sections 4.2 and 4.3 is limited, whereas IMHO this is the most insightful section of the paper which deserves additional in-depth discussion. I think the authors should dedicate more space to discuss the implication of their findings.

Below are more detailed comments, questions, and suggestions that hopefully initiate a fruitful discussion and help improve the paper.

We thank the reviewer for their sincere comments and suggestions. We have made four major changes to the paper based on the reviewer's comments.

1) We have rewritten the *Section 2 Methods*, shortening *Section 2.3 An Overview of the LSTM and EALSTM* and moving a large part of the model description to *Appendix A: LSTM and EA LSTM Model Description*.
2) We have made the *Section 3 Results* more concise, reducing the number of figures in the main body of text. We have focussed more attention on the interesting patterns of LSTM performance, and away from the improved performance of the LSTM compared with the conceptual models. We have also focussed attention towards interpreting the model performances in the drier, groundwater dominated catchments of the South East.
3) We have expanded the discussion of these results, exploring more clearly our three research questions:
    a) *Section 4.1.1: How well do LSTM-based models simulate discharge in Great Britain?*
    b) *Section 4.1.2: How does the LSTM performance compare with the conceptual models used as benchmark?*
    c) *Section 4.1.3: Can we extract information from the spatial and temporal patterns in diagnostic measures?*
4) We have more critically engaged with our experimental structure and the intercomparison with the lumped conceptual models (*Section 2.4.1: Benchmark Models & Section 4.1.2: How does the LSTM performance compare with the conceptual models used as benchmark?*).

**ABSTRACT:**

I would suggest mentioning the challenges in present LSTM applications for hydrological modeling and what is to be addressed, otherwise, it is difficult to tell the significance and necessity of the work.

We have changed the abstract to more accurately reflect the gaps that our study is seeking to address. **L2-6** "*Previous studies have demonstrated the applicability of LSTM based models for rainfall-runoff modelling, however, LSTMs have not been tested on catchments in Great Britain (GB). Moreover, opportunities exist to use spatial and seasonal patterns in model performances to improve our understanding of hydrological processes, and to examine the advantages and disadvantages of LSTM-based models for hydrological simulation.*".

**INTRODUCTION:**

I do not think research gaps are well defined in the introduction. The research objectives should be motivated by the research gaps. The latter two of the three questions raised in the manuscript are related to overcoming limitations and model diagnosis, without indication of the explicit research gaps to be addressed. Are there some additional studies that investigate the correlation between LSTM model performance and catchment attributes?

The background should be more concise and emphasizes more about what is still to be investigated regarding the usage of LSTM models.

We have substantially changed the introduction to address the reviewer's comment. We explicitly outline the research gaps on **L57-68**. Our research questions then follow these gaps and are outlined on **L69-72**.

Furthermore, LSTM is but one of several machine learning frameworks used in rainfall-runoff modelling. Recent advances in evolutionary computation report theory guided and "hydrological informed" approaches that result in not only highly accurate but also readily interpretable models. See for example:

J Chadalawada, et al, 2020, Hydrologically Informed Machine Learning for Rainfall‑Runoff Modeling: A Genetic Programming‑Based Toolkit for Automatic Model Induction, Water Resources Research 56 (4), e2019WR026933

HMVV Herath, 2020, Hydrologically Informed Machine Learning for Rainfall-Runoff Modelling: Towards Distributed Modelling, Hydrology and Earth System Sciences Discussions, 1-42

We have updated our literature review to more accurately represent the diversity of data-driven modelling approaches. This can be seen on **L24-31**.

 Line 77: It seems only THREE research questions are being proposed.

Updated as proposed.

**METHODS:**

Section 2.3: It is more suitable to use the term "layer" (e.g., LSTM layer and EA LSTM layer) when describing the specific layer structure.

We have moved the discussion of the LSTM and EA LSTM structure to **L507-557** *Appendix A: LSTM and EA LSTM Model Description*.

Line 158: Please keep consistent notation using curly quotes or straight quotes throughout the manuscript.

Updated as proposed.

Figure 2: In EA LSTM cell, is the input gate "i_t" or "i"? (see Equation 8)

This has been updated as proposed.

Lines 203-206: The fully connected layer should be a part of the model architecture. It seems strange to introduce them in this subsection (model training).

We include this information in the complete description of the model architectures in Appendix A, **L556-557**.

Section 2.5.1: A brief description of the process-based models is required, especially what hydrological processes are included in the respective models because the discussion section involves the consideration of processes.

We include a more complete description of the lumped conceptual models from **L196-201**.

**RESULTS AND DISCUSSION**

Table 3, Figure 3, Figure 4, Figure 6, and Figure 7: All the results seem to merely be used to show the outperformance of LSTM models than other models in various cases. I think this part should be more concise if the result is not out of expectations, and more other implications should be discussed from the results.

We have kept the CDFs of catchment metrics (Figure 1), the maps showing seasonal NSE scores (Figure 2). Previous figures that are no longer included have been moved to Appendices, Appendix E: Spatial Performances of Error Metrics. We have reduced the focus of the results on the outperformance of the LSTM compared with the benchmark models.

Lines 496-503: The speculation of "connectivity" is interesting, while how the connectivity can be "learned" by LSTM models should be clarified, say whether the connectivity can be represented by hidden information within data or the model architecture (such as the memory of LSTM).

We have updated the text to reflect how connectivity information could be learned: "Connectivity information could be represented by the hidden state ($h_t$), or cell state vectors ($C_t$)" **L460-461**.

Lines 505-507: A simple strategy to examine the speculation is to train an LSTM model with/without crop_perc included for checking its role in improving the representation of hydrology in those catchments with a strong agricultural signal.

This is a very useful suggestion. Ultimately, in order to properly account for the contribution of many different factors we would require a more comprehensive analysis, rather than performing a somewhat ad-hoc analysis on a single variable. All reviewers agree that we need to make the paper more concise, however, we intend on expanding the discussion of the hydrological conditions in which the LSTM outperforms the benchmarking models. Therefore, we have flagged this ablation study (removing inputs from the model training) as a topic warranting further discussion in our upcoming paper on LSTM interpretability and added a sentence to explain our intentions of pursuing this in future work.

We have updated the text to "*In order to test this hypothesis, one could perform an ablation study, removing input features and determining the impact on model performances.*

*Alternatively, sensitivity analysis could be used to determine the relative contribution of the input features to the discharge prediction, thus revealing what input features are important for the model simulations. We intend to pursue this idea in upcoming papers.* ” (**L466-470**).

**Anonymous Reviewer #2:**

Comments/Text of Anonymous Referee posted in **black**, our text in **blue**.

This manuscript investigates the following research questions:
- Are (regional) LSTM-based models able to simulate the rainfall-runoff process in GB and how do they compare against different, well established hydrological models?
- Can we extract any insights from this comparison, e.g. are there certain types of catchments that are consistently modeled better by one class of models (here LSTMs), which may hint to a missing representation for a dominant hydrological process in the other model class (here the benchmark models)?

Overall, I think this is a very good manuscript that only needs minor modifications. No additional experiments are required. The list of my suggestions might seem long in the first place, however most things should be very easy to fix. I tried to list everything I found, because I hope that will make the manuscript better. However, I am happily open to discuss any of my points.

Before I start with my comments, I want to highlight the points of the manuscript that I liked:
- The study is conducted on a large-sample, public dataset that was never used before for this kind of studies.
- All code is published and it seems trivial to reproduce the results of this manuscript.
- The benchmarking includes model outputs from different research groups, making it less likely that the model comparison is biased.
- The evaluation is performed on multiple metrics that account for different parts of the hydrograph.
- The discussion and analysis of the results w.r.t. the hydrological context/region was insightful.

We thank Reviewer #2 for their careful reading of the manuscript and thoughtful comments. They have identified the research aims of our paper and we are keen to incorporate their suggestions into our revision.

Next, a few general points:

**Format**

- The mathematical notation is inconsistent and not in line with the HESS guidelines. E.g.
- Vectors (e.g. all gates, the cell and hidden state, and the inputs at a particular timestep) should be boldface italics lower case.
- Matrices (e.g. weights) should be printed in upper boldface roman (upright) font.

- Abbreviations are no in line with the HESS guidelines. E.g. "Figure" and "Equation" (as the entire word) should only be used at the beginning of a sentence. Mid-sentence "Fig." and "Eq." should be used.
- Dates are not in line with the HESS guidelines. The format of the dates should be dd mm yyyy (e.g. 31 December 2008).

We have updated all notation and abbreviations to be in line with the HESS guidelines. Thank you very much for the relevant information included in this comment.

**Paper length**

- The manuscript is quite long but there is potential for shortening certain parts. I think a shorter, more concise paper will ultimately make this manuscript more read by people. I do have a few suggestions, where the manuscript could be shortened but feel free to ignore all of them if you would like these sections as is:

We have moved a number of figures to the appendices (from the old manuscript Fig 1, Fig 2, Fig 4, Fig 5) and removed others (from the old manuscript Fig 7). We have kept the CDFs of catchment metrics (*Figure 1*), the maps showing seasonal NSE scores (*Figure 2*), the correlation of catchment attributes and model performance scores (*Figure 4*) the Budyko curves (*Figure 5*) and histograms showing model performances in "leaky" catchments (*Figure 5*).

- The model description of the LSTM and EA-LSTM is quite long and little information is added in comparison to the original manuscript that is cited. Personally, I don't think that the equations and the entire formal explanation is needed and in my opinion it could be removed. I think it would suffice to have a short, one paragraph explanation of the main difference between those two models (maybe with Fig. 2) and then link the reader to the original citation. We have seen quite a few LSTM publications recently, and similar to papers with traditional hydrological models, in my opinion it is not needed again and again to write down the model equations.

We have rewritten *Section 2.2 An Overview of the LSTM and EALSTM* (**L101-124**) to summarise the differences between the two models, and moved a more complete analysis into *Appendix A: LSTM and EA LSTM Model Description*. The wiring diagram has been moved into the Appendix (*Figure A1*).

- Very similar in my opinion is the section about the evaluation protocol. Personally, I don't think we need to see the equation of e.g. the Nash-Sutcliffe-Efficiency in every hydrology paper. Also the other equations could maybe be removed and you only keep a short explanation of the metrics with a reference to the original manuscripts. If you want to keep all equations, why not list them in e.g. a compact table like in Best et al. (2015).

We have updated *Section 2.4.2: Evaluation Metrics* to be much more concise. We have kept only a short explanation of the metrics and a reference to the original papersas proposed.

- The manuscript contains a lot of figures. Generally I like that. However, due to the length of the manuscript, there are some figures that I think could be shortened or removed (or moved to the supplement). E.g. Figure 4 spans 3 pages, however most of these maps show the same pattern.

We moved old Figure 4 and Figure 5, to the appendices (*Figure E1, E2*). We have also removed the old Figure 7. This has substantially shortened the paper.

- Lastly, I was a bit confused about the section starting in L 452ff. At this point, we reached the discussion of the results. You evaluated and compared the LSTM-based models in hundreds of basins with established hydrological models. Why do you add another comparison in the discussion with two additional models in (only) 13 basins? I see no real motivation for this additional comparison and there are no new insights gained from this comparison. Personally, I think this section can be removed entirely, or I would like to see a better explanation why this additional comparison is wanted/needed and what we get out of it that we did not know from the first, very large-sample, comparison.

We have removed this section as proposed.

**Minor line-by-line comments:**

- L 38: "account for temporal dependence using a series of recurrent layers": RNNs account for temporal dependencies by processing the input time series timestep by timestep. This can (as in your LSTM model) also be done by a single layer. Using a "series of recurrent layers" would mean to stack RNN layers, each with it's own set of weights.
We have removed the line in question.

- L 42: "Long Short Term Memory" -> "Long Short-Term Memory"
Updated as proposed.

- L 50: The cited reference for the EA-LSTM did not investigate prediction in ungauged basins. It was done in a different publication, by the same authors though, in which they did not use the EA-LSTM however.
We have rewritten this subsection and removed the sentence in question.

- L 77: "Our study poses the following four research questions:" Your enumeration only contains three research questions.
We have updated as proposed (**L57**).

- L 109: Table 2 mentioned before Tab 1.
We have updated this as proposed (**L97**).

- L 114: "The static attributes we use to train the LSTM models are listed in Table 1." Table 1 does not list the static attributes but the notation of the mathematical symbols.
We have updated this section, the reference to the static variables is now as above (**L97**).

(Next points are only important if you decide to keep Section 2.3)
We have updated Section 2.3, moving the majority of offending sentences to *Appendix A*.

- In Eq. 1-4: To simplify, you can write [[X_t, A], h_t-1] as [X_t, A, h_t-1], since all three vectors are stacked. Although note, as stated above, that it should be boldface italics lowercase for all three vectors
We have updated everything in line with HESS guidelines (**L527-532**; **L536-540**).

- L 159 "hs" is explained in L. 219 but not here. Maybe better to explain "hs" at the first occurrence and remove the explanation in L 219.
This is now explained in *Table 2* and introduced in the text in **L164**.

- Figure 2: caption "we have 365 cells" maybe confusing with cells also regularly used to describe the number of memory cells. Maybe simply remove the last part of the Sentence.
We have updated this as proposed.

- L 185 y_hat is not explained.
We have updated *Table 2* to include y_hat.

- L 185, Eq. 12 M_theta (which is a function) should be typeset in roman (upright) font, see HESS guidelines for mathematical symbols and functions.
We have updated this as proposed (Table 2; **L139**).

- L 200 The following two sentences could be rephrased or maybe one could be removed: "We used 21 static inputs (A). Each catchment was characterised using 21 individual features describing the topographic, soil, land-cover, and climatic properties.". Maybe just write "Each catchment was characterised using 21 individual features (A) describing the topographic, soil, land-cover, and climatic properties.".
We have updated on **L153**: "*We selected 21 individual features describing each catchment's topographic, soil, land-cover, and climatic properties as static inputs (A)*".

- Table 2, The median for low_prec_freq is missing.
We have updated *Table 1*.

- L 207-208 Slightly repetitive to the preceding paragraphs. Can maybe be deleted!?
Deleted as proposed.

- L 212 Just to be sure, are you using the average discharge of the ensemble or are you later reporting the average metric value, calculated as the mean/median over the ensemble members? I think it is the former, but it would maybe help to be more explicit here.

We have updated text on **L238-239** which reads: "*For the LSTM-based models the evaluation metrics are calculated given the average discharge of the ensemble*". We have also updated the caption of *Table 3* to read: "*We have shown the median catchment score for the metric given the mean simulated discharge of our ensemble*".
(End of Section 2.3 comments)

 - L 247: "...chose parameters for the 4 lumped models from a grid of 10,000 parameters..." I think the formulation is slightly wrong. It is not a "grid of 10.000" parameters", but they sampel 10.000 parameter sets from a grid defined by the user defined parameter boundaries. Maybe "...chose parameters for the 4 lumped models by sampling 10,000 random parameter sets from a grid with predefined parameter boundaries..."?

We have updated the text to read: "*The benchmark study provides an assessment of conceptual model simulation performances across a large sample of GB catchments,  and  also  quantifies uncertainty  in  hydrological  simulations  due  to  parameter  uncertainty  and  model  structural uncertainty (Lane et al., 2019). Parameter values for each conceptual model were selected from 10,000 simulations of multi-dimensional parameter space. The best-estimate model parameter values were selected from these 10,000 samples using the Nash-Sutcliffe Efficiency score.*" (**L203-206**).

- L 245-254 What you describe here, especially the difference of how the model parameters are selected, is indeed a huge difference. The section however, is quite long and you could maybe try to rewrite this section in a more concise/structured way. If I understand you correctly, there are two main points you want to say:
  - - Lane et al. "calibrate" their models by sampling 10.000 different parameter sets and evaluating the models on the entire data record, then picking the parameter set with the highest NSE. In contrast, you train your model on one data split, and you use a different data split (of unseen data) to evaluate the models and to calculate the metrics.
  - - Lane et al. find individual parameter sets per basin. In contrast, you train one model with a single set of parameters for all basins at once.
  - Maybe you find a way to shorten this section and boil it down to the main points.

We have expanded our discussion to more critically engage with the experimental differences between our experiment and the approach taken for the conceptual model experiments. This discussion can be found on **L202-234**.

L 256f "An important difference between the LSTMs and the traditional hydrological models, is that traditional models perform best when calibrated for individual basins. The parameters that they use to produce simulations are unique to each basin. This often represents the state-of-the-art for traditional hydrological models." I think the last sentence can be removed, as you already said that in the first sentence. Although I agree, it might be good to have a reference for such a statement.
Updated as proposed, "*Finally, the LSTM-based models are trained on all basins, with a single set of weights for the whole of GB. Therefore, these LSTM models are regional models that are*

*able to reproduce behaviours across Great Britain. In contrast, most hydrological models perform best when calibrated on individual basins (Beven, 2006)."* **L419-420**.

- L 256 I feel like such a sentence needs either a reference or an experiment.
We have updated as proposed "*In contrast, most hydrological models perform best when calibrated on individual basins (Beven, 2006)."* **L419-420**.

- L 261ff "The conceptual models were calibrated and evaluated to produce simulated streamflows by Lane et al. (2019). We did not run these benchmarks ourselves. This is important because we have not biased the calibration of these models to favour the deep learning models. We have used the published time-series of model outputs to calculate performance scores for the conceptual Models." You can maybe remove this sentences, since you already stated the same at the beginning of Sect. 2.5.1. The only thing added here is the sentence of being unbiased. You could maybe add this to the first sentences of this paragraph as well. E.g. Maybe (L. 229) "We compare the performance of the LSTM based models against a range of lumped, conceptual models. To be unbiased on the model calibration, we used predicted discharge time series from Lane et al. (2019) who utilised the FUSE framework to train and evaluate four lumped conceptual models across Great Britain (Clark et al., 2008)."
We updated this sentence to: "*To be unbiased on the model calibration, we used simulated discharge time series from Lane et al. (2019) who calibrated and evaluated these four conceptual models on 1000 catchments across Great Britain*" (**L189-191**).

- L 263 Why does using simulations from someone else help you to better understand the seasonal and geographical patterns?
We removed the sentence as it was unclear.

- Table 3: Are the LSTM metrics the mean/median over the 10 repetition, or the metric value given the ensemble mean discharge? If the former, you could/should report the std/interquartile range as an error metric.
The LSTM metrics are the metric value given the ensemble mean discharge. We have updated text on **L238-239** which reads: "For the LSTM-based models the evaluation metrics are calculated given the average discharge of the ensemble". We have also updated the caption of *Table 3* to read: "We have shown the median catchment score for the metric given the mean simulated discharge of our ensemble".

- Table 3 and L 313: Since all models model the same basins, you should use the "paired" Wilcoxon test. Furthermore, I am a bit surprised by the results of %BiasFMS: Did you test for significance using the absolute metric values of the signed metric values? Because the LSTM is actually closer to zero as TOPMODEL, making it actually the better model. Since, in my opinion, neither over- nor underestimating is better, you should test for significance using the absolute metric values per basin for the metrics that go from -inf to +inf (with zero being the best).
We used the "Paired Wilcoxon Test" (Scipy Function with the "alternative" parameter set to "two-sided"). We clarified this in the revised manuscript: **L261**.

Our process was as follows:
1) Calculate the paired wilcoxon test for each model intercomparison and for each statistic:
    a) LSTM vs. TOPMODEL, SACRAMENTO, PRMS, VIC, EALSTM
    b) EALSTM vs. TOPMODEL, SACRAMENTO, PRMS, VIC
    c) TOPMODEL vs. , SACRAMENTO, PRMS, VIC
    d) SACRAMENTO vs. PRMS, VIC
    e) PRMS vs. VIC
2) We presented only the results showing significant difference between the *best* model, (which was VIC for %BiasFMS, with the median score closest to zero). The difference was significant for the comparison with the LSTM but insignificant for TOPMODEL.

I think the confusion comes from the fact that **the median score obscures the similarity in the distributions** of catchment %BiasFMS scores (in this case between TOPMODEL, ARNOVIC and LSTM). Also the median of the raw BiasFMS is a little confusing, because the absolute BiasFMS scores clearly show that the LSTM outperforms the other models, but in the spirit of fairness I wanted to be consistent in the application of the metric across all metrics and all models.

The raw catchment fms: (medians shown as dashed lines and score in the keys)

[Figure]

The absolute BiasFMS distributions:

[Figure]

L 332 The difference in low flow metrics is indeed interesting. I am not too familiar with the CAMELS GB data but I could imagine that one of the reasons might also be the difference between the two datasets (CAMELS US and GB). In CAMELS US, there are a number of basins that fall completely dry during long periods of the year, which are generally hard to model. What is "dry" in CAMELS GB, might not be in the spectrum that was reported as difficult for CAMELS US. So maybe the LSTM is not good at zero flow predictions but good for "lower flows" (with water)?!

We updated the text to reflect this comment, arguing that the results show a confirmation of the worse performance in drier catchments (**L439-440**). We removed the statement about the performance improvement relative to the results in the US.

These are the 28 catchments with the highest aridity, and we can see there are a small number of ephemeral streams in the GB dataset.

[Figure]

- Figure 4: As stated above, these are a lot of plots/pages. Maybe not all are necessary so the paper becomes shorter? E.g. for most metrics, the patterns are pretty much identical. There is only a visible difference in the pattern for %BiasFLV, where TOPMODEL is different to the other benchmark models. Maybe just include figures for one (or two?) metric(s) in the main paper and put all others into the supplementary?

We moved all of these spatial plots to *Appendix E: Spatial Performances of Error Metrics*, *Figure E1*.

- L 360f "An initial hypothesis is that hydrological conditions in the drier catchments with groundwater transfers remain difficult to model, requiring time-varying parameters and more detailed representation of hydrogeological properties." Or maybe different/better inputs? Something like groundwater transfer might be hard to learn from the limited inputs that are used in this study.

We have significantly expanded our discussion about the difficulties of learning groundwater transfers, or subsurface dynamics in catchments with significant subsurface flow pathways. Key sentences relating to this comment:

- "*This suggests that the underlying data does not contain sufficient information to model the full range of processes that influence the hydrograph in these catchments, including groundwater and abstractions. The catchment averaged information on soil texture (sand-silt-clay)provides a coarse proxy for catchment porosity. Furthermore, further data, such as groundwater time-series, might be necessary to obtain more accurate discharge predictions.*" **L447-456**.
- And in the conclusions: "*Finally, the data may not contain sufficient information to capture the percolation and connectivity dynamics that drive hydrological behaviour in catchments with significant groundwater processes*" (L**507-509**)

- L 362 "...but further research should address how the LSTM might be further improved in these low-flow regimes." Here, you are saying that LSTM performance suffers in low-flow regimes, which would be inline with the studies you referenced above (see L332f).

We have updated the text as proposed: "*The LSTM shows a performance decline in drier conditions (Fig. 4). This confirms the findings of other DL studies in the US, where the LSTM also struggled to reproduce hydrographs in drier conditions (Kratzert et al., 2019, 2018)*" (**L439-440**).

- L 376f "This means that the LSTM is overpredicting low flows, with a larger bias in the South East." Slightly repetitive to L 375, consider rephrasing.
We have removed this sentence.

- L 386 "The largest difference from GB average is 0.03 NSE". Isn't the difference 0.05?
The difference was compared to the GB average (|0.88 - 0.91| for SWESW and |0.88 - 0.85| for ANG). We have however, removed this section, and the figure has been moved to the Appendix *Figure E2*.

- L 389 "...the conceptual models show are clearly more capable in..." -> "the conceptual models are clearly more capable in..."
We have removed this section and replaced it with a discussion of where the conceptual models perform well, "*The catchments where the comparative performance difference is small, i.e. where the conceptual models perform almost as well as the LSTM, reflect areas where the conceptual models capture the majority of the information from the data, and the conceptual model well represents the hydrological process. This is the case in West Scotland, North West England & NorthWales and North East England (see Appendix Fig. E2).*" (**L422-425**).

- L 395 Delete "clearly".
Updated as proposed: "*The East-West gradient in model performances can be seen for all models, particularly in JJA*" **L291-292**.

- L 394ff I'm not sure if I agree with your summary. For me, it is almost easier to see the East-West gradient in the LSTM/EA-LSTM figures, because they switch from darker colors (all but JJA) to lighter colors (JJA), whereas the others have East-West gradient almost always (all but JJA), where as in JJA almost the entire map has lighter colors.
We updated this to read: "*The East-West gradient in model performances can be seen for all models, particularly in JJA. However, the range of errors is smaller for the LSTM based models when compared with the conceptual models.*" (**L291-293**).

- L 397ff. Figure 7 and Fig. 8 are not linked in the text and the order of the figures should probably be switched to account for the occurrence in the text (map before cdf). Also: I'm not sure if Fig. 7 and Fig. 8 are needed or if their results can be described with 2-3 sentences. Since the pattern and the results are basically always the same, i.e. LSTM is generally better everywhere and at any time. So this could be a good opportunity to shorten the paper.
We removed the old Figure 7 from the manuscript. Figure 8 results were kept in the new *Figure 3*, since the Delta NSE metrics are important for analysing the differences in model performances explicitly. **L305-329**.

- L 426f: "The catchment attributes alone are not sufficient to determine what information needs to be passed into the cell memory (Equation 8 compared with Equation 2). In other words, the LSTM learns more about the catchments' hydrological response to rainfall from the hydrographs themselves than from the static catchment attributes." I agree with the finding that EA-LSTM seems to be worse than LSTM, however I am not sure if I agree with the statement of these sentences. The EA-LSTM has the same discharge available to "learn from" and the LSTM has the same static attributes. You mention in other places that probably the main reason for the difference is that the EA-LSTM "freezes" one of the gates, making the EA-LSTM less flexible, which I think is the main reason for the difference.

*We have updated this as proposed, the text has been updated to read: "The EA LSTM is constrained to treat information that does not vary over time (catchment attributes) separately from information that varies over time (hydro-meteorological forcings). However, the constraint penalizes performance, which was also found by (Kratzert et al. 2019). The EA LSTM, in contrast, is forced to keep the input gate static through time. The input gate receives only information about catchment attributes. This means that no time-varying information is passed through the EA LSTM input gate. In contrast, the LSTM gates receive information from both time-varying meteorological inputs and static catchment attributes. The under performance of the EA LSTM relative to the LSTM suggests that this regularisation hurts performance in out-of-sample conditions." (**L394-399**)*

- L 428 "For example, we can imagine a snowy catchment where we also need the temperature information to decide whether to store snow water in the network memory. The LSTM has this temperature information fed through the input gate (Equation 2), whereas the EA LSTM does not (Equation 8)." The EA-LSTM is still able to model snow, as you described in this example. The only difference is that for the EA-LSTM, the cell-update gate has to model the entire process (i.e. "that there is snow" and "how much" snow is added to the cell). In the standard LSTM, the two things can be modeled by two gates (input gate + cell update gate), making it more flexible to learn this process.

*We removed the discussion of the snow process as above and focused the discussion on the key points, as shown above (**L394-399**).*

- L 436 I think you mean the correct thing but just to clarify. As far as I understand, both models run on GPUs, the difference is that the standard LSTM makes use of a CUDA optimized implementation in the background, while the EA-LSTM is custom code.

*We have updated this to read: "It is worth noting that the LSTM and EA-LSTM also differ in terms of practical computational requirements. The LSTM trains much faster than the EA-LSTM. The LSTM will train 30 epochs in 1 hour, compared with 30 epochs in 10 hours for the EA-LSTM. This is due to the LSTM being an in-built Pytorch (v.1.7.1) function that makes use of CUDA optimised code (for running the models on a GPU). In contrast, the EA-LSTM relies on custom code without the CUDA enabled optimisations." (**L401-404**)*

- L 438 At first, I was confused if the deltas are the differences of the means or medians, which is explained then in the next sentence. Maybe you could move this explanation to the beginning?
We have updated this as proposed (**L307-315**).

- L445ff Coming back to an earlier point of my review: I feel like it is worth repeating that you compare against the calibration period of the benchmark models. Most likely, a fair comparison, where you compare to out-of-sample periods of the benchmark models, would further increase the performance difference.

We have updated the text as proposed and included this point in the methods and reiterated in the discussion:
- **L223-225**: "*Therefore, the LSTM is evaluated on out-of-sample (in time) data, whereas, the conceptual model parameters were calibrated on data included in the evaluation period (in-sample evaluation).*"
- **L415-416**: "*Another difference is that the LSTM diagnostic scores are calculated on out-of-sample predictions, compared with the in-sample predictions for the benchmark conceptual models.* "

- L 478ff: I'm not familiar with all 4 conceptual models but if I'm not wrong, at least not all of them contain a snow-module in the setting used for this study. Maybe this does also explain some of the differences in North East Scotland?

We have updated the text to make this a clear conclusion:
- **L286-288**: "*We suggest that these differences in performance are due to the low rainfall and chalk aquifer in the South East of England, and to the lack of snow modules incorporated into the conceptual models for North East Scotland.*"
- **L324-326**: "*The conceptual models lack a snow module, and are therefore unable to capture snow melt or frozen ground processes, which are especially important in winter (DJF) and spring (MAM)*"
- **L429-432**: "*The performance differences in North East Scotland are very likely a result of the ability of the LSTM to learn a representation of snow processes from the input data, whereas, the conceptual models were simulating these catchments without a snow module.*"

- L 490ff: Isn't another possible option based on the way how those models are trained? Imagine a basin that has constant low flow (or zero flow) for an extended period each year. All those timesteps yield little information that can be used to update the weights, since for all different meteorological inputs, the output would always have to be the same. So the underlying physical processes can only be inferred from those timesteps with varying discharge.
We have included this as a hypothesis for the results: **L440-442**: "*Basins that have long periods of low flow contain little information, since changing meteorological inputs co-occurs with very little change in the target discharge. Therefore, the physical process relating meteorological inputs to river discharge can only be inferred from those catchments with varying discharge.* "

- L516ff I am not an expert with these models, but how strict is mass conservation really? We can't see more water than what has fallen as precipitation (upper bound) but there is no lower bound, or? Since the models are not calibrated on evapotranspiration, it can vary this model output at will, to e.g. remove less water from the system than it would evaporate in reality. Additionally, some conceptual hydrology models (e.g. SACRAMENTO) have an additional option to remove water from the system that then does not reach the channel, which is the baseloss flow. This is another degree of freedom, which can be fitted at will, since the models are only calibrated on discharge. What I want to say is: Conceptual models can't "invent" water (e.g. by water transfer from a different catchment) but water can be removed at will. So personally, I don't think that a "leaking catchment" (L. 518) has to be a problem, or?

This is a very interesting point and something that we have discussed. The particular models that we benchmark against here were constrained to not remove any more water than the maximum defined by the input potential evapotranspiration. You are correct that conceptual models often have a baseloss flow, however, in the models used for comparison here, baseloss flow parameters were set to zero (i.e. excluded) and there is no baseloss flow. We have updated the text to read: "*One of the key hydrological conditions that hydrological models struggle with is the lack of closure of the catchment water balance. The conceptual models we test here explicitly maintain mass balance. They define the topographic surface water catchment as the surface over which water is conserved, i.e. the surface water catchment is not expected to leak, nor should any water enter the catchment other than through measured precipitation.*" (**L345-248**)

- L 533ff "Alternatively, the fact that both LSTMs and conceptual models struggle in catchments where data does not meet the water balance constraints might suggest that human impacts on the hydrograph are ultimately unpredictable, such as abstraction and effluent returns." Or maybe just unpredictable from the given model inputs? Even if anthropogenic influences are included in the catchment attributes, water extraction is most likely a dynamic process and would require additional dynamic inputs. But conceptually, I don't see why a data-driven model should not be able to learn this process? The process is either driven by physical processes or by a human factor, which is most likely driven by a management plan. Both things could in theory be learned, given enough (informative) inputs.

We have reformulated the argument to reflect the reviewers comments. "*This suggests that the underlying data does not contain sufficient information to model the full range of processes that influence the hydrograph in these catchments, including groundwater and abstractions. The catchment averaged information on soil texture (sand-silt-clay) provides a coarse proxy for catchment porosity. Furthermore, further data, such as groundwater time-series, might be necessary to obtain more accurate discharge predictions. We suggest that different input data sets should be tested to try and improve LSTM performances enabling the LSTM to more properly account for the complex percolation and infiltration dynamics in these catchments.*" (**L450-456**).

- L 539 Do you mean to link Figure 10? The link to Figure 11 is not clear to me.

We have updated this to link to the Budyko-curve figure (*Figure 5*) "*We tested whether the LSTM was better able to simulate discharge in catchments with "excess" water (i.e. the points below the curved lines in Fig. 5, which are then represented by the orange kernel density estimate in Fig. 6).* " (**L365-366**)

- Figure 11: x-axis label (NSE) should be in capital letters

We have updated this as proposed.

- L 551 Again, since this is the conclusion, worth that your comparison is biased (towards the lumped hydrology models), since you compare your hold-out period to their calibration period.

We included various statements critically engaging with the intercomparison of the two experiments. We updated the manuscript to reflect the reviewers comment:

- **L223-225**: "*Therefore, the LSTM is evaluated on out-of-sample (in time) data, whereas, the conceptual model parameters were calibrated on data included in the evaluation period (in-sample evaluation).*"
- **L415-416**: "*Another difference is that the LSTM diagnostic scores are calculated on out-of-sample predictions, compared with the in-sample predictions for the benchmark conceptual models.* "

References:

Best, M. J., Abramowitz, G., Johnson, H. R., Pitman, A. J., Balsamo, G., Boone, A., ... & Vuichard, N. (2015). The plumbing of land surface models: benchmarking model performance. Journal of Hydrometeorology, 16(3), 1425-1442.

Beven, Keith. "A manifesto for the equifinality thesis." Journal of hydrology 320.1-2 (2006): 18-36.

**Anonymous Reviewer #3:**

Comments/Text of Anonymous Referee posted in **black**, our text in **blue**.

This paper describes two versions of a national scale deep learning hydrological model for GB and compares them to 4 conceptual hydrological models from the FUSE framework. The effectiveness of LSTM has been well established in previous studies, and so the novelty of this paper lies in its application to GB catchments. As the code, data and outputs are all freely available, I consider this to be a useful study to hydrologists concerned with modelling GB catchments. I wonder if given the limited scientific insights of this paper may be better placed in the Journal of Hydrology: Regional studies, or Environmental Modelling and Software rather than HESS.

I would like to commend the authors on a very clearly written paper- it was very easy to follow and understand.

We thank Reviewer #3 for their comments and effective summary of the paper. We take on board the claims about scientific novelty and have updated the paper to reduce the emphasis on outlining the performance improvement of the LSTM compared with the conceptual models. We have made four major changes to the paper based on the reviewers comments.
   1) We have rewritten the *Section 2 Methods*, shortening *Section 2.3 An Overview of the LSTM and EALSTM* and moving a large part of the model description to *Appendix A: LSTM and EA LSTM Model Description*.
   2) We have made the *Section 3 Results* more concise, reducing the number of figures in the main body of text. We have focussed more attention on the interesting patterns of LSTM performance, and away from the improved performance of the LSTM compared with the conceptual models. We have also focussed attention towards interpreting the model performances in the drier, groundwater dominated catchments of the South East.
   3) We have expanded the discussion of these results, exploring more clearly our three research questions:
       a) *Section 4.1.1: How well do LSTM-based models simulate discharge in Great Britain?*
       b) *Section 4.1.2: How does the LSTM performance compare with the conceptual models used as benchmark?*
       c) *Section 4.1.3: Can we extract information from the spatial and temporal patterns in diagnostic measures?*
   4) We have more critically engaged with our experimental structure and the intercomparison with the lumped conceptual models (*Section 2.4.1: Benchmark Models & Section 4.1.2: How does the LSTM performance compare with the conceptual models used as benchmark?*).

My major criticism of the paper is that the authors never demonstrate the model's applicability to a changing climate. Even if the application of LSTM (and all models that rely entirely on calibration) is only for near term flood forecasting, it is likely that we will be modelling events outside of the training data of the model with increasing frequency. I think that an alternative calibration/validation strategy should be examined where extreme events are left out of the calibration of the model, to provide some confidence in its ability to model beyond its training dataset.

We agree that understanding model performances on out-of-sample events is an exciting area of study. However, we believe the calls from all reviewers for a more concise paper mean that a complete exploration of this question is beyond the scope of this study.

My other major criticism is that the authors never **discuss the insights gained from the LSTM model**. There is no discussion of the sensitivity of the model to the different inputs and how the model ends up being structured. **They never provide any evidence to answer their third research question**. I think this would add a lot more value to the paper and make it worthy of

publication in HESS. In the conclusion the authors state that this will come in a subsequent paper, but I think it would be more valuable here (and some of the detail of the calibration/validation could be moved to the supplementary information).

We thank Reviewer #3 for their identification of research question 3 as the most scientifically valuable contribution of the paper. We have updated the results to more accurately address our third research question: "*Can we extract information from the spatial and temporal patterns in diagnostic measures? e.g. What is the relationship between LSTM performance and catchment attributes?*".

Section *3.3: In what hydrological conditions do model performances differ?* outlines the spatial (**L317-338**), and temporal (**L325-330**) patterns in model performances and explores the catchment attributes that correlate with model performances (**L335-344**). We explore in greater depth the impact of water balance closure on LSTM performances, highlighting that the LSTM performances are worse in catchments where the water balance does not close than in those catchments that are not "leaky" (Section 3.3.1, **L345-379**). We have also substantially increased the discussion of these results, outlining the reasons for the differences in model performances and learning from the differences in model performances. *Section 4.14* explicitly addresses the question that the Reviewer has highlighted. We first examine the performance differences in NE Scotland (**L430-433**) before exploring in more depth the conditions and explanation for differences in LSTM performances in the SE of England compared to elsewhere in GB (**L439-457**). We then explore two other aspects which warrant further discussion, firstly, improved performance in summer months (**L458-469**) and in catchments with a strong agricultural signal (**L470-475**).

**Some more specific comments follow:**

line 19: There are more modern PBSD models than SHE. Reference Parflow, SUMA, SHETRAN, Hydrogeosphere etc.
We have updated these references on **L16-18**.

line 77: there are only 3 research questions
We have updated this as proposed.

Figure 1: You can format text in python to include superscripts "$mm\ day^{-1}$". Reduce point size- they are overlapping and obscuring each other.
We have removed this figure in order to make the paper more concise.

Table 1: Nice! Very useful table. Temperature should be referred to with a capital T. Should Xt actually be Xn if it is representing the concatenation of dynamic and static input data for a single catchment?
We have updated *Table 2* as proposed. $X_t$ has become $X_{t,n}$ to reflect that it contains information for the target time period and the target catchment. Great spot!

line 176: Include the link to the prediction and error metrics at the end of the article too.
We have updated this as proposed (**L520**).

Table 2: Why these attributes? Was LSTM sensitive to all of these?
We have updated the text to read: "*These attributes were chosen to reflect hydrological information that the model can use to distinguish between catchment rainfall-runoff behaviours \citep{kratzert2019_ealstm}.* " (**L154-155**).

line 220: What is an epoch? how does this relate to number of catchments/years of data?
We have updated the text to define an epoch: **L176-178:** "*An epoch reflects a single pass of the training dataset through the model, such that every sample in the training dataset has been used to update the model weights. This reflects the fact that during the training of DL models, the data are often split into batches to allow large datasets to be read into memory.* "

Table 3: How is statistical significance calculated here? Double check that it is the appropriate method.
We used the "Paired Wilcoxon Test" (Scipy Function with the "alternative" parameter set to "two-sided"). We clarified this in the manuscript: **L262**.

Figure 3: Nice figure
Thank you!

line 366: I don't think that the catchments with significant snowfall should be included in the comparison if the snow modules of the conceptual models have not been turned on- this does not seem like a fair comparison. Recalculate the statistics leaving these catchments out.

This is a very interesting point and something that we have expanded our discussion about. One of the key benefits of using the LSTMs, and data-driven approaches, is that we do not need to pre-specify the modules/structures that need to be included. Instead we can learn this from the data. We believe that by providing a GB-wide benchmark it is important to show the performance across all of the catchments that have been modelled, especially since these results are being published as a comparison for future work.

We have recalculated the statistics excluding catchments with significant snow-processes and the results do not significantly change. We propose to leave the comparison as is for the reasons outlined above.

```
Original data all catchments (N=518)
\begin{tabular}{lrrrrrrr}
\toprule
{} &  nse &  bias\_error &  std\_error &  correlation &   fms &      flv &    fhv \\
\midrule
TOPMODEL   & 0.76 &       -0.04 &      -0.10 &         0.88 &  5.70 &    42.22 & -13.04 \\
ARNOVIC    & 0.78 &        0.06 &      -0.10 &         0.90 &  2.25 &   -60.34 & -14.66 \\
PRMS       & 0.77 &        0.03 &      -0.03 &         0.89 & 35.24 &  -315.25 & -15.11 \\
SACRAMENTO & 0.80 &       -0.01 &      -0.07 &         0.90 & 27.91 &  -195.92 & -16.19 \\
EALSTM     & 0.86 &       -0.02 &      -0.10 &         0.94 & -6.29 &    23.61 & -10.81 \\
LSTM       & 0.88 &       -0.02 &      -0.09 &         0.94 & -3.67 &    26.34 &  -9.09 \\
\bottomrule
\end{tabular}

Stations with frac_snow <= 0.03 (N=450)
\begin{tabular}{lrrrrrrr}
\toprule
{} &  nse &  bias\_error &  std\_error &  correlation &   fms &      flv &    fhv \\
\midrule
TOPMODEL   & 0.77 &       -0.03 &      -0.09 &         0.89 &  6.35 &    42.27 & -11.97 \\
ARNOVIC    & 0.79 &        0.07 &      -0.09 &         0.90 &  3.37 &   -56.83 & -14.11 \\
PRMS       & 0.78 &        0.04 &      -0.02 &         0.90 & 36.45 &  -307.09 & -14.27 \\
SACRAMENTO & 0.81 &       -0.01 &      -0.06 &         0.91 & 29.20 &  -194.96 & -15.61 \\
EALSTM     & 0.86 &       -0.02 &      -0.10 &         0.94 & -6.12 &    25.11 &  -9.82 \\
LSTM       & 0.88 &       -0.02 &      -0.08 &         0.95 & -3.67 &    27.95 &  -8.48 \\
\bottomrule
\end{tabular}

Stations with frac_snow > 0.03 N=(68)
\begin{tabular}{lrrrrrrr}
\toprule
{} &  nse &  bias\_error &  std\_error &  correlation &   fms &      flv &    fhv \\
\midrule
TOPMODEL   & 0.71 &       -0.06 &      -0.14 &         0.85 & -0.96 &    41.42 & -15.59 \\
ARNOVIC    & 0.75 &        0.04 &      -0.14 &         0.87 & -7.35 &   -83.36 & -18.01 \\
PRMS       & 0.73 &       -0.00 &      -0.09 &         0.86 & 25.33 &  -361.99 & -19.45 \\
SACRAMENTO & 0.76 &       -0.03 &      -0.12 &         0.87 & 19.61 &  -202.01 & -19.60 \\
EALSTM     & 0.84 &       -0.03 &      -0.16 &         0.93 & -8.17 &    15.66 & -16.88 \\
LSTM       & 0.87 &       -0.03 &      -0.11 &         0.93 & -4.11 &    21.38 & -12.35 \\
\bottomrule
\end{tabular}
```

line 367-371: this is a repetition of the previous paragraph.
We have removed the repeated sentence.

Figure 5: cut. This is a long paper with a lot of figures. I don't think this figure adds much to the maps.
We have moved this figure to the Appendix (*Fig. E2*) as proposed. We have kept the CDFs of catchment metrics (Figure 1), the maps showing seasonal NSE scores (Figure 2). Previous figures that are no longer included have been moved to Appendices, Appendix E: Spatial Performances of Error Metrics. We have reduced the focus of the results on the outperformance of the LSTM compared with the benchmark models.

Figure 6. Label missing on the colorbar
We have updated the colorbar as proposed on *Figure 2*.

Discussion: Cut all references to the physically based models. The comparisons are not rigorous and so should not be presented.
We have removed this section from the manuscript.

Figure 9: significant correlations are not clear. consider showing this in an alternative way.
We have increased the size of the marks (*) as proposed on *Figure 4*.

line 537: I think this is the most interesting point in the whole paper - I would love to read a lot more about this in the discussion.

We have significantly expanded the discussion about the information included in the data. Section *4.1.3 Can we extract information from the spatial and temporal patterns in diagnostic measures?* explicitly deals with the information available in the underlying dataset, outlining what we can and cannot learn from the CAMELS-GB dataset. We have proposed two hypotheses about information that the LSTM captures that could explain performance improvements in summer months (**L457-469**), and in catchments with a strong agricultural signal (**L470-475**). We have also explored what are the limits to the information available in the underlying data, exploring the difficulty in modelling groundwater dominated catchments only with meteorological datasets and coarse geological information (**L447-456**). We offer an expansive concluding paragraph for this discussion section, outlining the conditions we should focus our model improvement efforts on given the information available in the underlying dataset (**L475-481**).

**Uncertainty**: I would like to see some discussion of training models to uncertain flows and uncertain inputs.

We agree that addressing uncertainty in inputs and outputs is of vital importance for hydrological modelling. While we feel that full treatment of uncertainty in inputs and outputs is beyond the scope of this manuscript, we have addressed this important point in two ways.
  1) Expanded discussion of recent advances in LSTM based modelling with uncertainty inputs.
      a) In particular, see Kratzert et al (2021) on the performance boost of using multiple rainfall datasets, highlighting that the LSTM can flexibly incorporate new information from highly co-linear input datasets (**L593-594**: "*Uncertainties in observations can be estimated and accounted for by using multiple forcing products \citep{kratzert2021synergy} or by resampling the input data.* ").
      b) Secondly, work by Klotz et al (2021) demonstrating three different methods for uncertainty quantification (**L603-605**: "*A more principled treatment of uncertainty, which benchmarks various methods for using DL models to directly simulate a distribution can be found in \citet{klotz2020}.*").

2) In *Appendix D: Model Uncertainty* **L574-594** we have explored the uncertainty represented by the variability in the ensemble of 8 LSTM models.
   a) We present the spatial distribution of ensemble variability as a % of discharge ("Coefficient of Variability" - *Figure D2*)
   b) We present the standard deviation of ensemble simulations for different flow exceedances (*Figure D1*)
   c) We presented hydrographs in *Appendix 2* with uncertainty bands reflecting one standard deviation of ensemble member simulations.